# All-or-nothing statistical and computational phase transitions in sparse spiked matrix estimation

**Jean Barbier**
International Center for Theoretical Physics
Strada Costiera 11, 34151 Trieste, Italy
`jbarbier@ictp.it`

**Nicolas Macris**
Ecole Polytechnique Fédérale de Lausanne
CH 1015 Lausanne, Switzerland
`nicolas.macris@epfl.ch`

**Cynthia Rush**
Department of Statistics, Columbia University
New York, NY 10025
`cynthia.rush@columbia.edu`

## Abstract

We determine statistical and computational limits for estimation of a rank-one matrix (the spike) corrupted by an additive gaussian noise matrix, in a sparse limit, where the underlying hidden vector (that constructs the rank-one matrix) has a number of non-zero components that scales sub-linearly with the total dimension of the vector, and the signal-to-noise ratio tends to infinity at an appropriate speed. We prove explicit low-dimensional variational formulas for the asymptotic mutual information between the spike and the observed noisy matrix and analyze the approximate message passing algorithm in the sparse regime. For Bernoulli and Bernoulli-Rademacher distributed vectors, and when the sparsity and signal strength satisfy an appropriate scaling relation, we find all-or-nothing phase transitions for the asymptotic minimum and algorithmic mean-square errors. These jump from their maximum possible value to zero, at well defined signal-to-noise thresholds whose asymptotic values we determine exactly. In the asymptotic regime the statistical-to-algorithmic gap diverges indicating that sparse recovery is hard for approximate message passing.

## 1 Introduction and setting

In modern machine learning and high dimensional statistics one often faces regression, classification, or estimation tasks, where the dimension of the feature vectors is much larger than the effective underlying dimensionality of the structure at hand. For example, hand-written MNIST digits are presented as vectors consisting of $28 \times 28$ pixels, in other words, they are binary vectors with $784$ dimensions, whereas [1, 2] estimate their effective dimension to be in the orders of 10's. Similarly the ISOMAP face database consists of images (256 levels of gray) of size $64 \times 64$, i.e., vectors in $\mathbb{R}^{4096}$, whereas the correct intrinsic dimension is only 3 (for the vertical, horizontal pause and lighting direction). Natural images, which are generally sparse in a wavelet basis [3], are another popular example of low effective dimensionality. For natural images, a very simple model of low-dimensional structure, namely vectors with a sparse number of non-zero components, has proven immensely useful for studying these types of data structures and has led to the development of the whole area of compressed sensing [4, 5]. Similarly, matrix completion can be performed successfully when the number of sampled matrix elements is much smaller than the total number of elements, as long as one assumes the matrix is low-rank [6].

These and other developments have amply justified the "bet on sparsity principle", which, in a nutshell, says that intrinsic low-dimensionality is often a crucial ingredient for the interpretability of high dimensional statistical models [7, 8]. In this context, it is of great importance to determine computational limits of estimation and to establish fundamental information theoretical (i.e., statistical) limits as benchmarks. Broadly speaking, exact results in the direction of computational or information theoretic limits usually fall in two categories. The first direction, traditional in statistics and computer science, derives *finite size* bounds on thresholds marking the onset of feasible signal recovery or learning [9, 10]. Such results usually leave out exact constants or do not always give the exact asymptotics. The second approach, is an *average case* approach (in the spirit of the statistical mechanics treatment of high dimensional systems), that models feature vectors by a *random ensemble*, taken as a set of random vectors with independently identically distributed (i.i.d.) components, and a small but fixed fraction of non-zero components. For example, the distribution might be a Bernoulli distribution, denoted $\mathrm{Ber}(\rho_n)$ with $0 < \rho_n < 1$ and $\rho_n \to \rho > 0$ fixed, as the dimension of the vectors $n \to +\infty$. In Bayesian settings with known priors and hyper-parameters this approach has been highly successful, yielding exact formulas for the mutual information and minimum mean-square error (MMSE), as well as exact expressions (with constants) for statistical and computational message passing phase transition thresholds in the limit of *infinite dimensions* [11]. While the mathematical analysis of this approach is well developed in compressed sensing, generalized linear estimation, or rank-one noisy matrix and tensor estimation [12–23], the cited works all fall short of addressing the "true" sparse limit where $\rho_n \to 0$ instead of the limit being fixed (i.e., $\rho_n \to \rho > 0$) as $n \to +\infty$; to be more precise we manage to tackle the regime $\rho_n = \Omega(n^{-\beta})$ for $\beta \in [0, 1/6)$ for the information-theoretic analysis, and $\rho_n = \Omega((\ln n)^{-\alpha})$ for any positive fixed $\alpha$ for the algorithmic results. The terminology "true sparsity" is employed in order to emphasize this contrast. To the best of our knowledge the only works addressing this "true" sparse limit, in the average case approach for statistical phase transitions, are [24, 25] which consider linear regression.

In this work, we address the issue of "true" sparsity in the average case approach for the problem of rank-one matrix estimation from noisy observations of the entries. Low-rank matrix estimation (or factorization) is an important problem with numerous applications in image processing, principal component analysis (PCA), machine learning, DNA microarray data, and tensor decompositions. We determine information theoretic limits of the problem as well as computational limits of an approximate message passing algorithm [26–31] for signal estimation in the case of a noisy symmetric rank-one matrix model. Let us now introduce the model.

**Setting:** In the *sparse spiked Wigner matrix model* we consider a sparse signal-vector $\boldsymbol{X} = (X_1, \ldots, X_n) \in \mathbb{R}^n$ with i.i.d. components distributed according to $P_{X,n} = \rho_n p_X + (1 - \rho_n)\delta_0$, where $\delta_0$ is the Dirac mass at zero and $\rho_n \in (0, 1]^{\mathbb{N}}$ is a sequence of weights that will eventually tend to 0; the signal has in expectation a *sub-linear* number $n\rho_n$ of non-zero components. For the distribution $p_X$ we assume that $i$) it is independent of $n$, $ii$) it has finite support in an interval $[-S, S]$, $iii$) it has second moment equal to 1 (without loss of generality). One has access to the symmetric data matrix $\boldsymbol{W} \in \mathbb{R}^{n \times n}$ with noisy entries

$$\boldsymbol{W} = \sqrt{\frac{\lambda_n}{n}} \boldsymbol{X} \otimes \boldsymbol{X} + \boldsymbol{Z}, \text{ or componentwise } W_{ij} = \sqrt{\frac{\lambda_n}{n}} X_i X_j + Z_{ij}, \quad 1 \le i < j \le n \quad (1)$$

where $\lambda_n > 0$ controls the strength of the signal and the noise is i.i.d. gaussian $Z_{ij} \sim \mathcal{N}(0, 1)$ for $i < j$ and symmetric $Z_{ij} = Z_{ji}$. Notice that the matrix $\boldsymbol{W}$ can be viewed as a sum of a gaussian matrix from the Wigner ensemble perturbed by a rank-one matrix, $\boldsymbol{X}\boldsymbol{X}^\mathsf{T}$ (the "spike"). We focus, in particular, on binary $\boldsymbol{X}$ generated with i.i.d. Bernoulli entries $X_i \sim P_{X,n} = \mathrm{Ber}(\rho_n)$, or Bernoulli-Rademacher entries, $X_i \sim P_{X,n} = (1 - \rho_n)\delta_0 + \rho_n \frac{1}{2}(\delta_{-1} + \delta_1)$. In the Bayesian setting, we suppose that the prior $P_{X,n}$ and hyper-parameters are known. As we will see, when $\rho_n \to 0$, non-trivial estimation is possible only when $\lambda_n \to +\infty$.

The goal is to estimate the sparse spike $\boldsymbol{X} \otimes \boldsymbol{X}$ from the data $\boldsymbol{W}$. In the spiked Wigner model with linear sparsity, a class of polynomial-time algorithms, referred to as approximate message passing or AMP, have been shown to provide Bayes-optimal signal estimation for some problem settings asymptotically as $n \to +\infty$ [32–34]. Moreover, AMP algorithms have been applied successfully for signal recovery to a number of other low-rank matrix estimation problems [35–38] and, based on bold conjectures from the statistical physics literature, it is suggested that the estimation performance of AMP is the best among polynomial-time algorithms. Again, AMP is also provably optimal in some

parameters regimes. In this work, we study the properties of an AMP algorithm designed for signal estimation for the spiked Wigner matrix model in the sub-linear sparsity regime and compare its performance to benchmarks established by the information theoretic limits. This analysis provides a better understanding of the computational vs. theoretical gaps posed by the problem.

**Some background and related work:** In recent years, there has been much progress in understanding such spiked matrix models, which have played a crucial role in the analysis of threshold phenomena in high-dimensional statistical models for almost two decades, but most of this work has focused on standard settings, by which we mean problem settings where the distribution $P_X$ is fixed independent of the problem dimension $n$. This means that the expected number of non-zero components of $\boldsymbol{X}$, even if "small", will scale *linearly* with $n$. Early rigorous results found in [39] determined the location of the information theoretic phase transition point in a spiked covariance model using spectral methods, and [40, 41] did the same for the Wigner case. More recently, the information theoretic limits and those of hypothesis testing have been derived, with the additional structure of sparse vectors, for large but finite sizes [42–44]. A lot of efforts have also been devoted to computational aspects of sparse PCA with many remarkable results [33, 43–50]. The picture that has emerged is that the information theoretic and computational phase transition regimes are not on the same scale and that the computational-to-statistical gap diverges in the limit of vanishing sparsity. However, the exact thresholds with constants as well as the behaviour of the mean-square errors remained unknown.

Using heuristic methods from the statistical physics of spin glass theory (the so-called replica method [51]), the authors of [52] observed an interesting phenomenology of the information theoretical and computational limits with sharp phase transitions as $n \to +\infty$. The rigorous mathematical theory of these phase transitions is now largely under control. On one hand, an approximate message passing algorithm for signal recovery can be rigorously analyzed via its state evolution [27,28,53], and on the other hand, the asymptotic mutual information per variable between the hidden spike and data matrices has been rigorously computed in a series of works using various methods (cavity method, spatial coupling, interpolation methods, PDE techniques) [16–23, 54–57]. The information theoretic phase transitions are then signaled by singularities, as a function of the signal strength, in the limit of the mutual information per variable when $n \to +\infty$. The phase transition also manifests itself as a jump discontinuity in the minimum mean-square error (MMSE)[1]. Once the mutual information is known, it is usually possible to deduce the MMSE using so-called I-MMSE relations [58, 59]. Essentially, the MMSE can be accessed by differentiating the mutual information with respect to the signal-to-noise strength. Closed form expressions for the asymptotic mutual information therefore allow to benchmark the fundamental information theoretical limits of estimation. We also point the reader towards the works [60–62] which derive limits of detecting the presence of a spike in a noisy matrix, rather than estimating it.

Finally, similar phase transitions in sub-linear sparsity regimes for binary signals have been studied in the context of high-dimensional linear regression or compressed sensing for support recovery [24, 25]. These works focus on the MMSE and prove the occurrence of the $0 - 1$ phase transition, which they called an "all-or-nothing" phenomenon. We note that our approach is technically very different in that it determines the variational expressions for mutual informations and finds the transitions as a consequence. Moreover these works do not deal with algorithmic phase transitions, while we consider here the one of AMP.

**Our contributions:** We provide new results in sparse limits along two main lines:

- The exact statistical threshold for the sharp all-or-nothing statistical transition at the level of the MMSE. This follows from a rigorous derivation of the mutual information in the form of a variational problem.

- The AMP algorithmic threshold and all-or-nothing transition at the level of the AMP mean-square error. This follows from a "finite sample" analysis of the approximate message passing algorithm, allowing to rigorously track its performance in sparse regimes.

Let us explain these contributions in detail.

In this work, we identify the correct *scaling regimes* of vanishing sparsity and diverging signal strength in which non-trivial information theoretic and algorithmic AMP phase transitions occur. Moreover, we determine the statistical-to-algorithmic gap in the scaling regime. These scalings, thresholds, as well as formulas for the mutual information, were first heuristically and numerically derived in [52] using the non-rigorous replica method of spin-glass theory and the state evolution equations for AMP. However, it must be stressed that, not only were these calculations far from rigorous, but more importantly the limit $n \to +\infty$ is taken first for a fixed parameter $\rho_n = \rho$, and the sparse limit $\rho \to 0_+$ is taken only after. Although the thresholds found in this way agree with our derivations, this is far from evident a priori. In contrast, our results are entirely rigorous and valid in the truly sparse limit. Therefore the picture found in [52] is fully vindicated. In addition, we also establish that the MMSE and AMP phase transitions are of the all-or-nothing type, a novelty of the present work.

The information theoretic analysis is done via the adaptive interpolation method [19, 20, 22], first introduced in the non-sparse matrix estimation problems, to provide for the sparse limit, closed form expressions of the mutual information in terms of low-dimensional variational expressions (theorem 1 in section 2). That the adaptive interpolation method can be extended to the sparse limit is interesting and not a priori obvious. Using the I-MMSE relation and the solution of the variational problems for Bernoulli and Bernoulli-Rademacher distributions of the sparse signal, we then find that the MMSE displays an all-or-nothing phase transition (corollary 1) and we determine the exact threshold (with constants).

A useful property of AMP is that in the large system limit $n \to +\infty$, its performance can be exactly characterized and rigorously analyzed through its so-called *state evolution*. When $\rho_n \to \rho > 0$, the validity of the state evolution analysis for AMP for low-rank matrix estimation follows from the standard AMP theory [27, 28] (with some additional work needed to deal with technicalities relating to the algorithm's initialization [34]), however, in the sub-linear sparsity regime considered here, proving the validity of the state evolution characterization requires a new and non-trivial analysis using "finite sample" techniques, first developed in [63]. We find that the algorithmic MSE, denoted $\mathrm{MSE}_{\mathrm{AMP}}$ displays an all-or-nothing transition as well and we determine the scaling of the threshold (the constant being obtained numerically). Interestingly, the transition is on a very different signal-to-noise scale as compared to the MMSE (theorem 2 found in section 3).

Let us describe in a bit more detail the sparse regimes we study and the corresponding thresholds. To gain some intuition, we first note that for sub-linear sparsity, phase transitions can appear only if the signal strength tends to infinity. This can be seen from the following heuristic argument: notice that the total signal-to-noise ratio per non-zero component[2] scales as $(\lambda_n/n)\rho_n^2 n^2/(\rho_n n) = \lambda_n \rho_n$, meaning that $\lambda_n \to +\infty$ is necessary in order to have enough energy to estimate the non-zero components. Our analysis shows that non-trivial information theoretic and AMP phase transitions occur at different scales:

- **Statistical phase transition regime:** While our results are more general (see appendix A and theorem 3) our main interest is in a regime of the form

$$\lambda_n = 4\gamma|\ln\rho_n|\rho_n^{-1}, \qquad \rho_n = \Omega(n^{-\beta}), \tag{2}$$

  for $\beta, \gamma \in \mathbb{R}_{\geq 0}$ and $\beta$ small enough. We prove that in this regime a phase transition occurs as function of $\gamma$.

- **Algorithmic AMP phase transition regime:** We control the performance of AMP for a number of time-iterations $t = o(\frac{\ln n}{\ln \ln n})$ and rigorously prove that the all-or-nothing transition occurs for

$$\lambda_n = w\rho_n^{-2}, \qquad \rho_n = \Omega((\ln n)^{-\alpha}), \tag{3}$$

  where $w, \alpha \in \mathbb{R}_{\geq 0}$ are fixed constants (note that we can take any $\alpha > 1$). Controlling the AMP iterations in this regime is already highly non-trivial, however, we conjecture that the result still holds when $\rho_n = \Omega(n^{-\beta})$ for $\beta > 0$ small enough, but refining the analysis in appendix K to find the stronger result is left for future work.

The relation $\lambda_n \sim \rho_n^{-2}$ for the AMP threshold was obtained in [52] based on a stability analysis of the linearized state evolution. However, we recall that in their setting $\rho_n = \rho$, $n \to +\infty$, and not only is the sparse limit $\rho \to 0_+$ taken after the high-dimensional limit, but also the AMP iterations are not controlled. In appendix G in the supplementary material we provide a simpler alternative argument that does not require linearizing the recursion.

We focus in particular on binary signals with $P_{X,n}$ equal to $\mathrm{Ber}(\rho_n)$ or Bernoulli-Rademacher $(1 - \rho_n)\delta_0 + \rho_n \frac{1}{2}(\delta_{-1} + \delta_1)$. For these distributions we prove the existence of all-or-nothing transitions for the MMSE and $\mathrm{MSE}_{\mathrm{AMP}}$ for the specific sparsity regimes stated above. This is illustrated in figures 1 and 2, found in sections 2 and 3, which display, for the Bernoulli prior, the explicit asymptotic values to which the finite $n$ mutual information and MMSE converge. The results are similar for the Bernoulli-Rademacher distribution. In figure 1, we see that as $\rho_n \to 0_+$ the (suitably normalized) mutual information approaches the broken line with an angular point at $\lambda/\lambda_c(\rho_n) = 1$ where $\lambda_c(\rho_n) = 4|\ln \rho_n|/\rho_n$. Moreover the (suitably normalized) MMSE tends to its maximum possible value 1 for $\lambda/\lambda_c(\rho_n) < 1$, develops a jump discontinuity at $\lambda/\lambda_c(\rho_n) = 1$, and takes the value 0 when $\lambda/\lambda_c(\rho_n) > 1$ as $\rho_n \to 0$. In figure 2, we observe the same behavior for $\mathrm{MSE}_{\mathrm{AMP}}$ as a function of $\lambda/\lambda_{\mathrm{AMP}}(\rho_n)$, but now the algorithmic threshold is $\lambda_{\mathrm{AMP}}(\rho_n) = 1/(e\rho_n^2)$, where the constant $1/e$ is approximated numerically. Note that the same asymptotic behavior is observed in the related problem of finding a small hidden community in a graph, see figure 5 in [64].

# 2 Statistical phase transition

The phase transition manifests itself as a singularity (more precisely a discontinuous first order derivative) in the mutual information $I(\boldsymbol{X} \otimes \boldsymbol{X}; \boldsymbol{W}) = H(\boldsymbol{W}) - H(\boldsymbol{W}|\boldsymbol{X} \otimes \boldsymbol{X})$. Note that because the data $\boldsymbol{W}$ depends on $\boldsymbol{X}$ only through $\boldsymbol{X} \otimes \boldsymbol{X}$ we have $H(\boldsymbol{W}|\boldsymbol{X} \otimes \boldsymbol{X}) = H(\boldsymbol{W}|\boldsymbol{X})$ and therefore $I(\boldsymbol{X} \otimes \boldsymbol{X}; \boldsymbol{W}) = I(\boldsymbol{X}; \boldsymbol{W})$. From now on we use the form $I(\boldsymbol{X}; \boldsymbol{W})$.

To state the result, we define the *potential function*:

$$i_n^{\mathrm{pot}}(q, \lambda, \rho) \equiv \frac{\lambda}{4}(q - \rho)^2 + I_n(X; \sqrt{\lambda q}X + Z), \tag{4}$$

where $I_n(X; \sqrt{\lambda q}X + Z)$ is the mutual information for a scalar gaussian channel, with $X \sim P_{X,n}$ and $Z \sim \mathcal{N}(0, 1)$. The mutual information $I_n$ is indexed by $n$ because of its dependence on $P_{X,n}$.

**Theorem 1** (Mutual information for the sparse spiked Wigner model). *Let the sequences $\lambda_n$ and $\rho_n$ verify (2) with $\beta \in [0, 1/6)$ and $\gamma > 0$. There exists $C > 0$ independent of $n$ such that*

$$\frac{1}{\rho_n|\ln \rho_n|}\left|\frac{1}{n}I(\boldsymbol{X}; \boldsymbol{W}) - \inf_{q \in [0, \rho_n]} i_n^{\mathrm{pot}}(q, \lambda_n, \rho_n)\right| \leq C \frac{(\ln n)^{1/3}}{n^{(1-6\beta)/7}}. \tag{5}$$

The mutual information is thus given, to leading order, by a one-dimensional variational problem

$$I(\boldsymbol{X}; \boldsymbol{W}) = n\rho_n|\ln \rho_n| \inf_{q \in [0, \rho_n]} i_n^{\mathrm{pot}}(q, \lambda_n, \rho_n) + \text{correction terms}.$$

The factor $\rho_n|\ln \rho_n|$ is related to the entropy (in nats) of the support of the signal given by $-n(\rho_n \ln \rho_n + (1 - \rho_n)\ln(1 - \rho_n))$, which behaves like $n\rho_n|\ln \rho_n|$ for $\rho_n \to 0_+$. In particular, for both the *Bernoulli* and *Bernoulli-Rademacher* distributions an analytical solution of the variational problem, given in appendix F, shows that $(\rho_n|\ln \rho_n|)^{-1}\inf_{q \in [0, \rho_n]} i_n^{\mathrm{pot}}(q, \lambda_n, \rho_n)$ tends to the singular function $\gamma\mathbb{I}(\gamma \leq 1) + \mathbb{I}(\gamma \geq 1)$ as $n \to +\infty$ and $\rho_n \to 0$, where we recall that $\lambda_n = 4\gamma|\ln \rho_n|\rho_n^{-1}$. See figure 1. Let us mention that $\beta < 1/6$ is probably not a fundamental limit to the validity of the result but rather is an artefact of the sub-optimality of our proof technique.

We now turn to the consequences for the MMSE. It is convenient to work with the "matrix" MMSE defined as $\mathrm{MMSE}((X_iX_j)_{i<j}|\boldsymbol{W}) \equiv \mathbb{E}\|(X_iX_j)_{i<j} - \mathbb{E}[(X_iX_j)_{i<j}|\boldsymbol{W}]\|_{\mathrm{F}}^2$. This quantity satisfies the I-MMSE relation [58, 59] (see also appendix I for a self-contained derivation),

$$\frac{d}{d\lambda_n}\frac{1}{n}I(\boldsymbol{X}; \boldsymbol{W}) = \frac{1}{2n^2}\mathrm{MMSE}((X_iX_j)_{i<j}|\boldsymbol{W}).$$

In appendix J we prove:

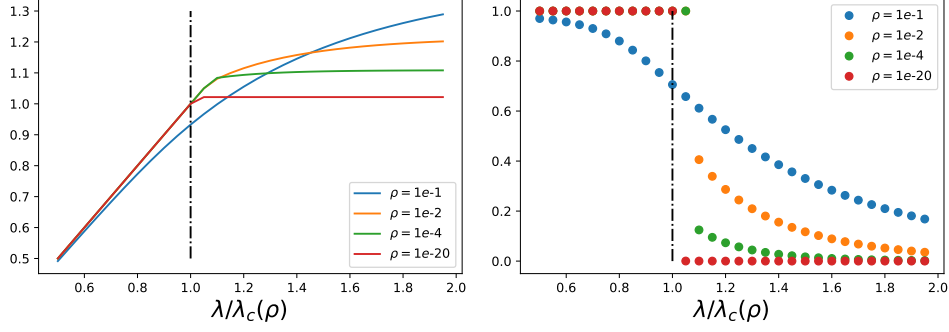

Figure 1: A sequence of suitably normalized asymptotic mutual information $(\rho|\ln\rho|)^{-1}\inf_{q\in[0,\rho]}i_n^{\mathrm{pot}}(q,\lambda,\rho)$ (left) and associated minimum mean-square error (MMSE) $\rho^{-2}\frac{d}{d\lambda}\inf_{q\in[0,\rho]}i_n^{\mathrm{pot}}(q,\lambda,\rho)$ (right) curves as a function of $\lambda/\lambda_c(\rho)$ with $\lambda_c(\rho)=4|\ln\rho|/\rho$ for the model $X_i\sim\mathrm{Ber}(\rho)$ and various $\rho=\rho_n$ values (that can be converted to signal sizes through $\rho_n=\Omega(n^{-\beta})$ given a sparsity scaling $\beta$) using the potential function defined in (4). These are the curves towards which, respectively, the finite size mutual information $(n\rho|\ln\rho|)^{-1}I(\boldsymbol{X};\boldsymbol{W})$ and minimum mean-square error $(n\rho)^{-2}\mathrm{MMSE}((X_iX_j)_{i<j}|\boldsymbol{W})$, converge: see theorem 1 and corollary 1. In the sparse limit $\rho\to0$, the MMSE curves approach a $0$–$1$ phase transition with the discontinuity at $\lambda=\lambda_c(\rho)$. This corresponds to an angular point for the mutual information (by the I-MMSE relation).

**Corollary 1** (Minimum mean-square error for the sparse spiked Wigner model). *Let* $\frac{1}{2}m_n(\lambda,\rho_n)\equiv\rho_n^{-2}\frac{d}{d\lambda}\inf_{q\in[0,\rho_n]}i_n^{\mathrm{pot}}(q,\lambda,\rho_n)$. *Let* $\epsilon>0$ *and sequences* $\lambda_n$ *and* $\rho_n$ *verifying* (2) *with* $\beta\in[0,1/13)$. *There exists* $C'>0$ *independent of* $n$ *such that*

$$m_n(\lambda_n+\epsilon,\rho_n)-\frac{C'}{\epsilon}\frac{(\ln n)^{4/3}}{n^{(1-13\beta)/7}}\leq\frac{\mathrm{MMSE}((X_iX_j)_{i<j}|\boldsymbol{W})}{(n\rho_n)^2}\leq m_n(\lambda_n-\epsilon,\rho_n)+\frac{C'}{\epsilon}\frac{(\ln n)^{4/3}}{n^{(1-13\beta)/7}}.$$

Concretely the derivative $(d/d\lambda_n)\inf_{q\in[0,\rho_n]}i_n^{\mathrm{pot}}(q,\lambda_n,\rho_n)$ is computed, using the envelope theorem [65], as $(\partial/\partial\lambda_n)i_n^{\mathrm{pot}}(q_n^*,\lambda_n,\rho)$ where $q_n^*=q_n^*(\lambda_n,\rho_n)$ is the solution of the variational problem, which is unique almost everywhere (except at the phase transition point, see e.g. [15] for such proofs). For Bernoulli and Bernoulli-Rademacher distributions, we easily compute the limiting behavior $m_n(\lambda_n,\rho_n)$ from the solution of the variational problem stated above, and find that $(n\rho_n)^{-2}\mathrm{MMSE}((X_iX_j)_{i<j}|\boldsymbol{W})$ tends to $\mathbb{I}(\gamma\leq1)$ as $n\to+\infty$.

Figure 1 shows the mutual information and MMSE computed from the numerical solution of the variational problem for a sequence of $\mathrm{Ber}(\rho_n)$ distributions. We check that the limiting curves are indeed approached as $\rho_n\to0$ and, in particular, the suitably rescaled MMSE displays the all-or-nothing transition at $\lambda/\lambda_c(\rho_n)=1$ as $n\to+\infty$ with $\lambda_c(\rho_n)=4|\ln\rho_n|/\rho_n$. For the Bernoulli-Rademacher distribution the transition location is the same, suggesting that the hardness of the inference is only related, for discrete priors, to the recovery of the support. For more generic distributions than these two cases the situation is richer. Although one generically observes phase transitions *in the same scaling regime*, the limiting curves appear to be more complicated than the simple staircase shape and the jumps are not necessarily located at $\gamma=1$. A classification of these transitions is an interesting problem that is out of the scope of this paper.

## 3 AMP algorithmic phase transition

Approximate message passing (AMP) is a low complexity algorithm that iteratively updates estimates of the unknown signal, which, in the case of the spiked Wigner model is $\boldsymbol{X}$, from the noisy data $\boldsymbol{W}$. The iterative estimates are denoted $\{\boldsymbol{x}^t\}_{t\geq1}$. Let $\boldsymbol{A}\equiv\boldsymbol{W}/\sqrt{n}$ and initialize with $f_0(\boldsymbol{x}^0)$ independent of $\boldsymbol{W}$, such that $\langle f_0(\boldsymbol{x}^0),\boldsymbol{X}\rangle>0$. Then let $\boldsymbol{x}^1=\boldsymbol{A}f_0(\boldsymbol{x}^0)$, and for $t\geq1$, compute

$$\boldsymbol{x}^{t+1}=\boldsymbol{A}f_t(\boldsymbol{x}^t)-\mathsf{b}_tf_{t-1}(\boldsymbol{x}^{t-1}),\qquad\mathsf{b}_t=\frac{1}{n}\sum_{i=1}^nf_t'(x_i^t),\tag{6}$$

where the scalar function $f_t:\mathbb{R}\to\mathbb{R}$ is applied elementwise to vector input, i.e., $f_t(\boldsymbol{x})=(f_t(x_1),\ldots,f_t(x_n))$ for a vector $\boldsymbol{x}\in\mathbb{R}^n$, and its exact value is given in what follows (in (9)). We

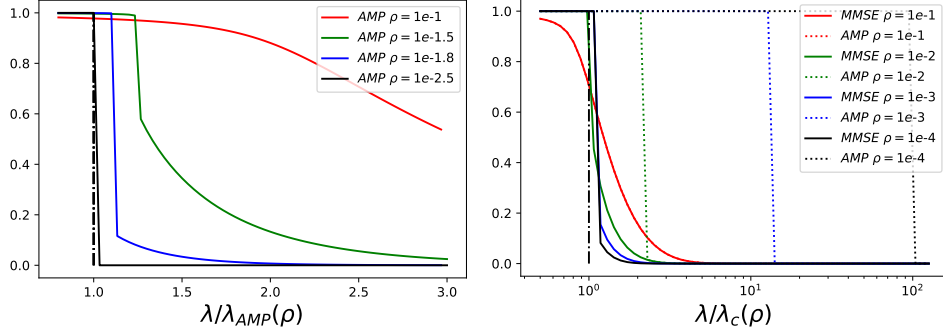

Figure 2: Left: The mean-square error towards which the suitably normalized matrix-MSE of the AMP algorithm, $\mathrm{MSE}_{\mathrm{AMP}}$, converges for various sparsity levels, see theorem 2. An all-or-nothing transition appears as $\rho = \rho_n \to 0$ at $\lambda_{\mathrm{AMP}}(\rho) = 1/(e\rho)^2$. Comparing to figure 1 the transition becomes sharper much faster as $\rho$ decreases. Right: Horizontal axis is on a log scale. The statistical-to-algorithmic gap diverges as $\rho \to 0$.

refer to the functions $\{f_t\}_{t\geq 0}$ as "denoisers", for reasons that will become clear momentarily. Notice that (6) gives both matrix estimates $\hat{\boldsymbol{X}}\hat{\boldsymbol{X}}^{\mathsf{T}} = f_t(\boldsymbol{x}^t)[f_t(\boldsymbol{x}^t)]^{\mathsf{T}}$ and signal estimates $\hat{\boldsymbol{X}} = f_t(\boldsymbol{x}^t)$.

A key property of AMP is that, asymptotically as $n \to \infty$, a deterministic, scalar recursion referred to as *state evolution* exactly characterizes its performance, in the sense that the estimates $x_i^t$ converge to random variables with mean and variance governed by the state evolution. For the sub-linear sparsity regime, we introduce an $n$-dependent state evolution, reflecting that our sparsity level $\rho_n$ and signal strength $\lambda_n$ both now change as $n$ grows. We will show, based on measure concentration arguments, that the usual asymptotic characterization also gives a finite sample approximation, meaning that for any $n$ fixed but large, $x_i^t$ is *approximately* distributed as a $x_i^t \overset{d}{\approx} \mu_t^n X_0^n + \sqrt{\tau_t^n}Z$ where $\mu_t^n$ and $\tau_t^n$ are characterized by the state evolution below with $X_0^n \sim P_{X,n}$ independent of standard gaussian $Z$. The $n$-dependent state evolution is defined as follows: for $t \geq 1$,

$$\mu_1^n = \sqrt{\lambda_n}\langle f_0(\boldsymbol{x}^0), \boldsymbol{X}\rangle/n, \qquad \tau_1^n = \|f_0(\boldsymbol{x}^0)\|^2/n, \qquad (7)$$

$$\mu_{t+1}^n = \sqrt{\lambda_n}\,\mathbb{E}\left\{X_0^n f_t\left(\mu_t^n X_0^n + \sqrt{\tau_t^n}Z\right)\right\}, \qquad \tau_{t+1}^n = \mathbb{E}\left\{\left[f_t\left(\mu_t^n X_0^n + \sqrt{\tau_t^n}Z\right)\right]^2\right\}, \quad (8)$$

where we include the $n$ superscript to emphasize the dependence.

A well-motivated choice of denoiser functions $\{f_t\}_{t\geq 0}$ are the conditional expectation denoisers. Namely, given that we have knowledge of the prior distribution of the signal elements, and considering the approximate characterization of the estimate $x_i^t$ via the state evolution, the Bayes-optimal way to update our signal estimate at any iteration is the following: for $t \geq 1$,

$$f_t(x) = \mathbb{E}\left\{X_0^n \mid \mu_t^n X_0^n + \sqrt{\tau_t^n}Z = x\right\}, \qquad (9)$$

with $X_0^n \sim P_{X,n}$ independent of standard gaussian $Z$. Strictly speaking, $f_t(\cdot)$ also has an $n$-dependency, so to be consistent we should label $f_t(\cdot) \equiv f_t^n(\cdot)$, however we drop this for simplicity. With this choice of denoiser function, the state evolution (8) simplifies: by the Law of Total Expectation, $\mathbb{E}\{X_0^n f_t(\mu_t^n X_0^n + \sqrt{\tau_t^n}Z)\} = \mathbb{E}\{[f_t(\mu_t^n X_0^n + \sqrt{\tau_t^n}Z)]^2\}$, thus $\mu_t^n = \sqrt{\lambda_n}\tau_t^n$, and so

$$\tau_{t+1}^n = \mathbb{E}\left\{\left[\mathbb{E}\left\{X_0^n \mid \sqrt{\lambda_n}\tau_t^n X_0^n + \sqrt{\tau_t^n}Z\right\}\right]^2\right\}. \qquad (10)$$

The performance guarantees given by the state evolution are stated informally in what follows, with a more formal result given in appendix K. The proof extends and refines[3] the finite sample analysis of AMP given in [63, Theorem 1]. These guarantees concern the convergence of the empirical

distribution of $x_i^t$ to its approximating distribution determined by the state evolution and specifically apply to the AMP algorithm using the denoiser in (9). For all order 2 pseudo-Lipschitz functions[4], denoted $\psi : \mathbb{R}^2 \to \mathbb{R}$ with Lipschitz constant $L_\psi > 0$, we have that for $\epsilon \in (0, 1)$ and $t \geq 1$,

$$\mathbb{P}\Big(\Big|\frac{1}{n}\sum_{i=1}^{n}\psi(X_i, f_t(x_i^t)) - \mathbb{E}\Big\{\psi\big(X_0^n, f_t(\mu_t^n X_0^n + \sqrt{\tau_t^n}Z)\big)\Big\}\Big| \geq \epsilon\Big) \leq CC_t \exp\Big\{\frac{-cc_t n\epsilon^2}{L_\psi^2 \gamma_n^t}\Big\} \quad (11)$$

where $\boldsymbol{X} = (X_1, \ldots, X_n)$ is the true signal and $C, C_t, c, c_t$ are universal constants not depending on $n$ or $\epsilon$, but with $C_t, c_t$ depending on the iteration $t$ and whose exact value is given in theorem 2. Finally, $\gamma_n^t$ characterizes the way the bound depends on the state evolution parameters and its exact value is given in (14). We want to consider, specifically, the vector-MSE and matrix-MSE of AMP, namely $\frac{1}{n}\|\boldsymbol{X} - f_t(\boldsymbol{x}^t)\|^2$ and $\frac{1}{n^2}\|\boldsymbol{X}\boldsymbol{X}^\intercal - f_t(\boldsymbol{x}^t)[f_t(\boldsymbol{x}^t)]^\intercal\|_F^2$, for any $t \geq 1$.

**Theorem 2** (Finite sample state evolution). *Consider AMP in* (6) *using the conditional expectation denoiser in* (9). *Then for* $\epsilon \in (0, 1)$ *and* $t \geq 1$, *let* $bound_t \equiv CC_t \exp\{-cc_t n\epsilon^2/\gamma_n^t\}$, *then*

$$\mathbb{P}\Big(\Big|\frac{1}{n}\|\boldsymbol{X} - f_t(\boldsymbol{x}^t)\|^2 - (\rho_n - \tau_{t+1}^n)\Big| \geq \epsilon\Big) \leq bound_t, \quad (12)$$

$$\mathbb{P}\Big(\Big|\frac{1}{n^2}\|\boldsymbol{X}\boldsymbol{X}^\intercal - f_t(\boldsymbol{x}^t)[f_t(\boldsymbol{x}^t)]^\intercal\|_F^2 - (\rho_n^2 - (\tau_{t+1}^n)^2)\Big| \geq \epsilon\Big) \leq bound_t, \quad (13)$$

*where* $X_0^n \sim P_{X,n}$ *and* $\tau_t^n$ *is defined in* (10). *The values* $C, c$ *are universal constants not depending on* $n$ *or* $\epsilon$ *with* $C_t, c_t$ *given by* $C_t = C_1^t(t!)^{C_2}, c_t = [c_1^t(t!)^{c_2}]^{-1}$. *Finally,*

$$\gamma_n^t \equiv \lambda_n^{2t-1}(\nu^n + \tau_1^n)(\nu^n + \tau_1^n + \tau_2^n)\cdots\Big(\nu^n + \sum_{i=1}^{t}\tau_i^n\Big) \quad (14)$$
$$\times \max\{1, \hat{b}_1\}\max\{1, \hat{b}_2\}\cdots\max\{1, \hat{b}_{t-1}\},$$

*where* $\nu^n$ *is the variance factor of sub-Gaussian* $\boldsymbol{X}^n$ *(for* $P_{X,n} = \mathrm{Ber}(\rho_n)$ *we have* $\nu^n \leq 1/4$; *see lemma 14 in the supplementary material and* $\hat{b}_t = \mathbb{E}\{f_t'(\mu_t^n X_0^n + \sqrt{\tau_t^n}Z)\}$.

Theorem 2 follows from the finite sample guarantees given in (11), and, in appendix K, we discuss in more detail the proof of theorem 2 and result 11. We make a few remarks on the result here.

**Remark 1:** $\rho_n$ **normalization and all-or-nothing transition.** To be consistent with the previously stated results, we could renormalize the MSEs as follows and the result still holds as

$$\mathbb{P}\Big(\Big|\frac{1}{\rho_n n}\|\boldsymbol{X} - f_t(\boldsymbol{x}^t)\|^2 - \Big(1 - \frac{\tau_{t+1}^n}{\rho_n}\Big)\Big| \geq \epsilon\Big) \leq CC_t \exp\{-cc_t n\rho_n^2\epsilon^2/\gamma_n^t\},$$

$$\mathbb{P}\Big(\Big|\frac{1}{(\rho_n n)^2}\|\boldsymbol{X}\boldsymbol{X}^\intercal - f_t(\boldsymbol{x}^t)[f_t(\boldsymbol{x}^t)]^\intercal\|_F^2 - \Big(1 - \Big(\frac{\tau_{t+1}^n}{\rho_n}\Big)^2\Big)\Big| \geq \epsilon\Big) \leq CC_t \exp\{-cc_t n\rho_n^4\epsilon^2/\gamma_n^t\}.$$

In appendix G we show that $\tau_{t+1}^n/\rho_n \to 0$ for $\lambda_n\rho_n^2 \to 0$ and $\tau_{t+1}^n/\rho_n \to 1$ for $\lambda_n\rho_n^2 \to +\infty$. This is consistent with the numerics on figure 2 where we see a transition for $\lambda_n\rho_n^2 = 1/e^2$.

**Remark 2: AMP regime and statistical-to-algorithmic gap.** We apply theorem 2 in the regime where $t = o(\frac{\ln n}{\ln \ln n})$, which, as discussed in [63], is the regime where the state evolution predictions are meaningful with respect to the values of $C_t, c_t$ and the constraints they specify on how large $t$ can be compared to the dimension $n$. In our work, we also have constraints related to the $\gamma_n^t$ value in (14) that appears in the denominator of the rate of concentration. Considering these constraints, we apply theorem 2 for signal strength and sparsity scaling like $\lambda_n\rho_n^2 = w$ and $\rho_n = \Omega((\ln n)^{-\alpha})$ with $w, \alpha \in \mathbb{R}_+$, and show that the above probabilities indeed tend to zero as $n \to +\infty$. Appendix L provides the details of this calculation.

Note that since theorem 1 and corollary 1 hold for $\rho_n = \Omega(n^{-\beta})$ and thus for $\rho_n = \Omega((\ln n)^{-\alpha})$ as well, then both the statistical and algorithmic transitions (and therefore the statistical-to-computational gap) are proven for $\rho_n = \Omega((\ln n)^{-\alpha})$.

**Remark 3:** $\lambda_n, \tau^n$ **dependence.** The $\lambda_n$ dependence in $\gamma_n^t$ defined in (14) comes from the (pseudo-)Lipschitz constants $L_f$ in (11). The dependence on the Lipschitz constants, and on the state evolution

parameters $\tau_t^n$, was not stated explicitly in the original concentration bound in [63, Theorem 1] as the authors assume these values do not change with $n$ and, thus, can be absorbed into the universal constants. By examining the proof of [63, Theorem 1], one gets that the dependence takes the form in (14). More details on how we arrive at the rates in theorem 2 can be found in appendix K.

**Remark 4: Algorithm initialization.** We assume that the AMP algorithm in (6) was initialized with $f_0(\boldsymbol{x}^0)$ independent of $\boldsymbol{W}$ such that $\langle f_0(\boldsymbol{x}^0), \boldsymbol{X} \rangle > 0$. The second condition ensures that $\mu_0^n \neq 0$ (which would mean $\mu_t^n = 0$ for all $t \geq 0$). If $P_{X,n}$ is $\mathrm{Ber}(\rho_n)$, one could use, for example, $f_0(\boldsymbol{x}^0) = \mathbf{1}$, since the mean of the signal elements is positive. However, if $P_{X,n}$ is Bernoulli-Rademacher, a more complicated initialization procedure is needed since initializing in this way would cause the algorithm to get stuck in an unstable fixed point. We refer the reader to [34] for a discussion of an appropriate spectral initialization for this setting. However, such an initialization violates the assumption of independence with $\boldsymbol{W}$. The theoretical idea in [34] that allows one to get around this dependence is to analyze AMP in (6) with a matrix $\widetilde{\boldsymbol{A}}$ that is an *approximate* representation of the conditional distribution of $\boldsymbol{A}$ given the initialization, and then to show that with high probability the two algorithms will be close each other. We believe that incorporating these ideas with the finite sample guarantee in (11) would be straightforward, and theorem 2 could be extended to the setting of AMP with a spectral initialization.

# Broader impact

One cannot underestimate the relevance of sparse estimation in modern technology, and although this work is valid within the limits of a theoretical model, it participates towards better fundamental understanding of necessary resources in terms of energy and quantity of data when this data is sparse. Besides radical transitions in behaviour under small changes of control parameters, we also show that an estimation task can become computationally hard or impossible, even with (practically) unbounded signal strengths. Broadly speaking, such results provide guidelines for better design and less wasteful engineering systems.

# Acknowledgments

J.B. acknowledges discussions with Galen Reeves during his visit of Duke University. C.R. acknowledges support from NSF CCF #1849883 and N.M. from Swiss National Foundation for Science grant number 200021E 17554.

## Footnotes

[1]This is the generic singularity and one speaks of a first order transition. In special cases the MMSE may be continuous with a higher discontinuous derivative of the mutual information.

[2]In more detail, this is equal to the signal-to-noise ratio per observation $(\lambda_n/n)\rho_n^2$ times the number of observations $\Theta(n^2)$ divided by the expected number of non-zero components $\rho_n n$.

[3]The result in [63] is a general AMP algorithm with a "rectangular" structure that does not cover the "symmetric" AMP in (6). However, extensions of this result to the symmetric case are straightforward, as discussed in [63, Section 1], but technical. Moreover, the dependence on $n$ for the state evolution requires that these values are tracked carefully through the proof, whereas this was not done in [63], as these values were assumed to be universal constants. For simplicity of exposition in this document, we do not elaborate further on these technicalities at this point and put these details in appendix K.

[4]For any $n, m \in \mathbb{N}_{>0}$, a function $\phi : \mathbb{R}^n \to \mathbb{R}^m$ is *pseudo-Lipschitz of order* 2 if there exists a constant $L > 0$ such that $\|\phi(\boldsymbol{x}) - \phi(\boldsymbol{y})\| \leq L\left(1 + \|\boldsymbol{x}\| + \|\boldsymbol{y}\|\right)\|\boldsymbol{x} - \boldsymbol{y}\|$ for $\boldsymbol{x}, \boldsymbol{y} \in \mathbb{R}^n$.

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
