[Supplementary Material]

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

# A  General results on the mutual information

In this appendix we give a more general form of theorem 1 in section 2. Our analysis by the adaptive interpolation method works for any regime where the sequences $\lambda_n$ and $\rho_n$ verify:

$$C \leq \lambda_n \rho_n = O(n^\gamma) \quad \text{for some constants} \quad \gamma \in [0, 1/2) \quad \text{and} \quad C > 0\,. \tag{15}$$

Of course this contains the regime (2) as a special case. Our general result is a statement on the smallness of

$$\Delta I_n \equiv \frac{1}{\rho_n |\ln \rho_n|} \left| \frac{1}{n} I(\boldsymbol{X}; \boldsymbol{W}) - \inf_{q \in [0, \rho_n]} i_n^{\text{pot}}(q, \lambda_n, \rho_n) \right|\,.$$

The analysis of section B leads to the following general theorem.

**Theorem 3** (Sparse spiked Wigner model). *Let the sequences $\lambda_n$ and $\rho_n$ verify (15) and let $\alpha > 0$. There exists a constant $C > 0$ independent of $n$, such that the mutual information for the Wigner spike model verifies*

$$\Delta I_n \leq \frac{C}{|\ln \rho_n|} \max \left\{ \frac{1}{n^\alpha},\, \frac{\lambda_n}{n \rho_n},\, \left( \frac{\lambda_n^4}{n^{1-4\alpha} \rho_n^2} \left(1 + \lambda_n \rho_n^2\right) \right)^{1/3} \right\}\,.$$

In particular, choosing $\lambda_n = \Theta(|\ln \rho_n|/\rho_n)$ (which is the appropriate scaling to observe a phase transition),

$$\Delta I_n \leq C \max \left\{ \frac{1}{n^\alpha |\ln \rho_n|}, \frac{1}{n\rho_n^2}, \left( \frac{|\ln \rho_n|}{n^{1-4\alpha}\rho_n^6} \right)^{1/3} \right\} .$$

If, in addition, we set $\rho_n = \Omega(n^{-\beta})$ for $\beta \geq 0$ (which is the regime in (2)), then we have

$$\Delta I_n \leq C \max \left\{ \frac{1}{n^\alpha \ln n}, \frac{1}{n^{1-2\beta}}, \left( \frac{\ln n}{n^{1-4\alpha-6\beta}} \right)^{1/3} \right\} .$$

This bound vanishes as $n$ grows if $\beta \in [0, 1/6)$ and $\alpha \in (0, (1-6\beta)/4]$. The final bound is optimized (up to polylog factors) by setting $\alpha = (1-6\beta)/7$. In this case (again, when $\lambda_n = \Theta(|\ln \rho_n|/\rho_n)$ and $\rho_n = \Omega(n^{-\beta})$),

$$\Delta I_n \leq C \frac{(\ln n)^{1/3}}{n^{(1-6\beta)/7}} .$$

# B  Information theoretic analysis by the adaptive interpolation method

In this section we provide the essential architecture for the proof of theorem 1 which relies on the adaptive interpolation method [19, 20]. The proof requires concentration properties for "free energies" and "overlaps" which are deferred to appendices C and D. When no confusion is possible we use the notation $\mathbb{E}\|A\|^2 = \mathbb{E}[\|A\|^2]$.

## B.1   The interpolating model.

Let $\epsilon \in [s_n, 2s_n]$, for a sequence $s_n$ tending to zero as $s_n = n^{-\alpha}/2 \in (0, 1/2)$, for $\alpha > 0$ chosen later. Let $q_n : [0, 1] \times [s_n, 2s_n] \mapsto [0, \rho_n]$ and set

$$R_n(t, \epsilon) \equiv \epsilon + \lambda_n \int_0^t ds\, q_n(s, \epsilon) .$$

Consider the following interpolating estimation model, where $t \in [0, 1]$, with accessible data $(W_{ij}(t))_{i,j}$ and $(\tilde{W}_i(t, \epsilon))_i$ obtained through

$$\begin{cases} W_{ij}(t) = W_{ji}(t) & = \sqrt{(1-t)\frac{\lambda_n}{n}}\, X_i X_j + Z_{ij}, \quad 1 \leq i < j \leq n, \\ \tilde{W}(t, \epsilon) & = \sqrt{R_n(t, \epsilon)}\, X + \tilde{Z}, \end{cases}$$

with standard gaussian noise $\tilde{Z} \sim \mathcal{N}(0, \mathrm{I}_n)$, and $Z_{ij} = Z_{ji} \sim \mathcal{N}(0, 1)$. The posterior associated with this model reads (here $\| - \|$ is the $\ell_2$ norm)

$$dP_{n,t,\epsilon}(\boldsymbol{x}|\boldsymbol{W}(t), \tilde{\boldsymbol{W}}(t, \epsilon)) = \frac{1}{\mathcal{Z}_{n,t,\epsilon}(\boldsymbol{W}(t), \tilde{\boldsymbol{W}}(t, \epsilon))} \left( \prod_{i=1}^n dP_{X,n}(x_i) \right)$$

$$\times \exp \left\{ \sum_{i<j}^n \left( (1-t)\frac{\lambda_n}{n}\frac{x_i^2 x_j^2}{2} - \sqrt{(1-t)\frac{\lambda_n}{n}}\, x_i x_j W_{ij}(t) \right) + R_n(t, \epsilon)\frac{\|\boldsymbol{x}\|^2}{2} \right.$$

$$\left. - \sqrt{R_n(t, \epsilon)}\boldsymbol{x} \cdot \tilde{\boldsymbol{W}}(t, \epsilon) \right\}.$$

The normalization factor $\mathcal{Z}_{n,t,\epsilon}(\dots)$ is also called partition function. We also define the mutual information density for the interpolating model

$$i_n(t, \epsilon) \equiv \frac{1}{n} I\big( \boldsymbol{X}; (\boldsymbol{W}(t), \tilde{\boldsymbol{W}}(t, \epsilon)) \big) . \tag{16}$$

The $(n, t, \epsilon, R_n)$-dependent Gibbs-bracket (that we simply denote $\langle - \rangle_t$ for the sake of readability) is defined for functions $A(\boldsymbol{x}) = A$

$$\langle A(\boldsymbol{x}) \rangle_t = \int dP_{n,t,\epsilon}(\boldsymbol{x}|\boldsymbol{W}(t), \tilde{\boldsymbol{W}}(t, \epsilon))\, A(\boldsymbol{x}) . \tag{17}$$

**Lemma 1** (Boundary values). *The mutual information for the interpolating model verifies*

$$\begin{cases} i_n(0,\epsilon) = \frac{1}{n}I(\boldsymbol{X};\boldsymbol{W}) + O(\rho_n s_n)\,, \\ i_n(1,\epsilon) = I_n(X; \{\lambda_n \int_0^1 dt\, q_n(t,\epsilon)\}^{1/2} X + Z) + O(\rho_n s_n)\,. \end{cases} \quad (18)$$

*where $I_n(X; \{\lambda_n \int_0^1 dt\, q_n(t,\epsilon)\}^{1/2} X + Z)$ is the mutual information for a scalar gaussian channel with input $X \sim P_{X,n}$ and noise $Z \sim \mathcal{N}(0,1)$.*

*Proof.* We start with the chain rule for mutual information:

$$i_n(0,\epsilon) = \frac{1}{n}I(\boldsymbol{X};\boldsymbol{W}(0)) + \frac{1}{n}I(\boldsymbol{X};\tilde{\boldsymbol{W}}(0,\epsilon)|\boldsymbol{W}(0)).$$

Note that, by the definition of $\boldsymbol{W}(t)$,

$$I(\boldsymbol{X};\boldsymbol{W}(0)) = I(\boldsymbol{X};\boldsymbol{W})\,.$$

Moreover we claim $\frac{1}{n}I(\boldsymbol{X};\tilde{\boldsymbol{W}}(0,\epsilon)|\boldsymbol{W}(0)) = O(\rho_n s_n)$, which yields the first identity in (18). This claim simply follows from the I-MMSE relation (appendix I) and $R_n(0,\epsilon) = \epsilon$:

$$\frac{d}{d\epsilon}\frac{1}{n}I(\boldsymbol{X};\tilde{\boldsymbol{W}}(0,\epsilon)|\boldsymbol{W}(0)) = \frac{1}{2n}\text{MMSE}(\boldsymbol{X}|\tilde{\boldsymbol{W}}(0,\epsilon),\boldsymbol{W}(0)) \le \frac{\rho_n}{2}\,. \quad (19)$$

The last inequality above is true because $\text{MMSE}(\boldsymbol{X}|\tilde{\boldsymbol{W}}(0,\epsilon),\boldsymbol{W}(0)) \le \mathbb{E}\|\boldsymbol{X} - \mathbb{E}\,\boldsymbol{X}\|^2 = n\text{Var}(X_1) \le n\rho_n$, as the components of $\boldsymbol{X}$ are i.i.d. from $P_{X,n}$. Therefore $\frac{1}{n}I(\boldsymbol{X};\tilde{\boldsymbol{W}}(0,\epsilon)|\boldsymbol{W}(0))$ is $\frac{\rho_n}{2}$-Lipschitz in $\epsilon \in [s_n, 2s_n]$. Moreover, we have that $I(\boldsymbol{X};\tilde{\boldsymbol{W}}(0,0)|\boldsymbol{W}(0)) = 0$. This implies the claim.

The proof of the second identity in (18) again starts from the chain rule for mutual information

$$i_n(1,\epsilon) = \frac{1}{n}I(\boldsymbol{X};\tilde{\boldsymbol{W}}(1,\epsilon)) + \frac{1}{n}I(\boldsymbol{X};\boldsymbol{W}(1)|\tilde{\boldsymbol{W}}(1,\epsilon))\,.$$

Note that $I(\boldsymbol{X};\boldsymbol{W}(1)|\tilde{\boldsymbol{W}}(1,\epsilon)) = 0$ as $\boldsymbol{W}(1)$ does not depend on $\boldsymbol{X}$. Moreover,

$$\frac{1}{n}I(\boldsymbol{X};\tilde{\boldsymbol{W}}(1,\epsilon)) = I_n(X; \sqrt{R_n(1,\epsilon)}X + Z)$$
$$= I_n(X; \{\lambda_n \int_0^1 dt\, q_n(t,\epsilon)\}^{1/2} X + Z) + O(\rho_n s_n)\,.$$

because $I_n(X; \sqrt{\gamma}X + Z)$ is a $\frac{\rho_n}{2}$-Lipschitz function of $\gamma$, by an application of the I-MMSE relation (appendix I) $\frac{d}{d\gamma}I_n(X; \sqrt{\gamma}X + Z) = \text{MMSE}(X|\sqrt{\gamma}X + Z)/2 \le \text{Var}(X)/2 \le \rho_n/2$. $\qquad \square$

## B.2 Fundamental sum rule.

**Proposition 1** (Sum rule). *The mutual information verifies the following sum rule:*

$$\frac{1}{n}I(\boldsymbol{X};\boldsymbol{W}) = i_n^{\text{pot}}\big(\int_0^1 dt\, q_n(t,\epsilon); \lambda_n, \rho_n\big) + \frac{\lambda_n}{4}\big(\mathcal{R}_1 - \mathcal{R}_2 - \mathcal{R}_3\big) + O(\rho_n s_n) + O\big(\frac{\lambda_n}{n}\big) \quad (20)$$

*with non-negative "remainders" that depend on $(n, \epsilon, R_n)$,*

$$\begin{cases} \mathcal{R}_1 \equiv \int_0^1 dt\, \big(q_n(t,\epsilon) - \int_0^1 ds\, q_n(s,\epsilon)\big)^2\,, \\ \mathcal{R}_2 \equiv \int_0^1 dt\, \mathbb{E}\big\langle\big(Q - \mathbb{E}\langle Q\rangle_t\big)^2\big\rangle_t, \\ \mathcal{R}_3 \equiv \int_0^1 dt\, \big(q_n(t,\epsilon) - \mathbb{E}\langle Q\rangle_t\big)^2\,, \end{cases} \quad (21)$$

*where $Q = \frac{1}{n}\boldsymbol{x} \cdot \boldsymbol{X}$ is called the overlap. The constants in the $O(\cdots)$ terms are independent of $n, t, \epsilon$.*

*Proof.* By the fundamental theorem of calculus $i_n(0,\epsilon) = i_n(1,\epsilon) - \int_0^1 dt\, \frac{d}{dt}i_n(t,\epsilon)$. Note that $i_n(0,\epsilon)$ and $i_n(1,\epsilon)$ are given by (18). The $t$-derivative of the interpolating mutual information is simply computed combining the I-MMSE relation with the chain rule for derivatives

$$\frac{d}{dt}i_n(t,\epsilon) = -\frac{\lambda_n}{2}\frac{1}{n^2}\sum_{i<j}\mathbb{E}\big[(X_iX_j - \langle x_ix_j\rangle_t)^2\big] + \frac{\lambda_n q_n(t,\epsilon)}{2}\frac{1}{n}\mathbb{E}\|\boldsymbol{X} - \langle\boldsymbol{x}\rangle_t\|^2 \quad (22)$$

$$= -\frac{\lambda_n}{4}\frac{1}{n^2}\mathbb{E}\|\boldsymbol{X} \otimes \boldsymbol{X} - \langle\boldsymbol{x} \otimes \boldsymbol{x}\rangle_t\|_{\text{F}}^2 + \frac{\lambda_n q_n(t,\epsilon)}{2}\frac{1}{n}\mathbb{E}\|\boldsymbol{X} - \langle\boldsymbol{x}\rangle_t\|^2 + O\big(\frac{\lambda_n}{n}\big)\,. \quad (23)$$

The correction term in (23) comes from completing the diagonal terms in the sum $\sum_{i<j}$ in order to construct the matrix-MMSE for $\boldsymbol{X} \otimes \boldsymbol{X}$, namely the first term on the r.h.s. of (23). This expression can be simplified by application of the Nishimori identities (appendix H contains a proof of these general identities). Starting with the second term (a vector-MMSE)

$$
\frac{1}{n}\mathbb{E}\|\boldsymbol{X} - \langle \boldsymbol{x} \rangle_t\|^2 = \mathbb{E}\big[\|\boldsymbol{X}\|^2 + \|\langle \boldsymbol{x} \rangle_t\|^2 - 2\boldsymbol{X} \cdot \langle \boldsymbol{x} \rangle_t\big]
$$

$$
= \frac{1}{n}\mathbb{E}\big[\|\boldsymbol{X}\|^2 - \boldsymbol{X} \cdot \langle \boldsymbol{x} \rangle_t\big] = \rho_n - \mathbb{E}\langle Q \rangle_t \,, \tag{24}
$$

were we used $\mathbb{E}\|\boldsymbol{X}\|^2 = n\rho_n$ and the Nishimori identity $\mathbb{E}\|\langle \boldsymbol{x} \rangle_t\|^2 = \mathbb{E}[\boldsymbol{X} \cdot \langle \boldsymbol{x} \rangle_t]$. By similar manipulations we obtain for the matrix-MMSE

$$
\frac{1}{n^2}\mathrm{MMSE}(\boldsymbol{X} \otimes \boldsymbol{X}|\tilde{\boldsymbol{W}}(t,\epsilon), \boldsymbol{W}(t)) = \frac{1}{n^2}\mathbb{E}\|\boldsymbol{X} \otimes \boldsymbol{X} - \langle \boldsymbol{x} \otimes \boldsymbol{x} \rangle_t\|_{\mathrm{F}}^2 = \rho_n^2 - \mathbb{E}\langle Q^2 \rangle_t \,. \tag{25}
$$

From (18), (23), (24), (25) and the fundamental theorem of calculus we deduce

$$
\frac{1}{n}I(\boldsymbol{X};\boldsymbol{W}) = I_n\big(X; \{\lambda_n \textstyle\int_0^1 dt\, q_n(t,\epsilon)\}^{1/2}X + Z\big)
$$

$$
+ \frac{\lambda_n}{4}\int_0^1 dt\,\Big\{\rho_n^2 - \mathbb{E}\langle Q^2 \rangle_t - 2q_n(t,\epsilon)(\rho_n - \mathbb{E}\langle Q \rangle_t)\Big\} + O(\rho_n s_n) + O\Big(\frac{\lambda_n}{n}\Big).
$$

The terms on the r.h.s can be re-arranged so that the potential (4) appears, and this gives immediately the sum rule (20). $\qquad\square$

Theorem 1 follows from the upper and lower bounds proven below, and applied for $s_n = \frac{1}{2}n^{-\alpha}$.

## B.3 Upper bound: linear interpolation path.

**Proposition 2** (Upper bound). *We have*

$$
\frac{1}{n}I(\boldsymbol{X};\boldsymbol{W}) \leq \inf_{q \in [0,\rho_n]} i_n^{\mathrm{pot}}(q, \lambda_n, \rho_n) + O(\rho_n s_n) + O\Big(\frac{\lambda_n}{n}\Big).
$$

*Proof.* Fix $q_n(t,\epsilon) = q_n \in [0,\rho_n]$ a constant independent of $\epsilon, t$. The interpolation path $R_n(t,\epsilon)$ is therefore a simple linear function of time. From (21) $\mathcal{R}_1$ cancels and since $\mathcal{R}_2$ and $\mathcal{R}_3$ are non-negative we get from Proposition (1)

$$
\frac{1}{n}I(\boldsymbol{X};\boldsymbol{W}) \leq i_n^{\mathrm{pot}}(q, \lambda_n, \rho_n) + O(\rho_n s_n) + O\Big(\frac{\lambda_n}{n}\Big).
$$

Note that the error terms $O(\cdots)$ are bounded independently of $q_n$. Therefore optimizing the r.h.s over the free parameter $q_n \in [0,\rho_n]$ yields the upper bound. $\qquad\square$

## B.4 Lower bound: adaptive interpolation path.

We start with a definition: the map $\epsilon \mapsto R_n(t,\epsilon)$ is called *regular* if it is a $\mathcal{C}^1$-diffeomorphism whose jacobian is greater or equal to one for all $t \in [0,1]$.

**Proposition 3** (Lower bound). *Consider sequences $\lambda_n$ and $\rho_n$ satisfying $c_1 \leq \lambda_n \rho_n \leq c_2 n^\gamma$ for some constants positive constant $c_1, c_2$ and $\gamma \in [0, 1/2[$. Then*

$$
\frac{1}{n}I(\boldsymbol{X};\boldsymbol{W}) \geq \inf_{q \in [0,\rho_n]} i_n^{\mathrm{pot}}(q, \lambda_n, \rho_n) + O(\rho_n s_n) + O\Big(\frac{\lambda_n}{n}\Big) + O\Big(\Big(\frac{\lambda_n^4 \rho_n}{n s_n^4}\Big)^{1/3}\Big). \tag{26}
$$

*Proof.* First note that the regime (2) for the sequences $\lambda_n, \rho_n$ satisfies the more general condition assumed in this lemma (this is the condition in theorem 3 of appendix A). Assume for the moment that the map $\epsilon \mapsto R_n(t,\epsilon)$ is regular. Then, based on Proposition 7 and identity (38) (appendix

D), we have a bound on the overlap fluctuation. Namely, for some numerical constant $C \geq 0$ independent of $n$

$$\frac{\lambda_n}{s_n} \int_{s_n}^{2s_n} d\epsilon \, \mathcal{R}_2 = \frac{\lambda_n}{s_n} \int_{s_n}^{2s_n} d\epsilon \int_0^1 dt \, \mathbb{E} \big\langle (Q - \mathbb{E}\langle Q \rangle_{n,t,R_n(t,\epsilon)})^2 \big\rangle_{n,t,R_n(t,\epsilon)}$$

$$\leq C \Big( \frac{\lambda_n^4 \rho_n}{n s_n^4} \Big)^{1/3}. \tag{27}$$

Using this concentration result, and $\mathcal{R}_1 \geq 0$, and averaging the sum rule (20) over $\epsilon \in [s_n, 2s_n]$ (recall the error terms are independent of $\epsilon$) we find

$$\frac{I(\boldsymbol{X};\boldsymbol{W})}{n} \geq \frac{1}{s_n} \int_{s_n}^{2s_n} d\epsilon i_n^{\mathrm{pot}} \big( \int_0^1 dt \, q_n(t,\epsilon), \lambda_n, \rho_n \big) - \frac{\lambda_n}{4} \frac{1}{s_n} \int_{s_n}^{2s_n} d\epsilon \int_0^1 dt \, \big( q_n(t,\epsilon) - \mathbb{E}\langle Q \rangle_t \big)^2$$

$$+ O(\rho_n s_n) + O\Big(\frac{\lambda_n}{n}\Big) + O\Big( \Big( \frac{\lambda_n^4 \rho_n}{n s_n^4} \Big)^{1/3} \Big). \tag{28}$$

At this stage it is natural to see if we can choose $q_n(t,\epsilon)$ to be the solution of $q_n(t,\epsilon) = \mathbb{E}\langle Q \rangle_t$. Setting $F_n(t, R_n(t,\epsilon)) \equiv \mathbb{E}\langle Q \rangle_{n,t,R_n(t,\epsilon)}$, we recognize a *first order ordinary differential equation*

$$\frac{d}{dt} R_n(t,\epsilon) = F_n(t, R_n(t,\epsilon)) \quad \text{with initial condition} \quad R_n(0,\epsilon) = \epsilon. \tag{29}$$

As $F_n(t, R_n(t,\epsilon))$ is $\mathcal{C}^1$ with bounded derivative w.r.t. its second argument the Cauchy-Lipschitz theorem implies that (29) admits a unique global solution $R_n^*(t,\epsilon) = \epsilon + \int_0^t ds \, q_n^*(s,\epsilon)$, where $q_n^* : [0,1] \times [s_n, 2s_n] \mapsto [0, \rho_n]$. Note that any solution must satisfy $q_n^*(t,\epsilon) \in [0, \rho_n]$ because $\mathbb{E}\langle Q \rangle_{n,t,\epsilon} \in [0, \rho_n]$ as can be seen from a Nishimori identity (appendix H) and (24).

We check that $R_n^*$ is regular. By Liouville's formula the jacobian of the flow $\epsilon \mapsto R_n^*(t,\epsilon)$ satisfies

$$\frac{d}{d\epsilon} R_n^*(t,\epsilon) = \exp \Big\{ \int_0^t ds \, \frac{d}{dR} F_n(s, R) \Big|_{R=R_n^*(s,\epsilon)} \Big\}.$$

Applying repeatedly the Nishimori identity of Lemma 7 (appendix H) one obtains (this computation does not present any difficulty and can be found in section 6 of [19])

$$\frac{d}{dR} F_n(s, R) = \frac{1}{n} \sum_{i,j=1}^n \mathbb{E}\big[ (\langle x_i x_j \rangle_{n,s,R} - \langle x_i \rangle_{n,s,R} \langle x_j \rangle_{n,s,R})^2 \big] \geq 0 \tag{30}$$

so that the flow has a jacobian greater or equal to one. In particular it is locally invertible (surjective). Moreover it is injective because of the unicity of the solution of the differential equation, and therefore it is a $C^1$-diffeomorphism. Thus $\epsilon \mapsto R_n^*(t,\epsilon)$ is regular. With the choice $R_n^*$, i.e., by suitably *adapting* the interpolation path, we cancel $\mathcal{R}_3$. This yields

$$\frac{1}{n} I(\boldsymbol{X};\boldsymbol{W}) \geq \frac{1}{s_n} \int_{s_n}^{2s_n} d\epsilon \, i_n^{\mathrm{pot}} \big( \int_0^1 dt \, q_n^*(t,\epsilon), \lambda_n, \rho_n \big) + O(\cdots)$$

$$\geq \inf_{q \in [0, \rho_n]} i_n^{\mathrm{pot}}(q, \lambda_n, \rho_n) + O(\cdots)$$

where the $O(\cdots)$ is a shorthand notation for the three error terms in (28). This the desired result. $\square$

# C  Concentration of free energy

For this appendix it is convenient to use the language of statistical mechanics.

## C.1  Statistical mechanics notations

We express the posterior of the interpolating model

$$dP_{n,t,\epsilon}(\boldsymbol{x}|\boldsymbol{W}(t), \tilde{\boldsymbol{W}}(t,\epsilon)) = \frac{1}{\mathcal{Z}_{n,t,\epsilon}(\boldsymbol{W}(t), \tilde{\boldsymbol{W}}(t,\epsilon))}$$

$$\times \Big( \prod_{i=1}^n dP_{X,n}(x_i) \Big) \exp \big\{ -\mathcal{H}_{n,t,\epsilon}(\boldsymbol{x}, \boldsymbol{W}(t), \tilde{\boldsymbol{W}}(t,\epsilon)) \big\} \tag{31}$$

with normalization constant (partition function) $\mathcal{Z}_{n,t,\epsilon}$ and "hamiltonian"

$$\mathcal{H}_{n,t,\epsilon}(\boldsymbol{x}, \boldsymbol{W}(t), \tilde{\boldsymbol{W}}(t,\epsilon)) = \mathcal{H}_{n,t,\epsilon}(\boldsymbol{x}, \boldsymbol{X}, \boldsymbol{Z}, \tilde{\boldsymbol{Z}}) \tag{32}$$

$$\equiv \sum_{i<j}^{n} \left( (1-t)\frac{\lambda_n}{n}\frac{x_i^2 x_j^2}{2} - \sqrt{(1-t)\frac{\lambda_n}{n}} x_i x_j W_{ij}(t) \right) + R_n(t,\epsilon)\frac{\|\boldsymbol{x}\|^2}{2} - \sqrt{R_n(t,\epsilon)}\boldsymbol{x}\cdot\tilde{\boldsymbol{W}}(t,\epsilon)$$

$$= (1-t)\lambda_n \sum_{i<j}^{n} \left( \frac{x_i^2 x_j^2}{2n} - \frac{x_i x_j X_i X_j}{n} - \frac{x_i x_j Z_{ij}}{\sqrt{n(1-t)\lambda_n}} \right) \tag{33}$$

$$+ R_n(t,\epsilon)\left( \frac{\|\boldsymbol{x}\|^2}{2} - \boldsymbol{x}\cdot\boldsymbol{X} - \frac{\boldsymbol{x}\cdot\tilde{\boldsymbol{Z}}}{\sqrt{R_n(t,\epsilon)}} \right).$$

It will also be convenient to work with "free energies" rather than mutual informations. The free energy $F_n(t,\epsilon)$ and (its expectation $f_n(t,\epsilon)$) for the interpolating model is simply minus the (expected) log-partition function:

$$F_{n,t,\epsilon}(\boldsymbol{W}(t), \tilde{\boldsymbol{W}}(t,\epsilon)) \equiv -\frac{1}{n}\ln \mathcal{Z}_{n,t,\epsilon}(\boldsymbol{W}(t), \tilde{\boldsymbol{W}}(t,\epsilon)), \tag{34}$$

$$f_n(t,\epsilon) \equiv \mathbb{E}\, F_{n,t,\epsilon}(\boldsymbol{W}(t), \tilde{\boldsymbol{W}}(t,\epsilon)). \tag{35}$$

The expectation $\mathbb{E}$ carries over the data. The averaged free energy is related to the mutual information $i_n(t,\epsilon)$ given by (16) through

$$i_n(t,\epsilon) = f_n(t,\epsilon) + \frac{n-1}{n}\frac{\rho^2 \lambda(1-t)}{4} + \frac{\rho R_n(t,\epsilon)}{2}. \tag{36}$$

## C.2 Free energy concentration

In this section we prove a concentration identity for the free energy (34) onto its average (35).

**Proposition 4** (Free energy concentration for the spiked Wigner model). *We have*

$$\mathbb{E}\left[ \left( F_{n,t,\epsilon}(\boldsymbol{W}(t), \tilde{\boldsymbol{W}}(t,\epsilon)) - f_n(t,\epsilon) \right)^2 \right] \leq \frac{2\rho_n S^2}{n}\left( (2s_n + \lambda_n \rho_n)^2 + S^4 \right) + \frac{3}{2}\frac{\lambda_n \rho_n^2}{n} + 2\frac{s_n \rho_n}{n}.$$

*Considering sequences $\lambda_n$ and $\rho_n$ verifying (15) and with $s_n = (1/2)n^{-\alpha} \to 0_+$ the bound simplifies to $C(S)\lambda_n^2\rho_n^3/n$ with positive constant $C(S) \leq \frac{5}{2} + 8S^2 + 2S^6$.*

The proof is based on two classical concentration inequalities,

**Proposition 5** (Gaussian Poincaré inequality). *Let $\boldsymbol{U} = (U_1, \ldots, U_N)$ be a vector of $N$ independent standard normal random variables. Let $g : \mathbb{R}^N \to \mathbb{R}$ be a continuously differentiable function. Then*

$$\mathrm{Var}(g(\boldsymbol{U})) \leq \mathbb{E}\|\nabla g(\boldsymbol{U})\|^2.$$

**Proposition 6** (Efron-Stein inequality). *Let $\mathcal{U} \subset \mathbb{R}$, and a function $g : \mathcal{U}^N \to \mathbb{R}$. Let $\boldsymbol{U} = (U_1, \ldots, U_N)$ be a vector of $N$ independent random variables with law $P_U$ that take values in $\mathcal{U}$. Let $\boldsymbol{U}^{(i)}$ a vector which differs from $\boldsymbol{U}$ only by its $i$-th component, which is replaced by $U_i'$ drawn from $P_U$ independently of $\boldsymbol{U}$. Then*

$$\mathrm{Var}(g(\boldsymbol{U})) \leq \frac{1}{2}\sum_{i=1}^{N} \mathbb{E}_{\boldsymbol{U}}\mathbb{E}_{U_i'}\left[ (g(\boldsymbol{U}) - g(\boldsymbol{U}^{(i)}))^2 \right].$$

We start by proving the concentration w.r.t. the gaussian variables. It is convenient to make explicit the dependence of the partition function of the interpolating model in the independent quenched variables instead of the data: $\mathcal{Z}_{n,t,\epsilon}(\boldsymbol{X}, \boldsymbol{Z}, \tilde{\boldsymbol{Z}}) = \mathcal{Z}_{n,t,\epsilon}(\boldsymbol{W}(t), \tilde{\boldsymbol{W}}(t,\epsilon))$.

**Lemma 2** (Concentration w.r.t. the gaussian variables). *We have*

$$\mathbb{E}\left[ \left( \frac{1}{n}\ln \mathcal{Z}_{n,t,\epsilon}(\boldsymbol{X}, \boldsymbol{Z}, \tilde{\boldsymbol{Z}}) - \frac{1}{n}\mathbb{E}_{\boldsymbol{Z}, \tilde{\boldsymbol{Z}}}\ln \mathcal{Z}_{n,t,\epsilon}(\boldsymbol{X}, \boldsymbol{Z}, \tilde{\boldsymbol{Z}}) \right)^2 \right] \leq \frac{3}{2}\frac{\lambda_n \rho_n^2}{n} + 2\frac{s_n \rho_n}{n}.$$

*Proof.* Fix all variables except $\boldsymbol{Z}, \tilde{\boldsymbol{Z}}$. Let $g(\boldsymbol{Z}, \tilde{\boldsymbol{Z}}) \equiv -\frac{1}{n} \ln \mathcal{Z}_{n,t,\epsilon}(\boldsymbol{X}, \boldsymbol{Z}, \tilde{\boldsymbol{Z}})$ be the free energy seen as a function of the gaussian variables only. The free energy gradient reads $\mathbb{E}\|\nabla g\|^2 = \mathbb{E}\|\nabla_{\boldsymbol{Z}} g\|^2 + \mathbb{E}\|\nabla_{\tilde{\boldsymbol{Z}}} g\|^2$. Let us denote $\mathcal{H}(t) \equiv \mathcal{H}_{n,t,\epsilon}$ the interpolating Hamiltonian (32).

$$\mathbb{E}\|\nabla_{\boldsymbol{Z}} g\|^2 = \frac{1}{n^2} \mathbb{E}\|\langle \nabla_{\boldsymbol{Z}} \mathcal{H}(t) \rangle_t\|^2 = \frac{(1-t)\lambda_n}{n^3} \sum_{i<j} \mathbb{E}[\langle x_i x_j \rangle_t^2] \leq \frac{(1-t)\lambda_n}{n^3} \sum_{i<j} \mathbb{E}\langle (x_i x_j)^2 \rangle_t$$

$$\overset{\mathrm{N}}{=} \frac{(1-t)\lambda_n}{n^3} \sum_{i<j} \mathbb{E}[(X_i X_j)^2] \leq \frac{\lambda_n \rho_n^2}{2n}$$

where we used a Nishimori identity for the last equality. Similarly, and using $\lambda_n \rho_n \geq 1$ and $s_n < 1/2$,

$$\mathbb{E}\|\nabla_{\tilde{\boldsymbol{Z}}} g\|^2 = \frac{R(\epsilon)}{n^2} \mathbb{E}\|\langle \boldsymbol{x} \rangle_t\|^2 \leq \frac{R(\epsilon)}{n^2} \mathbb{E}\langle \|\boldsymbol{x}\|^2 \rangle_t \overset{\mathrm{N}}{=} \frac{R(\epsilon)}{n^2} \mathbb{E}\|\boldsymbol{X}\|^2 \leq \frac{(2s_n + \rho_n \lambda_n)\rho_n}{n}.$$

Therefore Proposition 5 directly implies the stated result. $\qquad \square$

We now consider the fluctuations due to the signal realization:

**Lemma 3** (Concentration w.r.t. the spike)**.** *We have*

$$\mathbb{E}\Big[\Big( -\frac{1}{n} \mathbb{E}_{\boldsymbol{Z}, \tilde{\boldsymbol{Z}}} \ln \mathcal{Z}_{n,t,\epsilon}(\boldsymbol{X}, \boldsymbol{Z}, \tilde{\boldsymbol{Z}}) - f_n(t, \epsilon) \Big)^2 \Big] \leq \frac{2\rho_n S^2}{n} \Big( (2s_n + \lambda_n \rho_n)^2 + S^4 \Big).$$

*Proof.* Let $g(\boldsymbol{X}) \equiv -\frac{1}{n} \mathbb{E}_{\boldsymbol{Z}, \tilde{\boldsymbol{Z}}} \ln \mathcal{Z}_{n,t,\epsilon}(\boldsymbol{X}, \boldsymbol{Z}, \tilde{\boldsymbol{Z}})$. Define $\boldsymbol{X}^{(i)}$ as a vector with same entries as $\boldsymbol{X}$ except the $i$-th one that is replaced by $X_i'$ drawn independently from $P_{X,n}$. Let us estimate $(g(\boldsymbol{X}) - g(\boldsymbol{X}^{(i)}))^2$ by interpolation. Let $\mathcal{H}(t, s\boldsymbol{X} + (1-s)\boldsymbol{X}^{(i)})$ be the interpolating Hamiltonian (32) with $\boldsymbol{X}$ replaced by $s\boldsymbol{X} + (1-s)\boldsymbol{X}^{(i)}$. Then

$$\mathbb{E}\big[ (g(\boldsymbol{X}) - g(\boldsymbol{X}^{(i)}))^2 \big] = \mathbb{E}\Big[\Big( \int_0^1 ds \frac{dg}{ds}(s\boldsymbol{X} + (1-s)\boldsymbol{X}^{(i)}) \Big)^2 \Big]$$

$$= \frac{1}{n^2} \mathbb{E}\Big[\Big( \int_0^1 ds \Big\langle \frac{d\mathcal{H}}{ds}(t, s\boldsymbol{X} + (1-s)\boldsymbol{X}^{(i)}) \Big\rangle_t \Big)^2 \Big]$$

$$= \frac{1}{n^2} \mathbb{E}\Big[\Big( (X_i - X_i') \Big\langle R_\epsilon(t) x_i + \frac{1-t}{n} x_i \sum_{j(\neq i)} X_j x_j \Big\rangle_t \Big)^2 \Big]$$

$$\leq \frac{2}{n^2} \mathbb{E}\Big[ (X_i - X_i')^2 \Big( \langle x_i \rangle_t^2 (2s_n + \lambda_n \rho_n)^2 + \frac{1}{n^2} \sum_{j,k(\neq i)} X_j X_k \langle x_i x_j \rangle_t \langle x_i x_k \rangle_t \Big) \Big]$$

$$\leq \frac{2}{n^2} \mathbb{E}[(X_i - X_i')^2] \Big( S^2 (2s_n + \lambda_n \rho_n)^2 + S^6 \Big)$$

$$\leq \frac{4\rho_n S^2}{n^2} \Big( (2s_n + \lambda_n \rho_n)^2 + S^4 \Big).$$

We used $(a+b)^2 \leq 2(a^2 + b^2)$ for the second inequality and $\mathbb{E}[(X_i - X_i')^2] = 2\mathrm{Var}(X_i) \leq 2\rho_n$. Therefore Proposition 6 implies the claim. $\qquad \square$

# D  Overlap concentration: proof of inequality (27)

The derivations below will apply for any $t \in [0, 1]$ so we drop all un-necessary notations and indices. Only the dependence of the free energies in $R(\epsilon) \equiv R_n(t, \epsilon)$ matters, so we denote $F(R(\epsilon)) \equiv F_{n,t,\epsilon}(\boldsymbol{W}(t), \tilde{\boldsymbol{W}}(t, \epsilon))$ and $f(R(\epsilon)) \equiv f_n(t, \epsilon)$.

Let $\mathcal{L}$ be the $R(\epsilon)$-derivative of the Hamiltonian (32) divided by $n$:

$$\mathcal{L}(\boldsymbol{x}, \boldsymbol{X}, \tilde{\boldsymbol{Z}}) = \mathcal{L} \equiv \frac{1}{n} \frac{d\mathcal{H}_{n,t,\epsilon}}{dR(\epsilon)} = \frac{1}{n} \Big( \frac{\|\boldsymbol{x}\|^2}{2} - \boldsymbol{x} \cdot \boldsymbol{X} - \frac{\boldsymbol{x} \cdot \tilde{\boldsymbol{Z}}}{2\sqrt{R(\epsilon)}} \Big). \tag{37}$$

The overlap fluctuations are upper bounded by those of $\mathcal{L}$, which are easier to control, as

$$\mathbb{E}\big\langle (Q - \mathbb{E}\langle Q \rangle_t)^2 \big\rangle_t \leq 4\, \mathbb{E}\big\langle (\mathcal{L} - \mathbb{E}\langle \mathcal{L} \rangle_t)^2 \big\rangle_t . \tag{38}$$

The bracket is again the expectation w.r.t. the posterior of the interpolating model (17). A detailed derivation of this inequality can be found in appendix E and involves only elementary algebra using the Nishimori identity and integrations by parts w.r.t. the gaussian noise $\tilde{Z}$.

We have the following identities: for any given realisation of the quenched disorder

$$\frac{dF}{dR(\epsilon)} = \langle \mathcal{L} \rangle_t , \tag{39}$$

$$\frac{1}{n} \frac{d^2 F}{dR(\epsilon)^2} = -\big\langle (\mathcal{L} - \langle \mathcal{L} \rangle_t)^2 \big\rangle_t + \frac{1}{4n^2 R(\epsilon)^{3/2}} \langle \boldsymbol{x} \rangle_t \cdot \tilde{\boldsymbol{Z}} . \tag{40}$$

The gaussian integration by part formula (59) with hamiltonian (32) yields

$$\frac{\mathbb{E}\langle \tilde{\boldsymbol{Z}} \cdot \boldsymbol{x} \rangle_t}{\sqrt{R(\epsilon)}} = \mathbb{E}\big\langle \|\boldsymbol{x}\|^2 \big\rangle_t - \mathbb{E}\|\langle \boldsymbol{x} \rangle_t\|^2 \overset{\mathrm{N}}{=} \mathbb{E}\big\langle \|\boldsymbol{x}\|^2 \big\rangle_t - \mathbb{E}\langle \boldsymbol{X} \cdot \boldsymbol{x} \rangle_t = \mathbb{E}\big\langle \|\boldsymbol{x}\|^2 \big\rangle_t - n\, \mathbb{E}\langle Q \rangle_t . \tag{41}$$

Therefore averaging (39) and (40) we find

$$\frac{df}{dR(\epsilon)} = \mathbb{E}\langle \mathcal{L} \rangle_t \overset{\mathrm{N}}{=} -\frac{1}{2}\mathbb{E}\langle Q \rangle_t , \tag{42}$$

$$\frac{1}{n} \frac{d^2 f}{dR(\epsilon)^2} = -\mathbb{E}\big\langle (\mathcal{L} - \langle \mathcal{L} \rangle_t)^2 \big\rangle_t + \frac{1}{4n^2 R(\epsilon)} \mathbb{E}\big\langle \|\boldsymbol{x} - \langle \boldsymbol{x} \rangle_t\|^2 \big\rangle_t . \tag{43}$$

We always work under the assumption that the map $\epsilon \in [s_n, 2s_n] \mapsto R(\epsilon) \in [R(s_n), R(2s_n)]$ is regular, and do not repeat this assumption in the statements below. The concentration inequality (27) is a direct consequence of the following result (combined with Fubini's theorem):

**Proposition 7** (Total fluctuations of $\mathcal{L}$). *Let the sequences $\lambda_n$ and $\rho_n$ verify (15). Then*

$$\int_{s_n}^{2s_n} d\epsilon \, \mathbb{E}\big\langle (\mathcal{L} - \mathbb{E}\langle \mathcal{L} \rangle_t)^2 \big\rangle_t \leq C \Big( \frac{\lambda_n \rho_n}{n s_n} \big(1 + \lambda_n \rho_n^2\big) \Big)^{1/3}$$

*for a constant $C > 0$ that is independent of $n$, as long as the r.h.s. is $\omega(1/n)$.*

The proof of this proposition is broken in two parts, using the decomposition

$$\mathbb{E}\big\langle (\mathcal{L} - \mathbb{E}\langle \mathcal{L} \rangle_t)^2 \big\rangle_t = \mathbb{E}\big\langle (\mathcal{L} - \langle \mathcal{L} \rangle_t)^2 \big\rangle_t + \mathbb{E}\big[ (\langle \mathcal{L} \rangle_t - \mathbb{E}\langle \mathcal{L} \rangle_t)^2 \big] .$$

Thus it suffices to prove the two following lemmas. The first lemma expresses concentration w.r.t. the posterior distribution (or "thermal fluctuations") and is a direct consequence of concavity properties of the average free energy and the Nishimori identity.

**Lemma 4** (Thermal fluctuations of $\mathcal{L}$). *We have*

$$\int_{s_n}^{2s_n} d\epsilon \, \mathbb{E}\big\langle (\mathcal{L} - \langle \mathcal{L} \rangle_t)^2 \big\rangle_t \leq \frac{\rho_n}{n} \Big(1 + \frac{\ln 2}{4}\Big) .$$

*Proof.* We emphasize again that the interpolating free energy (16) is here viewed as a function of $R(\epsilon)$. In the argument that follows we consider derivatives of this function w.r.t. $R(\epsilon)$. By (43)

$$\mathbb{E}\big\langle (\mathcal{L} - \langle \mathcal{L} \rangle_t)^2 \big\rangle_t = -\frac{1}{n} \frac{d^2 f}{dR(\epsilon)^2} + \frac{1}{4n^2 R(\epsilon)} \big( \mathbb{E}\big\langle \|\boldsymbol{x}\|^2 \big\rangle_t - \mathbb{E}\|\langle \boldsymbol{x} \rangle_t\|^2 \big)$$

$$\leq -\frac{1}{n} \frac{d^2 f}{dR(\epsilon)^2} + \frac{\rho_n}{4n\epsilon} , \tag{44}$$

where we used $R(\epsilon) \geq \epsilon$ and $\frac{1}{n}\mathbb{E}\big\langle \|\boldsymbol{x}\|^2 \big\rangle_t \overset{\mathrm{N}}{=} \mathbb{E}[X_1^2] = \rho_n$. We integrate this inequality over $\epsilon \in [s_n, 2s_n]$. Recall the map $\epsilon \mapsto R(\epsilon)$ has a Jacobian $\geq 1$, is $\mathcal{C}^1$ and has a well defined $\mathcal{C}^1$ inverse

since we have assumed that it is regular. Thus integrating (44) and performing a change of variable (to get the second inequality) we obtain

$$\int_{s_n}^{2s_n} d\epsilon\, \mathbb{E}\langle (\mathcal{L} - \langle \mathcal{L}\rangle_t)^2 \rangle_t \leq -\frac{1}{n}\int_{s_n}^{2s_n} d\epsilon\, \frac{d^2 f}{dR(\epsilon)^2} + \frac{\rho_n}{4n}\int_{s_n}^{2s_n} \frac{d\epsilon}{\epsilon}$$

$$\leq -\frac{1}{n}\int_{R(s_n)}^{R(2s_n)} dR(\epsilon)\, \frac{d^2 f}{dR(\epsilon)^2} + \frac{\rho_n}{4n}\int_{s_n}^{2s_n} \frac{d\epsilon}{\epsilon}$$

$$= \frac{1}{n}\Big(\frac{df}{dR(\epsilon)}(R(s_n)) - \frac{df}{dR(\epsilon)}(R(2s_n))\Big) + \frac{\rho_n}{4n}\ln 2\,.$$

We have $|f'(R(\epsilon))| = |\mathbb{E}\langle Q\rangle_t/2| \leq \rho_n/2$ so the first term is certainly smaller in absolute value than $\rho_n/n$. This concludes the proof of Lemma 4. $\qquad\square$

The second lemma expresses the concentration w.r.t. the quenched disorder variables and is a consequence of the concentration of the free energy onto its average (w.r.t. the quenched variables).

**Lemma 5** (Quenched fluctuations of $\mathcal{L}$). *Let the sequences $\lambda_n$ and $\rho_n$ verify (15). Then*

$$\int_{s_n}^{2s_n} d\epsilon\, \mathbb{E}\big[(\langle \mathcal{L}\rangle_t - \mathbb{E}\langle \mathcal{L}\rangle_t)^2\big] \leq C\Big(\frac{\lambda_n\rho_n}{ns_n}\big(1 + \lambda_n\rho_n^2\big)\Big)^{1/3}$$

*for a constant $C > 0$ that is independent of $n$, as long as the r.h.s. is $\omega(1/n)$.*

*Proof.* Consider the following functions of $R(\epsilon)$:

$$\tilde{F}(R(\epsilon)) \equiv F(R(\epsilon)) + S\frac{\sqrt{R(\epsilon)}}{n}\sum_{i=1}^{n}|\tilde{Z}_i|\,,$$

$$\tilde{f}(R(\epsilon)) \equiv \mathbb{E}\,\tilde{F}(R(\epsilon)) = f(R(\epsilon)) + S\sqrt{R(\epsilon)}\mathbb{E}\,|\tilde{Z}_1|\,. \tag{45}$$

Because of (40) we see that the second derivative of $\tilde{F}(R(\epsilon))$ w.r.t. $R(\epsilon)$ is negative so that it is concave. Note $F(R(\epsilon))$ itself is not necessarily concave in $R(\epsilon)$, although $f(R(\epsilon))$ is. Concavity of $f(R(\epsilon))$ is not obvious from (43) (obtained from differentiating $\mathbb{E}\langle \mathcal{L}\rangle_t$ w.r.t. $R(\epsilon)$) but can be seen from (61) (obtained instead by differentiating $-\frac{1}{2}\mathbb{E}\langle Q\rangle_t$) which reads $\frac{d}{dR(\epsilon)}\mathbb{E}\langle Q\rangle_t = -2\frac{d^2}{dR(\epsilon)^2}f \geq 0$. Evidently $\tilde{f}(R(\epsilon))$ is concave too. Concavity then allows to use the following standard lemma:

**Lemma 6** (A bound for concave functions). *Let $G(x)$ and $g(x)$ be concave functions. Let $\delta > 0$ and define $C_\delta^-(x) \equiv g'(x - \delta) - g'(x) \geq 0$ and $C_\delta^+(x) \equiv g'(x) - g'(x + \delta) \geq 0$. Then*

$$|G'(x) - g'(x)| \leq \delta^{-1}\sum_{u\in\{x-\delta,\, x,\, x+\delta\}}|G(u) - g(u)| + C_\delta^+(x) + C_\delta^-(x)\,.$$

First, from (45) we have

$$\tilde{F}(R(\epsilon)) - \tilde{f}(R(\epsilon)) = F(R(\epsilon)) - f(R(\epsilon)) + S\sqrt{R(\epsilon)}A_n \tag{46}$$

with $A_n \equiv \frac{1}{n}\sum_{i=1}^{n}|\tilde{Z}_i| - \mathbb{E}\,|\tilde{Z}_1|$. Second, from (39), (42) we obtain for the $R(\epsilon)$-derivatives

$$\tilde{F}'(R(\epsilon)) - \tilde{f}'(R(\epsilon)) = \langle \mathcal{L}\rangle_t - \mathbb{E}\langle \mathcal{L}\rangle_t + \frac{SA_n}{2\sqrt{R(\epsilon)}}\,. \tag{47}$$

From (46) and (47) it is then easy to show that Lemma 6 implies

$$|\langle \mathcal{L}\rangle_t - \mathbb{E}\langle \mathcal{L}\rangle_t| \leq \delta^{-1}\sum_{u\in\{R(\epsilon)-\delta,\, R(\epsilon),\, R(\epsilon)+\delta\}}\big(|F(u) - f(u)| + S|A_n|\sqrt{u}\big)$$

$$+ C_\delta^+(R(\epsilon)) + C_\delta^-(R(\epsilon)) + \frac{S|A_n|}{2\sqrt{\epsilon}} \tag{48}$$

where $C_\delta^-(R(\epsilon)) \equiv \tilde{f}'(R(\epsilon) - \delta) - \tilde{f}'(R(\epsilon)) \geq 0$ and $C_\delta^+(R(\epsilon)) \equiv \tilde{f}'(R(\epsilon)) - \tilde{f}'(R(\epsilon) + \delta) \geq 0$. We used $R(\epsilon) \geq \epsilon$ for the term $S|A_n|/(2\sqrt{\epsilon})$. Note that $\delta$ will be chosen later on strictly smaller

than $s_n$ so that $R(\epsilon) - \delta \geq \epsilon - \delta \geq s_n - \delta$ remains positive. Remark that by independence of the noise variables $\mathbb{E}[A_n^2] = (1 - 2/\pi)/n \leq 1/n$. We square the identity (48) and take its expectation. Then using $(\sum_{i=1}^p v_i)^2 \leq p \sum_{i=1}^p v_i^2$, and that $R(\epsilon) \leq 2s_n + \lambda_n \rho_n$, as well as the free energy concentration Proposition 4 (under the assumption that $\lambda_n$ and $\rho_n$ verify (15)),

$$\frac{1}{9}\mathbb{E}\big[(\langle \mathcal{L} \rangle_t - \mathbb{E}\langle \mathcal{L} \rangle_t)^2\big] \leq \frac{3}{n\delta^2}\Big(C\lambda_n^2 \rho_n^3 + S(2s_n + \lambda_n \rho_n + \delta)\Big)$$

$$+ C_\delta^+(R(\epsilon))^2 + C_\delta^-(R(\epsilon))^2 + \frac{S}{4n\epsilon}. \qquad (49)$$

Recall $|C_\delta^\pm(R(\epsilon))| = |\tilde{f}'(R(\epsilon) \pm \delta) - \tilde{f}'(R(\epsilon))|$. By (42), (45) and $R(\epsilon) \geq \epsilon$ we have

$$|\tilde{f}'(R(\epsilon))| \leq \frac{1}{2}\Big(\rho_n + \frac{S}{\sqrt{R(\epsilon)}}\Big) \leq \frac{1}{2}\Big(\rho_n + \frac{S}{\sqrt{\epsilon}}\Big) \qquad (50)$$

Thus, as $\epsilon \geq s_n$,

$$|C_\delta^\pm(R(\epsilon))| \leq \rho_n + \frac{S}{\sqrt{\epsilon - \delta}} \leq \rho_n + \frac{S}{\sqrt{s_n - \delta}}.$$

We reach

$$\int_{s_n}^{2s_n} d\epsilon \left\{ C_\delta^+(R(\epsilon))^2 + C_\delta^-(R(\epsilon))^2 \right\}$$

$$\leq \Big(\rho_n + \frac{S}{\sqrt{s_n - \delta}}\Big) \int_{s_n}^{2s_n} d\epsilon \left\{ C_\delta^+(R(\epsilon)) + C_\delta^-(R(\epsilon)) \right\}$$

$$\leq \Big(\rho_n + \frac{S}{\sqrt{s_n - \delta}}\Big) \int_{R(s_n)}^{R(2s_n)} dR(\epsilon) \left\{ C_\delta^+(R(\epsilon)) + C_\delta^-(R(\epsilon)) \right\}$$

$$= \Big(\rho_n + \frac{S}{\sqrt{s_n - \delta}}\Big)\Big[\Big(\tilde{f}(R(s_n) + \delta) - \tilde{f}(R(s_n) - \delta)\Big)$$

$$+ \Big(\tilde{f}(R(2s_n) - \delta) - \tilde{f}(R(2s_n) + \delta)\Big)\Big]$$

where we used that the Jacobian of the $\mathcal{C}^1$-diffeomorphism $\epsilon \mapsto R(\epsilon)$ is $\geq 1$ (by regularity) for the second inequality. The mean value theorem and (50) imply $|\tilde{f}(R(\epsilon) - \delta) - \tilde{f}(R(\epsilon) + \delta)| \leq \delta(\rho_n + \frac{S}{\sqrt{s_n - \delta}})$. Therefore

$$\int_{s_n}^{2s_n} d\epsilon \left\{ C_\delta^+(R(\epsilon))^2 + C_\delta^-(R(\epsilon))^2 \right\} \leq 2\delta\Big(\rho_n + \frac{S}{\sqrt{s_n - \delta}}\Big)^2.$$

Set $\delta = \delta_n = o(s_n)$. Thus, integrating (49) over $\epsilon \in [s_n, 2s_n]$ yields

$$\int_{s_n}^{2s_n} d\epsilon\, \mathbb{E}\big[(\langle \mathcal{L} \rangle_t - \mathbb{E}\langle \mathcal{L} \rangle_t)^2\big]$$

$$\leq \frac{27 s_n}{n\delta_n^2}\Big(C\lambda_n^2 \rho_n^3 + S(2s_n + \lambda_n \rho_n + \delta_n)\Big) + 18\delta_n\Big(\rho_n + \frac{S}{\sqrt{s_n - \delta_n}}\Big)^2 + \frac{9S\ln 2}{4n}$$

$$\leq \frac{C s_n \lambda_n \rho_n}{n\delta_n^2}(1 + \lambda_n \rho_n^2) + \frac{C\delta_n}{s_n} + \frac{C}{n}$$

where the constant $C$ is generic, and may change from place to place. Finally we optimize the bound choosing $\delta_n^3 = s_n^2 \lambda_n \rho_n(1 + \lambda_n \rho_n^2)/n$. We verify the condition $\delta_n = o(s_n)$: we have $(\delta_n/s_n)^3 = O(\lambda_n \rho_n(1 + \lambda_n \rho_n^2)/(ns_n))$ which, by (15), indeed tends to $0_+$ for an appropriately chosen sequence $s_n$. So the dominating term $\delta_n/s_n$ gives the result. $\qquad \square$

# E  Proof of inequality (38)

Let us drop the index in the bracket $\langle - \rangle_t$ and simply denote $R \equiv R_n(t, \epsilon)$. We start by proving the identity

$$-2\,\mathbb{E}\langle Q(\mathcal{L} - \mathbb{E}\langle \mathcal{L} \rangle)\rangle = \mathbb{E}\langle (Q - \mathbb{E}\langle Q \rangle)^2\rangle + \mathbb{E}\langle (Q - \langle Q \rangle)^2\rangle. \qquad (51)$$

Using the definitions $Q \equiv \frac{1}{n} \boldsymbol{x} \cdot \boldsymbol{X}$ and (37) gives

$$2\,\mathbb{E}\langle Q(\mathcal{L} - \mathbb{E}\langle\mathcal{L}\rangle)\rangle = \mathbb{E}\Big[\frac{1}{n}\langle Q\|\boldsymbol{x}\|^2\rangle - 2\langle Q^2\rangle - \frac{1}{n\sqrt{R}}\langle Q(\tilde{\boldsymbol{Z}}\cdot\boldsymbol{x})\rangle\Big]$$
$$- \mathbb{E}\langle Q\rangle\,\mathbb{E}\Big[\frac{1}{n}\langle\|\boldsymbol{x}\|^2\rangle - 2\langle Q\rangle - \frac{1}{n\sqrt{R}}\tilde{\boldsymbol{Z}}\cdot\langle\boldsymbol{x}\rangle\Big]. \qquad (52)$$

The gaussian integration by part formula (59) with Hamiltonian (32) yields

$$\frac{1}{n\sqrt{R}}\mathbb{E}\langle Q(\tilde{\boldsymbol{Z}}\cdot\boldsymbol{x})\rangle = \frac{1}{n}\mathbb{E}\langle Q\|\boldsymbol{x}\|^2\rangle - \frac{1}{n}\mathbb{E}\langle Q(\boldsymbol{x}\cdot\langle\boldsymbol{x}\rangle)\rangle \overset{\mathrm{N}}{=} \frac{1}{n}\mathbb{E}\langle Q\|\boldsymbol{x}\|^2\rangle - \mathbb{E}[\langle Q\rangle^2].$$

Fort the last equality we used the Nishimori identity as follows

$$\frac{1}{n}\mathbb{E}\langle Q(\boldsymbol{x}\cdot\langle\boldsymbol{x}\rangle)\rangle = \frac{1}{n^2}\mathbb{E}\langle(\boldsymbol{x}\cdot\boldsymbol{X})(\boldsymbol{x}\cdot\langle\boldsymbol{x}\rangle)\rangle \overset{\mathrm{N}}{=} \frac{1}{n^2}\mathbb{E}\langle(\boldsymbol{X}\cdot\boldsymbol{x})(\boldsymbol{X}\cdot\langle\boldsymbol{x}\rangle)\rangle = \mathbb{E}[\langle Q\rangle^2].$$

Note that we already proved (41), namely

$$\frac{1}{n\sqrt{R}}\mathbb{E}\langle\tilde{\boldsymbol{Z}}\cdot\boldsymbol{x}\rangle = \frac{1}{n}\mathbb{E}\langle\|\boldsymbol{x}\|^2\rangle - \mathbb{E}\langle Q\rangle.$$

Therefore (52) finally simplifies to

$$2\,\mathbb{E}\langle Q(\mathcal{L} - \mathbb{E}\langle\mathcal{L}\rangle)\rangle = \mathbb{E}[\langle Q\rangle^2] - 2\,\mathbb{E}\langle Q^2\rangle + \mathbb{E}[\langle Q\rangle]^2$$
$$= -\big(\mathbb{E}\langle Q^2\rangle - \mathbb{E}[\langle Q\rangle]^2\big) - \big(\mathbb{E}\langle Q^2\rangle - \mathbb{E}[\langle Q\rangle^2]\big).$$

which is identity (51).

This identity implies the inequality

$$2\big|\mathbb{E}\langle Q(\mathcal{L} - \mathbb{E}\langle\mathcal{L}\rangle)\rangle\big| = 2\big|\mathbb{E}\langle(Q - \mathbb{E}\langle Q\rangle)(\mathcal{L} - \mathbb{E}\langle\mathcal{L}\rangle)\rangle\big| \geq \mathbb{E}\langle(Q - \mathbb{E}\langle Q\rangle)^2\rangle$$

and an application of the Cauchy-Schwarz inequality gives

$$2\big\{\mathbb{E}\langle(Q - \mathbb{E}\langle Q\rangle)^2\rangle\,\mathbb{E}\langle(\mathcal{L} - \mathbb{E}\langle\mathcal{L}\rangle)^2\rangle\big\}^{1/2} \geq \mathbb{E}\langle(Q - \mathbb{E}\langle Q\rangle)^2\rangle.$$

This ends the proof of (38).

# F Heurisitic derivation of the information theoretic phase transition

In this section we analyze the potential function in order to heuristically locate the information theoretic transition in the special case of the spiked Wigner model with Bernoulli prior $P_X = \mathrm{Ber}(\rho)$. The main hypotheses behind this computation are $i)$ that the SNR $\lambda = \lambda(\rho)$ varies with $\rho$ as $\lambda = 4\gamma|\ln\rho|/\rho$ with $\gamma > 0$ and independent of $\rho$; that $ii)$ in this SNR regime the potential possesses only two minima $\{q^+, q^-\}$ that approach, as $\rho \to 0_+$, the boundary values $q^- = o(\rho/|\ln\rho|)$ and $q^+ \to \rho$. For the Bernoulli prior the potential explicitly reads

$$i_n^{\mathrm{pot}}(q, \lambda, \rho)$$
$$\equiv \frac{\lambda(q^2 + \rho^2)}{4} - (1-\rho)\mathbb{E}\ln\Big\{1 - \rho + \rho e^{-\frac{1}{2}\lambda q + \sqrt{\lambda q}Z}\Big\} - \rho\,\mathbb{E}\ln\Big\{1 - \rho + \rho e^{\frac{1}{2}\lambda q + \sqrt{\lambda q}Z}\Big\}.$$

We used that

$$I(X; \sqrt{\gamma}X + Z) = -\mathbb{E}\ln\int dP_X(x)e^{-\frac{1}{2}\gamma x^2 + \gamma X x + \sqrt{\gamma}Z x} + \frac{1}{2}\mathbb{E}[X^2]\gamma. \qquad (53)$$

Let us compute this function around its assumed minima. Starting with $q^- = o(\rho/|\ln\rho|)$ (this means that this quantity goes to $0_+$ faster than $\rho/|\ln\rho|$ as $\rho$ vanishes) we obtain at leading order

after a careful Taylor expansion in $\lambda q^- \to 0_+$ (the symbol $\approx$ means equality up to lower order terms as $\rho \to 0_+$)

$$i_n^{\text{pot}}(q^-, \lambda, \rho) \approx \frac{\lambda(q^-)^2}{4} + \frac{\lambda\rho^2}{4} - \frac{\rho(\lambda q^-)^2}{8} \approx \frac{\lambda\rho^2}{4} = \gamma\rho|\ln\rho|. \tag{54}$$

For the other minimum $q^+ \to \rho$, because $\lambda q^+ \to +\infty$ the $Z$ contribution in the exponentials appearing in the potential can be dropped due to the precense of the square root. We obtain at leading order

$$i_n^{\text{pot}}(q^+, \lambda, \rho) \approx 2\gamma\rho|\ln\rho| - \ln\{1 + \rho^{1+2\gamma}\} - \rho\ln\{1 + \rho^{1-2\gamma}\}.$$

Here there are two cases to consider: $\gamma > 1/2$ and $0 < \gamma \leq 1/2$. We start with $\gamma > 1/2$. In this case the potential simplifies to

$$i_n^{\text{pot}}(q^+, \lambda, \rho) \approx \rho|\ln\rho|.$$

Now for $0 < \gamma \leq 1/2$ we have

$$i_n^{\text{pot}}(q^+, \lambda, \rho) \approx 2\gamma\rho|\ln\rho|.$$

The information theoretic threshold $\lambda_c = \lambda_c(\rho)$ is defined as the first non-analiticy in the mutual information. In the present setting this corresponds to a discontinuity of the first derivative w.r.t. the SNR of the mutual information (and we therefore speak about a "first-order phase transition"). By the I-MMSE formula this threshold manifests itself as a discontinuity in the MMSE. In the high sparsity regime $\rho \to 0_+$ the transition is actually as sharp as it can be with a $0$–$1$ behavior. This translates, at the level of the potential, as the SNR threshold where its minimum is attained at $q^-$ just below and instead at $q^+$ just above. So we equate $\lim_{\rho\to 0_+} i_n^{\text{pot}}(q^-, \lambda_c, \rho) = \lim_{\rho\to 0_+} i_n^{\text{pot}}(q^+, \lambda_c, \rho)$ and solve for $\lambda_c$. This is only possible, under the constraint $\gamma > 0$ independent of $\rho$, in the case $\gamma > 1/2$ and gives $\gamma = 1$ which is the claimed information theoretic threshold $\lambda_c(\rho) = 4|\ln\rho|/\rho$. Repeating this analysis for the Bernoulli-Rademacher prior $P_X = (1-\rho)\delta_0 + \frac{1}{2}\rho(\delta_{-1} + \delta_1)$ leads the same threshold, which suggests that the transition is only related (for discrete priors) to the recovery of the support of the signal.

Another piece of information gained from this analysis is that around the transition the mutual information divided by $n$ is $\Theta(\rho|\ln\rho|)$. Therefore the proper normalization for the mutual information is $(n\rho|\ln\rho|)^{-1}I(\boldsymbol{X}; \boldsymbol{W})$ for it to have a well defined non trivial limit in the regime $\rho \to 0_+$.

Finally for $\gamma \leq 1$ the minimum of the potential is attained at $q^-$ and the rescaled mutual information $(n\rho|\ln\rho|)^{-1}I(\boldsymbol{X}; \boldsymbol{W})$ equals $\gamma$ as seen from (54). If instead $\gamma \geq 1$ the minimum is attained at $q^+$ and the mutual information instead saturates to 1, so we get the asymptotic singular function $\gamma\mathbb{I}(\gamma \leq 1) + \mathbb{I}(\gamma \geq 1)$.

# G Heurisitic derivation of the AMP algorithmic transition

In this section we derive the AMP algorithmic transition for the spiked Wigner model in the Bernoulli case $P_{X,n} = \rho_n\delta_1 + (1-\rho_n)\delta_0$. The approach can be applied to the Bernoulli-Rademacher case as well (and probably more generically), and leads to the same scaling for the AMP threshold. The derivation starts from the state evolution recursion for the overlap of AMP (10), or equivalently,

$$\tau_{t+1}^n = \mathbb{E}\Big\{X_0^n \, \mathbb{E}\big\{X_0^n \mid \sqrt{\lambda_n}\tau_t^n X_0^n + \sqrt{\tau_t^n}Z\big\}\Big\}, \qquad \tau_0^n = 0,$$

which, in the Bernoulli case, reads as (recall $Z \sim \mathcal{N}(0,1)$),

$$\tau_{t+1}^n = \mathbb{E}\left\{\frac{\rho_n^2}{\rho_n + (1-\rho_n)\exp\{-\frac{1}{2}\lambda_n\tau_t^n - \sqrt{\lambda_n\tau_t^n}Z\}}\right\}. \tag{55}$$

Therefore by plugging $\tau_0^n = 0$ in the recursion we get $\tau_1^n = \rho_n^2$, and then

$$\tau_2^n = \mathbb{E}\left\{\frac{\rho_n^2}{\rho_n + (1-\rho_n)\exp\{-\frac{1}{2}\lambda_n\rho_n^2 - \sqrt{\lambda_n\rho_n^2}Z\}}\right\}.$$

Now depending on $\lambda_n \rho_n^2 \gg 1$ or $\lambda_n \rho_n^2 \ll 1$ the next step of the recursion has two very different behaviors. When $\lambda_n \rho_n^2 \gg 1$, it becomes

$$\lambda_n \rho_n^2 \gg 1 : \qquad \tau_2^n \approx \rho_n \qquad \text{and thus} \qquad \lambda_n \tau_2^n \approx \lambda_n \rho_n \gg 1.$$

Therefore, the recursion will remain stuck in this "reconstruction state" and converges towards $\tau_\infty^n \approx \rho_n$ which yields the minimal value of the MSE:

$$\frac{\mathrm{MSE}_{\mathrm{AMP}}^\infty}{(n\rho_n)^2} = 1 - \left(\frac{\tau_\infty^n}{\rho_n}\right)^2 \approx 0.$$

When $\lambda_n \rho_n^2 \ll 1$,

$$\lambda_n \rho_n^2 \ll 1 : \qquad \tau_2^n \approx \rho_n^2 = \tau_1^n.$$

In this case, the recursion converges towards the "no reconstruction state" $\tau_\infty^n \approx \rho_n^2$, which corresponds to the MSE of a random guess (according to the prior) for the spike signal-matrix, i.e., the MSE corresponding to take as estimator $\boldsymbol{X}' \otimes \boldsymbol{X}'$ where $\boldsymbol{X}' \sim P_{X,n}$ is independent from the ground-truth $\boldsymbol{X}$:

$$\frac{\mathrm{MSE}_{\mathrm{AMP}}^\infty}{(n\rho_n)^2} = 1 - \left(\frac{\tau_\infty^n}{\rho_n}\right)^2 \approx 1.$$

This reasoning shows that the behavior of the state evolution must change for a scaling $\lambda_n \rho_n^2 = O(1)$. This argument cannot catch the constant $\lambda_n \rho_n^2 \approx 1/e$, which was numerically approximated in [52].

# H  The Nishimori identity

**Lemma 7** (Nishimori identity). *Let $(\boldsymbol{X}, \boldsymbol{Y})$ be a couple of random variables with joint distribution $P(\boldsymbol{X}, \boldsymbol{Y})$ and conditional distribution $P(\boldsymbol{X}|\boldsymbol{Y})$. Let $k \geq 1$ and let $\boldsymbol{x}^{(1)}, \dots, \boldsymbol{x}^{(k)}$ be i.i.d. samples from the conditional distribution. We use the bracket $\langle - \rangle$ for the expectation w.r.t. the product measure $P(\boldsymbol{x}^{(1)}|\boldsymbol{Y})P(\boldsymbol{x}^{(2)}|\boldsymbol{Y})\dots P(\boldsymbol{x}^{(k)}|\boldsymbol{Y})$ and $\mathbb{E}$ for the expectation w.r.t. the joint distribution. Then, for all continuous bounded function $g$ we have*

$$\mathbb{E}\langle g(\boldsymbol{Y}, \boldsymbol{x}^{(1)}, \dots, \boldsymbol{x}^{(k)})\rangle = \mathbb{E}\langle g(\boldsymbol{Y}, \boldsymbol{X}, \boldsymbol{x}^{(2)}, \dots, \boldsymbol{x}^{(k)})\rangle.$$

*Proof.* This is a simple consequence of Bayes formula. It is equivalent to sample the couple $(\boldsymbol{X}, \boldsymbol{Y})$ according to its joint distribution or to sample first $\boldsymbol{Y}$ according to its marginal distribution and then to sample $\boldsymbol{X}$ conditionally on $\boldsymbol{Y}$ from the conditional distribution. Thus the two $(k+1)$-tuples $(\boldsymbol{Y}, \boldsymbol{x}^{(1)}, \dots, \boldsymbol{x}^{(k)})$ and $(\boldsymbol{Y}, \boldsymbol{X}, \boldsymbol{x}^{(2)}, \dots, \boldsymbol{x}^{(k)})$ have the same law. $\qquad\square$

# I  I-MMSE relation

In this appendix we prove the I-MMSE relation of [58, 59] for the convenience of the reader.

**Lemma 8** (I-MMSE formula). *Consider a signal $\boldsymbol{X} \in \mathbb{R}^n$ with $\boldsymbol{X} \sim P_X$ that has finite support, and gaussian corrupted data $\boldsymbol{Y} \sim \mathcal{N}(\sqrt{R}\,\boldsymbol{X}, \mathrm{I}_n)$ and possibly additional generic data $\boldsymbol{W} \sim P_{W|X}(\cdot\,|\boldsymbol{X})$ with $H(\boldsymbol{W})$ bounded. The* I-MMSE formula *linking the mutual information and the MMSE then reads*

$$\frac{d}{dR} I(\boldsymbol{X}; (\boldsymbol{Y}, \boldsymbol{W})) = \frac{d}{dR} I(\boldsymbol{X}; \boldsymbol{Y}|\boldsymbol{W}) = \frac{1}{2}\mathrm{MMSE}(\boldsymbol{X}|\boldsymbol{Y}, \boldsymbol{W}) = \frac{1}{2}\mathbb{E}\|\boldsymbol{X} - \langle \boldsymbol{x}\rangle\|^2, \qquad (56)$$

*where the Gibbs-bracket $\langle - \rangle$ is the expectation acting on $\boldsymbol{x} \sim P(\cdot\,|\boldsymbol{Y}, \boldsymbol{W})$.*

*Proof.* First note that by the chain rule for mutual information $I(\boldsymbol{X}; (\boldsymbol{Y}, \boldsymbol{W})) = I(\boldsymbol{X}; \boldsymbol{Y}|\boldsymbol{W}) + I(\boldsymbol{X}; \boldsymbol{W})$, so the derivatives in (56) are equal. We will now look at $\frac{d}{dR} I(\boldsymbol{X}; (\boldsymbol{Y}, \boldsymbol{W}))$. Since, conditionally on $\boldsymbol{X}$, $\boldsymbol{Y}$ and $\boldsymbol{W}$ are independent, we have

$$I(\boldsymbol{X}; (\boldsymbol{Y}, \boldsymbol{W})) = H(\boldsymbol{Y}, \boldsymbol{W}) - H(\boldsymbol{Y}, \boldsymbol{W}|\boldsymbol{X}) = H(\boldsymbol{Y}, \boldsymbol{W}) - H(\boldsymbol{Y}|\boldsymbol{X}) - H(\boldsymbol{W}|\boldsymbol{X}).$$

With gaussian noise contribution $H(\boldsymbol{Y}|\boldsymbol{X}) = \frac{n}{2}\ln(2\pi e)$. Therefore only $H(\boldsymbol{Y},\boldsymbol{W})$ depends on $R$. Let us then compute, using the change of variable $\boldsymbol{Y} = \sqrt{R}\,\boldsymbol{X} + \boldsymbol{Z}$,

$$
\begin{aligned}
\frac{d}{dR}I\big(\boldsymbol{X};(\boldsymbol{Y},\boldsymbol{W})\big) &= \frac{d}{dR}H(\boldsymbol{Y},\boldsymbol{W}) \\
&= -\frac{d}{dR}\int dP_X(\boldsymbol{X})d\boldsymbol{Y}\,d\boldsymbol{W}\,P_{W|X}(\boldsymbol{W}|\boldsymbol{X})\frac{e^{-\frac{1}{2}\|\boldsymbol{Y}-\sqrt{R}\boldsymbol{X}\|^2}}{(2\pi)^{n/2}} \\
&\qquad \times \ln\int dP_X(\boldsymbol{x})P_{W|X}(\boldsymbol{W}|\boldsymbol{x})\frac{e^{-\frac{1}{2}\|\boldsymbol{Y}-\sqrt{R}\boldsymbol{x}\|^2}}{(2\pi)^{n/2}} \\
&= -\int dP_X(\boldsymbol{X})d\boldsymbol{Z}\,d\boldsymbol{W}\,P_{W|X}(\boldsymbol{W}|\boldsymbol{X})\frac{e^{-\frac{1}{2}\|\boldsymbol{Z}\|^2}}{(2\pi)^{n/2}} \\
&\qquad \times \frac{d}{dR}\ln\int dP_X(\boldsymbol{x})P_{W|X}(\boldsymbol{W}|\boldsymbol{x})\frac{e^{-\frac{1}{2}\|\boldsymbol{Z}-\sqrt{R}(\boldsymbol{x}-\boldsymbol{X})\|^2}}{(2\pi)^{n/2}} \\
&= \frac{1}{2\sqrt{R}}\mathbb{E}_{\boldsymbol{X},\boldsymbol{Z},\boldsymbol{W}|\boldsymbol{X}}\big\langle (\boldsymbol{Z}+\sqrt{R}(\boldsymbol{X}-\boldsymbol{x}))\cdot(\boldsymbol{X}-\boldsymbol{x})\big\rangle
\end{aligned}
\tag{57}
$$

where $\boldsymbol{Z}\sim\mathcal{N}(0,\mathrm{I}_n)$ and the bracket notation is the expectation w.r.t. the posterior proportional to

$$
dP_X(\boldsymbol{x})dP_{W|X}(\boldsymbol{W}|\boldsymbol{x})d\boldsymbol{Z}\exp\Big\{-\frac{1}{2}\|\boldsymbol{Z}-\sqrt{R}(\boldsymbol{x}-\boldsymbol{X})\|^2\Big\}.
$$

In (57) the interchange of derivative and integrals is permitted by a standard application of Lebesgue's dominated convergence theorem in the case where the support of $P_X$ is bounded. Now we use the following gaussian integration by part formula: for any bounded function $\boldsymbol{g}:\mathbb{R}^n\mapsto\mathbb{R}^n$ of a standard gaussian random vector $\boldsymbol{Z}\sim\mathcal{N}(0,\mathrm{I}_n)$ we obviously have

$$
\mathbb{E}[\boldsymbol{Z}\cdot\boldsymbol{g}(\boldsymbol{Z})] = \mathbb{E}[\nabla_{\boldsymbol{Z}}\cdot\boldsymbol{g}(\boldsymbol{Z})].
\tag{58}
$$

This formula applied to a Gibbs-bracket associated to a general Gibbs distribution with hamiltonian $\mathcal{H}(\boldsymbol{x},\boldsymbol{Z})$ (depending on the Gaussian noise and possibly other variables) yields

$$
\begin{aligned}
\mathbb{E}[\boldsymbol{Z}\cdot\langle\boldsymbol{h}(\boldsymbol{x})\rangle] &= \mathbb{E}\,\nabla_{\boldsymbol{Z}}\cdot\frac{\int dP(\boldsymbol{x})e^{-\mathcal{H}(\boldsymbol{x},\boldsymbol{Z})}\boldsymbol{h}(\boldsymbol{x})}{\int dP(\boldsymbol{x}')e^{-\mathcal{H}(\boldsymbol{x}',\boldsymbol{Z})}} \\
&= -\mathbb{E}\,\frac{\int dP_X(\boldsymbol{x})e^{-\mathcal{H}(\boldsymbol{x},\boldsymbol{Z})}\boldsymbol{h}(\boldsymbol{x})\cdot\nabla_{\boldsymbol{Z}}\mathcal{H}(\boldsymbol{x},\boldsymbol{Z})}{\int dP_X(\boldsymbol{x}')e^{-\mathcal{H}(\boldsymbol{x}',\boldsymbol{Z})}} \\
&\qquad + \mathbb{E}\Big[\frac{\int dP_X(\boldsymbol{x})e^{-\mathcal{H}(\boldsymbol{x},\boldsymbol{Z})}\boldsymbol{h}(\boldsymbol{x})}{\int dP_X(\boldsymbol{x}')e^{-\mathcal{H}(\boldsymbol{x}',\boldsymbol{Z})}}\cdot\frac{\int dP_X(\boldsymbol{x})e^{-\mathcal{H}(\boldsymbol{x},\boldsymbol{Z})}\nabla_{\boldsymbol{Z}}\mathcal{H}(\boldsymbol{x},\boldsymbol{Z})}{\int dP_X(\boldsymbol{x}')e^{-\mathcal{H}(\boldsymbol{x}',\boldsymbol{Z})}}\Big] \\
&= -\mathbb{E}\langle\boldsymbol{h}(\boldsymbol{x})\cdot\nabla_{\boldsymbol{Z}}\mathcal{H}(\boldsymbol{x},\boldsymbol{Z})\rangle + \mathbb{E}\big[\langle\boldsymbol{h}(\boldsymbol{x})\rangle\cdot\langle\nabla_{\boldsymbol{Z}}\mathcal{H}(\boldsymbol{x},\boldsymbol{Z})\rangle\big].
\end{aligned}
\tag{59}
$$

Applied to (57), where the "hamiltonian" is $\mathcal{H}(\boldsymbol{x},\boldsymbol{Z}) = -\ln P_{W|X}(\boldsymbol{W}|\boldsymbol{x}) + \frac{1}{2}\|\boldsymbol{Z}-\sqrt{R}(\boldsymbol{x}-\boldsymbol{X})\|^2$, this identity gives

$$
\begin{aligned}
\frac{d}{dR}I\big(\boldsymbol{X};(\boldsymbol{Y},\boldsymbol{W})\big) &= \frac{1}{2}\mathbb{E}\big[\langle\|\boldsymbol{X}-\boldsymbol{x}\|^2\rangle + \frac{1}{\sqrt{R}}\nabla_{\boldsymbol{Z}}\cdot\langle\boldsymbol{X}-\boldsymbol{x}\rangle\big] \\
&= \frac{1}{2}\mathbb{E}\big[\langle\|\boldsymbol{X}-\boldsymbol{x}\|^2\rangle - \frac{1}{\sqrt{R}}\langle(\boldsymbol{X}-\boldsymbol{x})\cdot(\boldsymbol{Z}+\sqrt{R}(\boldsymbol{X}-\boldsymbol{x}))\rangle \\
&\qquad + \frac{1}{\sqrt{R}}\langle(\boldsymbol{X}-\boldsymbol{x})\rangle\cdot\langle\boldsymbol{Z}+\sqrt{R}(\boldsymbol{X}-\boldsymbol{x})\rangle\big] \\
&= \frac{1}{2}\mathbb{E}\|\boldsymbol{X}-\langle\boldsymbol{x}\rangle\|^2.
\end{aligned}
$$

$\square$

The MMSE cannot increase when the SNR increases. This translates into the concavity of the mutual information of gaussian channels as a function of the SNR.

**Lemma 9** (Concavity of the mutual information in the SNR). *Consider the same setting as Lemma 8. Then the mutual informations $I(\boldsymbol{X};(\boldsymbol{Y},\boldsymbol{W}))$ and $I(\boldsymbol{X};\boldsymbol{Y}|\boldsymbol{W})$ are concave in the SNR of the gaussian channel:*

$$\frac{d^2}{dR^2}I\big(\boldsymbol{X};(\boldsymbol{Y},\boldsymbol{W})\big) = \frac{d^2}{dR^2}I(\boldsymbol{X};\boldsymbol{Y}|\boldsymbol{W})$$

$$= \frac{1}{2}\frac{d}{dR}\mathrm{MMSE}(\boldsymbol{X}|\boldsymbol{Y},\boldsymbol{W}) = -\frac{1}{2n}\sum_{i,j=1}^{n}\mathbb{E}\big[(\langle x_i x_j\rangle - \langle x_i\rangle\langle x_j\rangle)^2\big] \leq 0$$

*where the Gibbs-bracket $\langle - \rangle$ is the expectation acting on $\boldsymbol{x} \sim P(\cdot\,|\boldsymbol{Y},\boldsymbol{W})$.*

*Proof.* Set $Q \equiv \boldsymbol{x}\cdot\boldsymbol{X}/n$ where $\boldsymbol{x} \sim P(\cdot\,|\boldsymbol{Y},\boldsymbol{W})$. From a Nishimori identity $\mathrm{MMSE}(\boldsymbol{X}|\boldsymbol{Y},\boldsymbol{W}) = \mathbb{E}_{P_X}[X^2] - \mathbb{E}\langle Q\rangle$. Thus by the I-MMSE formula we have, by a calculation similar to (59),

$$-2\frac{d^2}{dR^2}I\big(\boldsymbol{X};(\boldsymbol{Y},\boldsymbol{W})\big) = \frac{d\,\mathbb{E}\langle Q\rangle}{dR} = n\mathbb{E}[\langle Q\rangle\langle\mathcal{L}\rangle - \langle Q\mathcal{L}\rangle] \tag{60}$$

where we have set

$$\mathcal{L} \equiv \frac{1}{n}\Big(\frac{1}{2}\|\boldsymbol{x}\|^2 - \boldsymbol{x}\cdot\boldsymbol{X} - \frac{1}{2\sqrt{R}}\boldsymbol{x}\cdot\boldsymbol{Z}\Big).$$

Now we look at each term on the right hand side of this equality. The calculation of appendix E shows that

$$-\mathbb{E}\langle Q\mathcal{L}\rangle = \mathbb{E}\langle Q^2\rangle - \frac{1}{2}\mathbb{E}[\langle Q\rangle^2]$$

so it remains to compute

$$\mathbb{E}[\langle Q\rangle\langle\mathcal{L}\rangle] = \mathbb{E}\Big[\langle Q\rangle\frac{\langle\|\boldsymbol{x}\|^2\rangle}{2n} - \langle Q\rangle^2 - \langle Q\rangle\frac{\boldsymbol{Z}\cdot\langle\boldsymbol{x}\rangle}{2n\sqrt{R}}\Big].$$

By formulas (58) and (59) in which the Hamiltonian is (32) we have

$$-\frac{1}{2n\sqrt{R}}\mathbb{E}\big[\boldsymbol{Z}\cdot\langle\boldsymbol{x}\rangle\langle Q\rangle\big] = -\frac{1}{2n\sqrt{R}}\mathbb{E}\big[\langle Q\rangle\nabla_{\boldsymbol{Z}}\cdot\langle\boldsymbol{x}\rangle + \langle\boldsymbol{x}\rangle\cdot\nabla\langle Q\rangle\big]$$

$$= -\frac{1}{2n}\mathbb{E}\big[\langle Q\rangle\big(\langle\|\boldsymbol{x}\|^2\rangle - \|\langle\boldsymbol{x}\rangle\|^2\big) + \langle\boldsymbol{x}\rangle\cdot\big(\langle Q\boldsymbol{x}\rangle - \langle Q\rangle\langle\boldsymbol{x}\rangle\big)\big]$$

$$\overset{\mathrm{N}}{=} -\frac{1}{2n}\mathbb{E}\big[\langle Q\rangle\langle\|\boldsymbol{x}\|^2\rangle\big] + \frac{1}{n}\mathbb{E}\big[\langle Q\rangle\|\langle\boldsymbol{x}\rangle\|^2\big] - \frac{1}{2}\mathbb{E}[\langle Q\rangle^2].$$

In the last equality we used the following consequence of the Nishimori identity. Let $\boldsymbol{x},\boldsymbol{x}^{(2)}$ be two replicas, i.e., conditionally (on the data) independent samples from the posterior (31). Then

$$\frac{1}{n}\mathbb{E}\big[\langle\boldsymbol{x}\rangle\cdot\langle Q\boldsymbol{x}\rangle\big] = \frac{1}{n^2}\mathbb{E}\big\langle(\boldsymbol{x}^{(2)}\cdot\boldsymbol{x})(\boldsymbol{x}\cdot\boldsymbol{X})\big\rangle \overset{\mathrm{N}}{=} \frac{1}{n^2}\mathbb{E}\big\langle(\boldsymbol{x}^{(2)}\cdot\boldsymbol{X})(\boldsymbol{X}\cdot\boldsymbol{x})\big\rangle = \mathbb{E}[\langle Q\rangle^2].$$

Thus we obtain

$$\mathbb{E}[\langle Q\rangle\langle\mathcal{L}\rangle - \langle Q\mathcal{L}\rangle] = \mathbb{E}\langle Q^2\rangle - 2\mathbb{E}[\langle Q\rangle^2] + \frac{1}{n}\mathbb{E}\big[\langle Q\rangle\|\langle\boldsymbol{x}\rangle\|^2\big]$$

$$= \frac{1}{n^2}\mathbb{E}\big\langle(\boldsymbol{x}\cdot\boldsymbol{X})^2 - 2(\boldsymbol{x}\cdot\boldsymbol{X})(\boldsymbol{x}^{(1)}\cdot\boldsymbol{X}) + (\boldsymbol{x}\cdot\boldsymbol{X})(\boldsymbol{x}^{(2)}\cdot\boldsymbol{x}^{(3)})\big\rangle$$

$$\overset{\mathrm{N}}{=} \frac{1}{n^2}\mathbb{E}\big\langle(\boldsymbol{x}\cdot\boldsymbol{x}^{(0)})^2 - 2(\boldsymbol{x}\cdot\boldsymbol{x}^{(0)})(\boldsymbol{x}^{(1)}\cdot\boldsymbol{x}^{(0)}) + (\boldsymbol{x}\cdot\boldsymbol{x}^{(0)})(\boldsymbol{x}^{(2)}\cdot\boldsymbol{x}^{(3)})\big\rangle$$

where $\boldsymbol{x}^{(0)},\boldsymbol{x},\boldsymbol{x}^{(1)},\boldsymbol{x}^{(2)},\boldsymbol{x}^{(3)}$ are replicas and the last equality again follows from a Nishimori identity. Multiplying this identity by $n$ and rewriting the inner products component-wise we get

$$\frac{d\,\mathbb{E}\langle Q\rangle}{dR} = \frac{1}{n}\sum_{i,j=1}^{n}\mathbb{E}\big\langle x_i x_i^{(0)} x_j x_j^{(0)} - 2x_i x_i^{(0)} x_j^{(1)} x_j^{(0)} + x_i x_i^{(0)} x_j^{(2)} x_j^{(3)}\big\rangle$$

$$= \frac{1}{n}\sum_{i,j=1}^{n}\mathbb{E}\big[\langle x_i x_j\rangle^2 - 2\langle x_i\rangle\langle x_j\rangle\langle x_i x_j\rangle + \langle x_i\rangle^2\langle x_j\rangle^2\big] \tag{61}$$

Using (60) this ends the proof of the lemma. Note that we have also shown the positivity claimed in (30) of section B. $\qquad\square$

# J Proof of corollary 1

The proof of corollary 1 follows from a combination of theorem 1 and the I-MMSE relation (see [58, 59], and also appendix I). Denote

$$\mathrm{M}_n(s) \equiv \frac{1}{(n\rho_n)^2} \mathrm{MMSE}((X_iX_j)_{i<j}|\boldsymbol{W})|_{\lambda_n=s} \quad \text{and} \quad I_n(s) \equiv \frac{1}{n\rho_n^2} I(\boldsymbol{X};\boldsymbol{W})|_{\lambda_n=s}. \quad (62)$$

The I-MMSE relation in its integral formulation implies

$$\frac{I_n(s+\epsilon) - I_n(s)}{\epsilon} = \frac{1}{2\epsilon} \int_s^{s+\epsilon} \mathrm{M}_n(\lambda)d\lambda. \quad (63)$$

Because $M_n(s)$ is a non-increasing function ("information can't hurt", which is equivalent to the concavity of mutual information in the signal-to-noise ratio, see [58, 59] or lemma 9) the above identity implies

$$\frac{\mathrm{M}_n(s+\epsilon)}{2} \leq \frac{I_n(s+\epsilon) - I_n(s)}{\epsilon} \leq \frac{\mathrm{M}_n(s)}{2}. \quad (64)$$

Set

$$i_n(s) \equiv \frac{1}{\rho_n^2} \inf_{q\in[0,\rho_n]} i_n^{\mathrm{pot}}(q,s,\rho_n) \quad \text{so that} \quad \frac{1}{2}m_n(s,\rho_n) \equiv \frac{d}{ds}i_n(s).$$

Because $s \mapsto m_n(s,\rho_n)$ is also a non-increasing function (see, e.g., [15]) we obtain similarly

$$\frac{m_n(s+\epsilon,\rho_n)}{2} \leq \frac{i_n(s+\epsilon) - i_n(s)}{\epsilon} \leq \frac{m_n(s,\rho_n)}{2}. \quad (65)$$

Set $c_n \equiv C(\ln n)^{1/3}n^{-(1-6\beta)/7}|\ln\rho_n|/\rho_n$ which is the right-hand side of (5) multiplied by $(\rho_n|\ln\rho_n|)/\rho_n^2$. Theorem 1 then implies

$$\frac{\mathrm{M}_n(s+\epsilon)}{2} \leq \frac{i_n(s+\epsilon) - i_n(s) + 2c_n}{\epsilon} \leq \frac{m_n(s,\rho_n)}{2} + \frac{2c_n}{\epsilon}, \quad (66)$$

$$\frac{m_n(s+\epsilon,\rho_n)}{2} - \frac{2c_n}{\epsilon} \leq \frac{i_n(s+\epsilon) - i_n(s) - 2c_n}{\epsilon} \leq \frac{\mathrm{M}_n(s)}{2}. \quad (67)$$

Replacing $\rho_n = \Omega(n^{-\beta})$ with $\beta \in [0, 1/13)$ yields the claimed inequality:

$$m_n(s+\epsilon,\rho_n) - \frac{C'}{\epsilon}\frac{(\ln n)^{4/3}}{n^{(1-13\beta)/7}} \leq \mathrm{M}_n(s) \leq m_n(s-\epsilon,\rho_n) + \frac{C'}{\epsilon}\frac{(\ln n)^{4/3}}{n^{(1-13\beta)/7}}. \quad (68)$$

# K AMP algorithmic phase transition

In this appendix, we prove theorem 2. To do this, we begin by introducing a general 'symmetric' AMP algorithm in section K.1 and show it is quite similar to the AMP algorithm in (6). For this symmetric AMP algorithm, we provide finite sample guarantees like those given in [63] for various 'non-symmetric' AMP algorithms. However, we have an added challenge in that terms like the Lipschitz constant of the denoiser $f_t$ in (9) and the state evolution values in (10) depend on $n$ and therefore cannot be treated as universal constants in the rate of concentration, as they were in [63]. The main concentration result for the symmetric AMP is given in theorem 4 in section K.1. Then, in section K.2, we use theorem 4 to prove result (11), from which we prove theorem 2.

## K.1 Symmetric AMP finite sample guarantees

We begin by analyzing a 'symmetric' AMP algorithm, described now, that is similar to the AMP algorithm in (6). Assume the matrix $\boldsymbol{Z} \sim \mathrm{GOE}(n)$ is an $n \times n$ matrix form the gaussian orthogonal ensemble, i.e. $\boldsymbol{Z}$ is a symmetric matrix with $\{Z_{ij}\}_{1\leq i\leq j\leq n}$ i.i.d. $\mathcal{N}(0, 1/n)$, and $\{Z_{ii}\}_{1\leq i\leq n}$ i.i.d. $\mathcal{N}(0, 2/n)$. Start with an initial condition $\boldsymbol{h}^0 \in \mathbb{R}^n$, independent of $\boldsymbol{Z}$, and calculate for $t \geq 0$,

$$\boldsymbol{h}^{t+1} = \boldsymbol{Z}g_t(\boldsymbol{h}^t, \boldsymbol{X}^n) - \mathrm{c}_t g_{t-1}(\boldsymbol{h}^{t-1}, \boldsymbol{X}^n). \quad (69)$$

In the above, $g_t : \mathbb{R}^2 \to \mathbb{R}$ is Lipschitz and separable (i.e. it acts component-wise when applied to vectors) and may depend on $n$ through its Lipschitz constant, denoted $L_g^n$. The function $g_t$ takes as its second argument a random vector $\boldsymbol{X}^n \in \mathbb{R}^n$ with entries that are i.i.d. $P_{X,n}$, a sub-gaussian distribution, $\mathsf{c}_t = \frac{1}{n}\sum_{i=1}^n g_t'(h_i^t, X_i^n)$, the derivative taken with respect to the first argument, and all terms with negative indices take the value 0 (so that, for example, $\boldsymbol{h}^1 = \boldsymbol{Z} g_0(\boldsymbol{h}^0, \boldsymbol{X}^n)$). The key result, stated in theorem 4 below, is that for each $t \geq 1$, the empirical distribution of the components of $\boldsymbol{h}^t$ is approximately equal in distribution to a gaussian $\mathcal{N}(0, \sigma_t^n)$ where the variances $\{\sigma_t^n\}_{t\geq 0}$ are defined via the state evolution: initialize with $\sigma_1^n = \|g_0(\boldsymbol{h}^0, \boldsymbol{X}^n)\|^2/n$, calculate for $t \geq 1$,

$$\sigma_{t+1}^n = \mathbb{E}\left[\left(g_t(\sqrt{\sigma_t^n}Z, X_0^n)\right)^2\right], \tag{70}$$

where the expectation is with respect to standard gaussian $Z$ independent of $X_0^n \sim P_{X,n}$.

Before stating theorem 4 below, we give the assumptions on the model and the functions used to define the AMP. In what follows, $C, c > 0$ are generic positive constants whose values are not exactly specified but do not depend on $n$.

**Random Vectors:** The random vector $\boldsymbol{X}^n \in \mathbb{R}^n$ used in the denoising functions, is assumed to have entries that are i.i.d. according to a sub-gaussian distribution $P_{X,n}$, in particular, $P_{X,n}$ is $\mathrm{Ber}(\rho_n)$ or Bernoulli-Rademacher, where sub-gaussian random variables are defined in lemma 14.

**The function $g_t$:** The denoising function $g_t : \mathbb{R}^2 \to \mathbb{R}$ in (69) is defined as $g_t(\boldsymbol{h}^t, \boldsymbol{X}^n) = f_t(\boldsymbol{h}^t + \sqrt{\lambda_n}\sigma_t^n \boldsymbol{X}^n)$ where $f_t$ is the conditional expectation denoiser in (9). With this definition, $g_t$ is separable and Lipschitz continuous for each $t \geq 0$, with Lipschitz constant denoted $L_g^n > 0$, that depends on $n$. By the Lipschitz property, $g_t$ are weakly differentiable and the weak derivatives, denoted by $g_t'$, are also differentiable.

**Theorem 4.** *Consider the AMP algorithm in* (69) *for* $\boldsymbol{h}^0 \in \mathbb{R}^m$ *independent of* $\boldsymbol{A}$ *under the assumptions above. Then, for any (order-2) pseudo-Lipschitz function* $\phi : \mathbb{R}^2 \to \mathbb{R}$, $\epsilon \in (0,1)$, *and* $t \geq 1$.

$$\mathbb{P}\left(\left|\frac{1}{n}\sum_{i=1}^n \phi(h_i^t, X_i^n) - \mathbb{E}\left\{\phi\left(\sqrt{\sigma_t^n}Z, X_0^n\right)\right\}\right| \geq \epsilon\right) \leq CC_t \exp\left\{\frac{-cc_t n\epsilon^2}{L_\phi^2 \widetilde{\gamma}_n^t}\right\}, \tag{71}$$

*where the expectation is with respect to standard gaussian* $Z$ *independent of* $X_0^n \sim P_{X,n}$, *the state evolution values* $\sigma_t^n$ *are defined in* (70), *the constants* $C_t, c_t$ *are defined in theorem 2, and*

$$\widetilde{\gamma}_n^{t+1} := \lambda_n^{2t}(\nu^n + \sigma_1^n)(\nu^n + \sigma_1^n + \sigma_2^n)\cdots\left(\nu^n + \sum_{i=1}^{t+1}\sigma_i^n\right)\max\{1,\hat{c}_1\}\max\{1,\hat{c}_2\}\cdots\max\{1,\hat{c}_t\}, \tag{72}$$

*where* $\hat{c}_t = \mathbb{E}[g_t'(\sqrt{\sigma_t^n}Z, X)]$ *and* $\nu$ *is the variance factor of sub-gaussian* $X^n \sim P_{X,n}$ *(for* $P_{X,n} = \mathrm{Ber}(\rho_n)$ *we have* $\nu^n \leq 1/4$; *see lemma 14).*

The proof of theorem 4 is given in section K.3. The proof relies heavily on the proof of the finite sample guarantees for various 'non-symmetric' AMP algorithms given in [63, theorem 1] and we reference this result throughout. We will use theorem 4 to prove theorem 2, but before doing so, we make a few remarks about extensions of the result and the major differences between theorem 4 and the finite sample guarantees in [63].

**Remark 1: Spectral initialization.** We assume that the AMP iteration in (69) was initialized with $\boldsymbol{h}^0 \in \mathbb{R}^n$ independent of $\boldsymbol{Z}$. As mentioned previously in section 3, for estimation with a Bernoulli-Rademacher signal prior, one needs to instead use a spectral initialization that will not be independent of the matrix $\boldsymbol{Z}$. Theoretically, as introduced in [34], one deals with this dependency by analyzing the AMP iteration using a matrix $\widetilde{\boldsymbol{Z}}$ that is an *approximate* representation of the conditional distribution of $\boldsymbol{Z}$ given the initialization and then showing that the two algorithms are close each other with high probability. We do not give the details of this rather technical argument here, and instead analyze the simpler case using an independent initialization, though the generalization is likely straightforward.

**Remark 2: Rate of the concentration.** The rate of concentration depends on $\lambda_n$, $\rho_n$, and the state evolution values, $\sigma_t^n$, through $\widetilde{\gamma}_n^t$ defined in (72). In particular, the term $\lambda_n^{2(t-1)}$ in $\widetilde{\gamma}_n^t$,

appears through the dependency of the rate on the Lipschitz constant of $g_t$, where $g_t(\boldsymbol{h}^t, \boldsymbol{X}^n) = f_t(\boldsymbol{h}^t + \sqrt{\lambda_n}\sigma_t^n\boldsymbol{X}^n)$ and $f_t$ is the conditional expectation denoiser in (9). With this definition, $L_g^n = \sqrt{\lambda_n}$. The dependence on these values was not stated explicitly in the concentration bound of [63, theorem 1] as the authors assume that the Lipschitz constant, sparsity, and state evolution terms do not change with $n$ and, thus, can be absorbed into the universal constants.

The presence of these terms in our rate comes from the inductive portion of the proof where one must show that the values $\|g_t(\boldsymbol{h}^t, \boldsymbol{X}^n)\|^2/n$ concentrate to known constants. Essentially, this step will add a term $(L_g^n)^2(\nu^n + \sigma_t^n)$ in the rate at each step of the induction. To see this, we point the reader to three facts. First, notice that the approximate distribution of $h_i^t$ is gaussian with variance $\sigma_t^n$. Second, it is easy to see that a function $[f(x)]^2$ has the same pseudo-Lipschitz constant as $f(\cdot)$ if $|f(\cdot)|$ is bounded (as $n$ grows), which is the case for $g_t$ in our setting. (More generally, the Lipschitz constant of $[f(x)]^2$ will be no more than $L_f^2$.) Finally, we highlight that pseudo-Lipschitz functions taking gaussian and sub-gaussian input concentrate as in [63, Lemma B.4] with $L^2(\nu^n + \sigma_t^n)$ in the denominator of the rate where $L$ is the associated pseudo-Lipschitz constant, $\nu^n$ is the sub-gaussian variance factor, and $\sigma_t^n$ is the gaussian variance. Indeed, we restate [63, Lemma B.4] here for clarity.

**Lemma 10.** *[63, lemma B.4] Let $Z \in \mathbb{R}^n$ be an i.i.d. standard gaussian vector and $G \in \mathbb{R}^n$ a random vector with entries $G_1, \ldots, G_n$ i.i.d. $\sim p_G$, where $p_G$ is sub-gaussian with variance factor $\nu^n$. Then, for any pseudo-Lipschitz function $f : \mathbb{R}^2 \to \mathbb{R}$ with constant $L_f^n$, non-negative values $\sigma^n$, and $0 < \epsilon \leq 1$,*

$$\mathbb{P}\Big(\Big|\frac{1}{n}\sum_{i=1}^n f(\sqrt{\sigma^n}Z_i, G_i) - \mathbb{E}[f(\sqrt{\sigma^n}Z, G)]\Big| \geq \epsilon\Big) \leq 2\exp\Big\{\frac{-\kappa n\epsilon^2}{(L_f^n)^2[\nu^n + 4(\nu^n)^2 + \sigma^n + 4(\sigma^n)^2]}\Big\}.$$

Since $0 \leq \nu^n, \sigma^n \leq 1$ we drop the squared terms $(\nu^n)^2, (\sigma^n)^2$ from the rate since $\nu^n, \sigma^n$ dominate.

**Remark 3: Denoisers** The proof of [63, theorem 1] assumes that the weak derivative of the denoiser, $g_t'$, has bounded derivative everywhere it exists. Here, $g_t(\boldsymbol{h}^t, \boldsymbol{X}^n) = f_t(\boldsymbol{h}^t + \sqrt{\lambda_n}\sigma_t^n\boldsymbol{X}^n)$ where $f_t$ is the conditional expectation denoiser in (9) and $f_t'$ is given in lemma 16. In particular, $f_t'(x) = \sqrt{\lambda_n}f_t(x)(1 - f_t(x))$, which is not bounded (in $n$) since $\lambda_n$ grows with $n$. However, we can show that $f_t'(x)$ is also Lipschitz, with constant $L_f^2 = \lambda_n$, and we use this fact directly in the proof to get around the boundedness assumption originally used in [63, theorem 1].

### K.2 Proving theorem 2

Before we get to the proof of theorem 2, we discuss how we apply the result of theorem 4 to our problem. This will lead to the concentration result in (11), which concerns convergence within pseudo-Lipschitz loss functions of the empirical distribution of $x_i^t$, the iterate of the AMP algorithm in (6), to its approximating distribution with mean and variance determined by the state evolution. Recall the following definition of a pseudo-Lipschitz function.

**Definition 1.** *For any $n, m \in \mathbb{N}_{>0}$, a function $\phi : \mathbb{R}^n \to \mathbb{R}^m$ is pseudo-Lipschitz of order $2$ if there exists a constant $L > 0$ such that $\|\phi(\boldsymbol{x}) - \phi(\boldsymbol{y})\| \leq L(1 + \|\boldsymbol{x}\| + \|\boldsymbol{y}\|)\|\boldsymbol{x} - \boldsymbol{y}\|$ for $\boldsymbol{x}, \boldsymbol{y} \in \mathbb{R}^n$.*

Now we prove (11). Recall that in our model (1),

$$\frac{1}{\sqrt{n}}\boldsymbol{W} = \frac{\sqrt{\lambda_n}}{n}\boldsymbol{X} \otimes \boldsymbol{X} + \boldsymbol{Z}, \tag{73}$$

where $\lambda_n > 0$ controls the strength of the signal and the noise is i.i.d. gaussian $Z_{ij} \sim \mathcal{N}(0, 1/n)$ for $i < j$ and symmetric, $Z_{ij} = Z_{ji}$. The AMP algorithm for recovering $\boldsymbol{X}$ from the data $\boldsymbol{W}$ is given in (6).

Notice that the AMP algorithm in (6) is similar to (69), the only difference being that the matrix $\boldsymbol{A}$ in (6) is our data matrix, as opposed to it being $\text{GOE}(n)$ as in (69). If we plug the value of $\boldsymbol{W}$ from (73) into (6), we find the following iteration: $\boldsymbol{x}^1 = \frac{\sqrt{\lambda_n}}{n}\boldsymbol{X}\langle\boldsymbol{X}, f_0(\boldsymbol{x}^0)\rangle + \boldsymbol{Z}f_0(\boldsymbol{x}^0)$, and for $t \geq 1$,

$$\boldsymbol{x}^{t+1} = \frac{\sqrt{\lambda_n}}{n}\boldsymbol{X}\langle\boldsymbol{X}, f_t(\boldsymbol{x}^t)\rangle + \boldsymbol{Z}f_t(\boldsymbol{x}^t) - \mathsf{b}_t f_{t-1}(\boldsymbol{x}^{t-1}). \tag{74}$$

Now we define a related iteration to (74) as follows. Initialize with $\boldsymbol{h}^0 = \boldsymbol{x}^0$ with denoiser $g_0(\boldsymbol{h}^0, \boldsymbol{X}) := f_0(\boldsymbol{x}^0)$ and $\boldsymbol{h}^1 = \boldsymbol{Z}g_0(\boldsymbol{h}^0, \boldsymbol{X})$. Then calculate for $t \geq 1$,

$$\boldsymbol{h}^{t+1} = \boldsymbol{Z}g_t(\boldsymbol{h}^t, \boldsymbol{X}^n) - \mathsf{c}_t g_{t-1}(\boldsymbol{h}^{t-1}, \boldsymbol{X}^n), \qquad \mathsf{c}_t = \frac{1}{n}\sum_{i=1}^{n} g_t'(h_i^t, X_i^n), \tag{75}$$

where $g_t(h, X) = f_t(h + \mu_t^n X^n)$ for $f_t(\cdot)$ the conditional expectation denoiser used in (74) and $\mu_{t+1}^n$ calculated from the state evolution in (8) for our original iteration (i.e., the algorithm in (6)). In the above, $\boldsymbol{X}^n$ is the signal in (73) and we drop the $n$ superscript in what follows. Then, the iteration in (75) takes the exact form of the symmetric AMP in (69) with state evolution given by (70). In particular, the state evolution associated with (75) is $\tau_1^n = \|g_0(\boldsymbol{h}^0, \boldsymbol{X})\|^2/n$ and for $t \geq 1$,

$$\tau_{t+1}^n = \mathbb{E}\left[\left(g_t(\sqrt{\sigma_t^n}Z, X)\right)^2\right] = \mathbb{E}\left[\left(f_t(\sqrt{\sigma_t^n}Z + \mu_t^n X)\right)^2\right]. \tag{76}$$

The above state evolution is exactly the state evolution for the AMP algorithm in (74) defined in (10). For this reason, we used the $\tau$ notation.

As the AMP algorithm in (75) takes the exact form of the symmetric AMP in (69), we can apply theorem 4. The proof idea is to use theorem 4 to give performance guarantees to the algorithm in (75) and then to argue that the algorithm in (74) is asymptotically equivalent to the algorithm in (75) so the performance guarantees hold for (74) as well.

We apply theorem 4 to (75) using the pseudo-Lipschitz function $\phi(h_i^t, X_i^n) = \psi(h_i^t + \mu_t^n X_i^n, X_i^n)$, where $\psi$ is the order 2 pseudo-Lipschitz function in (11), to find that for $t \geq 1$,

$$\mathbb{P}\left(\left|\frac{1}{n}\sum_{i=1}^{n}\psi(h_i^t + \mu_t^n X_i, X_i) - \mathbb{E}\left\{\psi\left(\sqrt{\tau_t^n}Z + \mu_t^n X, X\right)\right\}\right| \geq \epsilon\right) \leq CC_t \exp\left\{\frac{-cc_t n\epsilon^2}{L_\psi^2 \widetilde{\gamma}_n^t}\right\}. \tag{77}$$

We have used that $L_\phi = 2L_\psi(1+\mu_t^n)^2$, which is shown in lemma 17, and that $L_\phi = 2L_\psi(1+\mu_t^n)^2 \leq \kappa L_\psi$, which follows from the fact that $\mu_t^n \leq \kappa'$ in the regime of interest, as discussed, for example, in (102) in section K.4.

To show how (11) follows from (77), we use the following lemma.

**Lemma 11.** *Define* $bound_t := CC_t \exp\left\{\frac{-cc_t n\epsilon^2}{L_\psi^2 \widetilde{\gamma}_n^t}\right\}$, *for* $\widetilde{\gamma}_n^t$ *in (72). Let* $\boldsymbol{h}^t$ *be defined by the algorithm in (75) and* $\boldsymbol{x}^t$ *by (74). Then for* $t \geq 1$, *the following are true*

$$\mathbb{P}\left(\frac{1}{\sqrt{n}}\|\boldsymbol{h}^t + \mu_t^n \boldsymbol{X}\| \geq \kappa_h\right) \leq CC_t e^{\frac{-cc_t n}{\widetilde{\gamma}_n^t}}, \tag{78}$$

$$\mathbb{P}\left(\frac{1}{n}\left\|\boldsymbol{x}^t - \boldsymbol{h}^t - \mu_t^n \boldsymbol{X}\right\|^2 \geq \frac{\kappa\epsilon^2}{L_\psi^2}\right) \leq bound_t, \tag{79}$$

$$\mathbb{P}\left(\frac{1}{\sqrt{n}}\|\boldsymbol{x}^t\| \geq \kappa_x\right) \leq CC_t e^{\frac{-cc_t n}{\widetilde{\gamma}_n^t}}, \tag{80}$$

$$\mathbb{P}\left(\left|\frac{1}{n}\sum_{i=1}^{n}\psi(X_i, x_i^t) - \psi(X_i, h_i^t + \mu_t^n X_i)\right| \geq \epsilon\right) \leq bound_t, \tag{81}$$

$$\mathbb{P}\left(\left|\frac{1}{n}\sum_{i=1}^{n}\psi(X_i, x_i^t) - \mathbb{E}\left\{\psi\left(X_0, \mu_t^n X_0 + \sqrt{\tau_t^n}Z\right)\right\}\right| \geq \epsilon\right) \leq bound_t. \tag{82}$$

*In (78) and (80) both* $\kappa_h$ *and* $\kappa_x$ *are universal constants.*

The proof of lemma 11 is rather long and technical, so we include it in full detail at the end of the appendix in section K.4 and give a high level sketch here.

The basic idea behind the proof of lemma 11 is that the results in (78) follow from the fact that $h_i^t + \mu_t^n X_i \approx \sqrt{\tau_t^n}Z + \mu_t^n X$ for $X \sim P_{X,n}$ independent of $Z$ standard gaussian by theorem 4. Thus, $\frac{1}{n}\|\boldsymbol{h}^t + \mu_t^n \boldsymbol{X}\|^2$ will concentrate to $\tau_t^n + (\mu_t^n)^2 \rho_n$. Then we use concentration to imply boundedness with high probability. The result (80) follows from the same ideas since it can be shown that $x_i^t \approx \sqrt{\tau_t^n}Z + \mu_t^n X$.

Next, results (81) and (82) follow immediately from (78)–(80). To see this, first notice that (82) follows directly from the bounds in (77) and (81) using lemma 19. Next, (81) follows from results (78) – (80). This can be seen by using the following upper bound due to Cauchy-Schwarz,

$$
\left| \frac{1}{n} \sum_{i=1}^{n} \psi(X_i, x_i^t) - \psi(X_i, h_i^t + \mu_t^n X_i) \right| \leq \frac{1}{n} \sum_{i=1}^{n} \left| \psi(X_i, x_i^t) - \psi(X_i, h_i^t + \mu_t^n X_i) \right|
$$

$$
\leq \frac{L_\psi}{n} \sum_{i=1}^{n} \left( 1 + \|(X_i, x_i^t)\| + \|(X_i, h_i^t + \mu_t^n X_i)\| \right) \left| x_i^t - h_i^t - \mu_t^n X_i \right| \tag{83}
$$

$$
\leq \frac{\kappa L_\psi}{\sqrt{n}} \left\| \boldsymbol{x}^t - \boldsymbol{h}^t - \mu_t^n \boldsymbol{X} \right\| \sqrt{\left( 1 + \frac{2}{n} \|\boldsymbol{X}\|^2 + \frac{1}{n} \|\boldsymbol{x}^t\|^2 + \frac{1}{n} \|\boldsymbol{h}^t + \mu_t^n \boldsymbol{X}\|^2 \right)},
$$

and the boundedness of the term $\|\boldsymbol{X}\|^2/n$. Thus, using $\kappa_B = \sqrt{1 + 4 + \kappa_x^2 + \kappa_h^2} > 0$, a universal constant, by the above bound it follows that

$$
\mathbb{P}\left( \left| \frac{1}{n} \sum_{i=1}^{n} \psi(X_i, x_i^t) - \psi(X_i, h_i^t + \mu_t^n X_i) \right| \geq \epsilon \right)
$$

$$
\leq \mathbb{P}\left( \frac{1}{\sqrt{n}} \left\| \boldsymbol{x}^t - \boldsymbol{h}^t - \mu_t^n \boldsymbol{X} \right\| \sqrt{\left( 1 + \frac{2}{n} \|\boldsymbol{X}\|^2 + \frac{1}{n} \|\boldsymbol{x}^t\|^2 + \frac{1}{n} \|\boldsymbol{h}^t + \mu_t^n \boldsymbol{X}\|^2 \right)} \geq \frac{\kappa \epsilon}{L_\psi} \right) \tag{84}
$$

$$
\leq \mathbb{P}\left( \frac{1}{\sqrt{n}} \left\| \boldsymbol{x}^t - \boldsymbol{h}^t - \mu_t^n \boldsymbol{X} \right\| \geq \frac{\kappa \epsilon}{\kappa_B L_\psi} \right) + \mathbb{P}\left( \frac{2}{n} \|\boldsymbol{X}\|^2 \geq 2(1 + \rho_n) \right)
$$

$$
+ \mathbb{P}\left( \frac{1}{n} \|\boldsymbol{x}^t\|^2 \geq \kappa_x^2 \right) + \mathbb{P}\left( \frac{1}{n} \|\boldsymbol{h}^t + \mu_t^n \boldsymbol{X}\|^2 \geq \kappa_h^2 \right).
$$

Note, we have used $\rho_n \leq 1$ so $\frac{1}{\kappa_B^2} \left( 1 + 2(1 + \rho_n) + \kappa_x^2 + \kappa_h^2 \right) \leq 1$. Considering the result in (84), we notice that result (81) follows directly from (78)–(80), since by Hoeffding's inequality (lemma 15),

$$
\mathbb{P}\left( \frac{2}{n} \|\boldsymbol{X}\|^2 \geq 2(1 + \rho_n) \right) \leq \mathbb{P}\left( \left| \frac{1}{n} \|\boldsymbol{X}\|^2 - \rho_n \right| \geq 1 \right) \leq 2 \exp\{-2n\}. \tag{85}
$$

Thus, (81) (hence, (82),) follows easily from (78)–(80) and the main technical piece of proving lemma 11 is then proving results (78)–(80) rigorously. This is done in section K.4.

Now we show that (11) follows from lemma 11 result (82), and then we finally prove theorem 2. Notice that (11) is recovered by applying (82) with pseudo-Lipschitz function $\widetilde{\psi}(X_i, x_i^t) = \psi(X_i, f_t(x_i^t))$, as the only difference between (11) and (82) is that $x_i^t$ in (82) is replaced with $f_t(x_i^t)$ in (11). With this choice of pseudo-Lipschitz function, an $L_f^2$ term is added in the denominator of the rate of concentration, since $L_{\widetilde{\psi}} = 3 L_\psi \max\{1, L_f\}$, which is shown in lemma 17.

We note that from (14) and (72) it is easy to see that $\widetilde{\gamma}_n^t L_f^2 = \widetilde{\gamma}_n^t \lambda_n = \gamma_n^t$, noting that $\hat{c}_t = \mathbb{E}[g_t'(\sqrt{\tau_t^n} Z, X)] = \mathbb{E}[f_t'(\sqrt{\tau_t^n} Z + \mu_t^n X)] = \hat{b}_t$. Moreover, the bound on the RHS of (11) equals $\text{bound}_t$ defined in theorem 2 when $L_\psi$ is a universal constant.

Now we prove theorem 2 using (11). First, notice that theorem 2 result (12) follows directly from (11) using pseudo-Lipschitz function $\psi(X_i, f_t(x_i^t)) = (X_i - f_t(x_i^t))^2$. This function is pseudo-Lipschitz with constant $L_\psi$ by lemma 16. To see how this proves result (12) in more details, notice that

$$
\frac{1}{n} \sum_{i=1}^{n} \psi(X_i, f_t(x_i^t)) = \frac{1}{n} \sum_{i=1}^{n} (X_i - f_t(x_i^t))^2 = \frac{1}{n} \|\boldsymbol{X} - f_t(\boldsymbol{x}^t)\|^2,
$$

and

$$
\mathbb{E}\left\{ \psi\left( X_0^n, f_t(\mu_t^n X_0^n + \sqrt{\tau_t^n} Z) \right) \right\} = \mathbb{E}\left\{ \left( X_0^n - f_t\left( \mu_t^n X_0^n - \sqrt{\tau_t^n} Z \right) \right)^2 \right\}
$$

$$
= \mathbb{E}\left\{ (X_0^n)^2 \right\} + \mathbb{E}\left\{ \left[ f_t\left( \mu_t^n X_0^n - \sqrt{\tau_t^n} Z \right) \right]^2 \right\} - 2 \mathbb{E}\left\{ X_0^n f_t\left( \mu_t^n X_0^n - \sqrt{\tau_t^n} Z \right) \right\} = \rho_n - \tau_{t+1}^n,
$$

where the final uses that $\mathbb{E}\{(X_0^n)^2\} = \rho_n$ when $P_{X,n}$ is $\text{Ber}(\rho_n)$ or Bernoulli-Rademacher, the Law of Total Expectation to give $\mathbb{E}\{X_0^n f_t(\sqrt{\lambda_n} \tau_t^n X_0^n + \sqrt{\tau_t^n} Z)\} = \mathbb{E}\{[f_t(\sqrt{\lambda_n} \tau_t^n X_0^n + \sqrt{\tau_t^n} Z)]^2\}$ in the case of the conditional expectation denoiser as in (9), and the state evolution definition in (10).

Now we prove theorem 2 result (13). Now considering the concentration result in (13), notice that

$$\frac{1}{n^2}\|\boldsymbol{X}\boldsymbol{X}^T - f_t(\boldsymbol{x}^t)[f_t(\boldsymbol{x}^t)]^T\|_F^2 = \frac{1}{n^2}\|\boldsymbol{X}\|^4 + \frac{1}{n^2}\|f_t(\boldsymbol{x}^t)\|^4 - \frac{2}{n^2}\langle\boldsymbol{X}, f_t(\boldsymbol{x}^t)\rangle^2.$$

Then we will prove the following three results: for $\text{bound}_t$ defined in the theorem 2 statement,

$$\mathbb{P}\Big(\Big|\frac{1}{n^2}\|\boldsymbol{X}\|^4 - \rho_n^2\Big| \geq \epsilon\Big) \leq 2e^{-2n\epsilon^2}, \tag{86}$$

$$\mathbb{P}\Big(\Big|\frac{1}{n^2}\|f_t(\boldsymbol{x}^t)\|^4 - (\tau_t^n)^2\Big| \geq \epsilon\Big) \leq \text{bound}_t, \tag{87}$$

$$\mathbb{P}\Big(\Big|\frac{1}{n^2}\langle\boldsymbol{X}, f_t(\boldsymbol{x}^t)\rangle^2 - (\tau_t^n)^2\Big| \geq \epsilon\Big) \leq \text{bound}_t. \tag{88}$$

Then the final concentration result in (13) follows from lemma 19 as follows:

$$\mathbb{P}\Big(\Big|\frac{1}{n^2}\|\boldsymbol{X}\boldsymbol{X}^T - f_t(\boldsymbol{x}^t)[f_t(\boldsymbol{x}^t)]^T\|_F^2 - (\rho_n^2 - (\tau_{t+1}^n)^2)\Big| \geq \epsilon\Big)$$

$$= \mathbb{P}\Big(\Big|\Big(\frac{1}{n^2}\|\boldsymbol{X}\|^4 - \rho_n^2\Big) + \Big(\frac{1}{n^2}\|f_t(\boldsymbol{x}^t)\|^4 - (\tau_{t+1}^n)^2\Big) - \Big(\frac{2}{n^2}\langle\boldsymbol{X}, f_t(\boldsymbol{x}^t)\rangle^2 - 2(\tau_{t+1}^n)^2\Big)\Big| \geq \epsilon\Big)$$

$$\leq \mathbb{P}\Big(\Big|\frac{1}{n^2}\|\boldsymbol{X}\|^4 - \rho_n^2\Big| \geq \frac{\epsilon}{3}\Big) + \mathbb{P}\Big(\Big|\frac{1}{n^2}\|f_t(\boldsymbol{x}^t)\|^4 - (\tau_{t+1}^n)^2\Big| \geq \frac{\epsilon}{3}\Big)$$

$$+ \mathbb{P}\Big(\Big|\frac{2}{n^2}\langle\boldsymbol{X}, f_t(\boldsymbol{x}^t)\rangle^2 - 2(\tau_{t+1}^n)^2\Big| \geq \frac{\epsilon}{3}\Big).$$

As a final step, notice that the bounds in (86) - (88) applied to the above give the result in (13).

Now we prove (86) - (88). First we prove (86) using Heoffding's Inequality, lemma 15,

$$\mathbb{P}\Big(\Big|\frac{1}{n}\|\boldsymbol{X}\|^2 - \rho_n\Big| \geq \epsilon\Big) = \mathbb{P}\Big(\Big|\frac{1}{n}\sum_{i=1}^{n}(X_i^2 - \mathbb{E}\{X_i^2\})\Big| \geq \epsilon\Big) \leq 2e^{-2n\epsilon^2}.$$

Then the result in (86) then follows from the above by lemma 20.

Next, for (87) we apply (11) using the function $\psi(X_i, f_t(x_i^t)) = [f_t(x_i^t)]^2$, which is pseudo-Lipschitz with constant $L_\psi = 2$ by lemma 16), to find

$$\mathbb{P}\Big(\Big|\frac{1}{n}\|f_t(\boldsymbol{x}^t)\|^2 - \tau_{t+1}^n\Big| \geq \epsilon\Big) = \mathbb{P}\Big(\Big|\frac{1}{n}\sum_{i=1}^{n}[f_t(x_i^t)]^2 - \tau_{t+1}^n\Big| \geq \epsilon\Big) \leq \text{bound}_t,$$

where we have used the definition of the state evolution in (10) to give

$$\mathbb{E}\Big\{\psi\big(X_0^n, f_t\big(\mu_t^n X_0^n + \sqrt{\tau_t^n}Z\big)\big)\Big\} = \mathbb{E}\Big\{\big[f_t\big(\mu_t^n X_0^n + \sqrt{\tau_t^n}Z\big)\big]^2\Big\} = \tau_{t+1}^n.$$

Then the result in (87) follows from the above by lemma 20 and the fact that $\tau_t^n \leq \rho_n$.

Finally we prove result (88) by applying (11) using the function $\psi(X_i, f_t(x_i^t)) = X_i f_t(x_i^t)$, which is pseudo-Lipschitz with constant $L_\psi = 2$ by lemma 16, to find

$$\mathbb{P}\Big(\Big|\frac{1}{n}\langle\boldsymbol{X}, f_t(\boldsymbol{x}^t)\rangle - \tau_{t+1}^n\Big| \geq \epsilon\Big) = \mathbb{P}\Big(\Big|\frac{1}{n}\sum_{i=1}^{n}X_i f_t(x_i^t) - \tau_{t+1}^n\Big| \geq \epsilon\Big) \leq \text{bound}_t,$$

where

$$\mathbb{E}\Big\{\psi\big(X_0^n, f_t\big(\mu_t^n X_0^n + \sqrt{\tau_t^n}Z\big)\big)\Big\} = \mathbb{E}\Big\{X_0^n f_t\big(\mu_t^n X_0^n + \sqrt{\tau_t^n}Z\big)\Big\} = \tau_{t+1}^n,$$

where the final step uses the Law of Total Expectation to give $\mathbb{E}\{X_0^n f_t(\mu_t^n X_0^n + \sqrt{\tau_t^n}Z)\} = \mathbb{E}\{[f_t(\mu_t X_0^n + \sqrt{\tau_t^n}Z)]^2\}$ in the case of the conditional expectation denoiser as in (9) and the state evolution definition in (10). Then the result in (88) follows from the above by lemma 20 $\tau_t^n \leq \rho_n$.

### K.3 Proof of theorem 4

*Proof.* The proof of theorem 4 proceeds in two steps. In the first step, one studies the conditional distribution of $\boldsymbol{Z}$ given the output of the algorithm up until iteration $t$, treating $\boldsymbol{Z}$ as random and the output as deterministic. In the non-symmetric AMP studied in [63, Theorem 1], the relevant measurement matrix has i.i.d. gaussian entries and this conditional distribution was originally studied in [27]. The result for the case of i.i.d. gaussian $\boldsymbol{Z}$ is concisely stated in [63, Lemma 4.2]. For the symmetric AMP of (69) that we are interested in, the matrix $\boldsymbol{Z}$ is GOE($n$) and so this conditioning argument needs to take into account the symmetry of the matrix entries (and consequently the added dependencies). This has been studied in other works that give asymptotic characterizations of the performance of symmetric AMP, for example in [66, Lemma 3], and these results apply directly to our case since this distributional characterization is already non-asymptotic and does not change in our setting. This then allows us to characterize the conditional distribution of the iterates $\boldsymbol{h}^{t+1}$, conditional on the previous output of the algorithm. We give this result in Lemma 12 below, but before stating the lemma, we introduce some useful notation.

First, denote $\mathbf{m}^0 := g_0(\boldsymbol{h}^0, \boldsymbol{X}^n), ..., \mathbf{m}^t := g_t(\boldsymbol{h}^t, \boldsymbol{X}^n)$ where the terms $g_t(\boldsymbol{h}^t, \boldsymbol{X}^n)$ are those used in the symmetric AMP in (69). Then we define $\mathscr{S}_0$ to be the sigma-algebra generated by $\{g_0(\boldsymbol{h}^0, \boldsymbol{X}^n), \boldsymbol{X}^n\}$ and $\mathscr{S}_t$ for $t \geq 1$ to be the sigma-algebra generated by

$$\boldsymbol{h}^1, ..., \boldsymbol{h}^t, \mathbf{m}^0, ..., \mathbf{m}^t, \text{ and } \boldsymbol{X}^n.$$

Using [66, Lemma 3] to characterize the distribution of $\boldsymbol{Z}$ conditioned on the sigma algebra $\mathscr{S}_t$, we are able to specify the conditional distributions of $\boldsymbol{h}^{t+1}$ given $\mathscr{S}_t$, by observing that conditioning on $\mathscr{S}_t$ for $t \geq 1$ is equivalent to conditioning on the linear constraint[5]

$$\boldsymbol{Z}\mathbf{M}_t = \boldsymbol{Y}_t,$$

where $\mathbf{M}_t \in \mathbb{R}^{n \times t}$ and $\mathbf{H}_t \in \mathbb{R}^{n \times t}$ are the matrices

$$\mathbf{M}_t = [\mathbf{m}^0 \mid ... \mid \mathbf{m}^{t-1}] \quad \text{and} \quad \mathbf{H}_t = [\boldsymbol{h}^1 \mid ... \mid \boldsymbol{h}^t],$$

and $\boldsymbol{Y}_t \in \mathbb{R}^{n \times t}$ is the matrix $\boldsymbol{Y}_1 = \mathbf{H}_1$ and $\boldsymbol{Y}_t = \mathbf{H}_t + [\mathbf{0} \mid \mathbf{M}_{t-1}]\mathbf{C}_{\boldsymbol{t}}$ for $t \geq 2$, where $\mathbf{C}_{\boldsymbol{t}} = \text{diag}(\mathrm{c}_0, ..., \mathrm{c}_{t-1})$. Note that $[\boldsymbol{c}_1 \mid \boldsymbol{c}_2 \mid ... \mid \boldsymbol{c}_k]$ denotes a matrix with columns $\boldsymbol{c}_1, ..., \boldsymbol{c}_k$.

We use the notation $\mathbf{m}_{\|}^{t+1}$ to denote the projection of $\mathbf{m}^{t+1}$ onto the column space of $\mathbf{M}_{t+1}$. Let

$$\boldsymbol{\alpha}^{t+1} := (\alpha_0^{t+1}, \alpha_1^{t+1}, \ldots, \alpha_t^{t+1})^\mathsf{T} \in \mathbb{R}^{t+1}, \tag{89}$$

be the coefficient vectors of these projections, i.e., $\mathbf{m}_{\|}^{t+1} := \sum_{i=0}^{t} \alpha_i^t \mathbf{m}^i$, meaning $\boldsymbol{\alpha}^t = (\mathbf{M}_{t+1}^\mathsf{T} \mathbf{M}_{t+1})^{-1}\mathbf{M}_{t+1}^\mathsf{T}\mathbf{m}^{t+1}$. The projections of $\mathbf{m}^{t+1}$ onto the orthogonal complement of $\mathbf{M}_{t+1}$, is denoted by $\mathbf{m}_{\perp}^{t+1} := \mathbf{m}^{t+1} - \mathbf{m}_{\|}^{t+1}$. Lemma 13 shows that for large $n$, the entries of $\boldsymbol{\alpha}$ concentrate around constants. In what follows we show that, for $t \geq 0$, the vector $\boldsymbol{\alpha}^{t+1} \in \mathbb{R}^{t+1}$ in (89) concentrates to the vector

$$\hat{\boldsymbol{\alpha}}^{t+1} := \left[0, \ldots, 0, \frac{\sigma_{t+2}^n}{\sigma_{t+1}^n}\right]^\mathsf{T} \in \mathbb{R}^{t+1}, \tag{90}$$

for the state evolution values given in (70). Similarly, Lemma 13 will show that for large $n$, the norm $\|\mathbf{m}_{\perp}^{t-1}\|^2/n$ concentrates to a constant $\sigma_t^{\perp}$, defined as $\sigma_1^{\perp} = \sigma_1^n$, and for $t \geq 2$,

$$\sigma_t^{\perp} := \sigma_t^n \left(1 - \frac{\sigma_t^n}{\sigma_{t-1}^n}\right). \tag{91}$$

With the above notation, we find the following result for the symmetric AMP in (69).

**Lemma 12** (Conditional Distribution Lemma). *For the vectors $\boldsymbol{h}^{t+1}$ defined in (69), the following hold for $t \geq 1$, provided $n > t$ and $\mathbf{M}_t^\mathsf{T}\mathbf{M}_t$ has full column rank.*

$$\boldsymbol{h}^1|_{\mathscr{S}_0} \overset{d}{=} \sqrt{\sigma_1^n}\,\boldsymbol{U}_0 + \boldsymbol{\Delta}_0, \qquad and \qquad \boldsymbol{h}^{t+1}|_{\mathscr{S}_t} \overset{d}{=} \hat{\alpha}_t^{t+1}\,\boldsymbol{h}^t + \sqrt{\sigma_{t+1}^{\perp}}\,\boldsymbol{U}_t + \boldsymbol{\Delta}_t, \tag{92}$$

where $\boldsymbol{U}_0, \boldsymbol{U}_t \in \mathbb{R}^n$ are random vectors with elements that are marginally standard gaussian random variables that are independent of the corresponding conditioning sigma-algebras. The terms $\hat{\alpha}_i^t$ for $i \in \{0, 1, ..., t\}$ are defined in (90) and the terms $\sigma_t^\perp$ in (91). The deviation terms are $\boldsymbol{\Delta}_t = \boldsymbol{0}$ and for $t > 0$,

$$\boldsymbol{\Delta}_t = \sum_{r=1}^{t} (\alpha_r^{t+1} - \hat{\alpha}_r^{t+1}) \boldsymbol{h}^r + \left[ \left( \frac{\|\mathbf{m}_\perp^t\|}{\sqrt{n}} - \sqrt{\sigma_{t+1}^\perp} \right) \mathsf{I} - \frac{\|\mathbf{m}_\perp^t\|}{\sqrt{n}} \mathsf{P}_{\mathbf{M}_t}^\| \right] \boldsymbol{U}_t$$

$$+ \mathbf{M}_{t-1} (\mathbf{M}_{t-1}^\intercal \mathbf{M}_{t-1})^{-1} \left[ \mathbf{H}_{t-1}^\intercal \mathbf{m}_\perp^t - \mathbf{M}_{t-1}^* \left( c_t \mathbf{m}^{t-1} - \sum_{i=1}^{t-1} c_i \alpha_i^t \mathbf{m}^{i-1} \right) \right]. \tag{93}$$

The second step of the proof is inductive on the iteration $t$, showing that if the result in (71) holds up to iteration $t-1$ then it will hold at iteration $t$ as well. This is done by showing that the standardized $\ell_2$ norms of the terms in the AMP algorithm in (69), like $\frac{1}{n}\|\boldsymbol{h}^t\|^2$ and $\frac{1}{n}\|\mathbf{m}^t\|^2$, concentrate on deterministic values predicted by the state evolution. This is done by relating these iteration $t$ values to the iteration $t-1$ values through the iteration in (71) and appealing to the conditional distributions from Lemma 12 and your inductive hypothesis. While the proof for the symmetric AMP is largely similar to that for the non-symmetric version, there are additional challenges due to the dependencies created by the symmetry of the GOE($n$) matrix. The details of the inductive proof are quite technical and long and are therefore not included here but we state the result for the symmetric AMP in (69).

For $t \geq 0$, let $\kappa_{-1} = K_{-1} = 1$, and

$$K_t = C(t+1)^5 K_{t-1}, \quad \kappa_t = \frac{\kappa_{t-1}}{c(t+1)^{11}}, \tag{94}$$

where $C, c > 0$ are universal constants (not depending on $t$, $n$, or e). To keep the notation compact, we use $K, \kappa, \kappa'$ to denote generic positive universal constants whose values may change through the lemma statement.

The result of theorem 4 follows from lemma 13 result (97) below.

**Lemma 13.** *The following statements hold for* $1 \leq t < T^*$ *and* $\epsilon \in (0, 1)$. *Define*

$$\gamma_n^{t+1} := (\nu^n + \sigma_1^n)(\nu^n + \sigma_1^n + \sigma_2^n) \cdots (\nu^n + \sum_{i=1}^{t+1} \sigma_i^n) \times \max\{1, \hat{c}_1\} \max\{1, \hat{c}_2\} \cdots \max\{1, \hat{c}_t\}, \tag{95}$$

*where* $\nu$ *is the variance factor of sub-gaussian* $\boldsymbol{X}^n$ *(for* $P_{X,n} = \text{Ber}(\rho_n)$ *we have* $\nu^n \leq 1/4$; *see lemma 14).*

1.

$$\mathbb{P}\left( \frac{1}{n} \|\boldsymbol{\Delta}_t\|^2 \geq \epsilon \right) \leq K(t+1)^2 K_{t-1} \exp\left\{ -\frac{\kappa \kappa_{t-1} n \epsilon}{(t+1)^4 L_g^{4t} \gamma_n^t \max\{1, \hat{c}_t\}} \right\}. \tag{96}$$

2. *Denote* $\mathbb{E}_\phi := \mathbb{E}\, \phi_h(\sqrt{\sigma_1^n}\tilde{Z}_1, \ldots, \sqrt{\sigma_{t+1}^n}\tilde{Z}_{t+1}, X^n)$. *Then for pseudo-Lipschitz function* $\phi : \mathbb{R}^{t+1} \to \mathbb{R}$ *with constant* $L_\phi$ *we have that*

$$\mathbb{P}\left( \left| \frac{1}{n} \sum_{i=1}^{n} \phi(h_i^1, \ldots, h_i^{t+1}, X_i^n) - \mathbb{E}_\phi \right| \geq \epsilon \right) \leq K(t+1)^3 K_{t-1} \exp\left\{ -\frac{\kappa \kappa_{t-1} n \epsilon^2}{(t+1)^7 L_\phi^2 L_g^{4t} \gamma_n^{t+1}} \right\}. \tag{97}$$

*The random variables* $\tilde{Z}_0, \ldots, \tilde{Z}_t$ *are jointly gaussian with zero mean and covariance given by* $\mathbb{E}[\tilde{Z}_r \tilde{Z}_t] = \sqrt{\sigma_t^n/\sigma_r^n}$ *for* $r < t$, *and are independent of* $\boldsymbol{X}^n \sim p_{X,n}$.

*Denote* $\mathbb{E}_{\phi_{Lip}} := \mathbb{E}\, \phi_{Lip}(\sqrt{\sigma_1^n}\tilde{Z}_1, \ldots, \sqrt{\sigma_{t+1}^n}\tilde{Z}_{t+1}, X^n)$. *Then for Lipschitz function* $\phi_{Lip} : \mathbb{R}^{t+1} \to \mathbb{R}$ *with constant* $L_{\phi_{Lip}}$ *we have that*

$$\mathbb{P}\left( \left| \frac{1}{n} \sum_{i=1}^{n} m_i^0 \phi_{Lip}(h_i^1, \ldots, h_i^{t+1}, X_i^n) - \tilde{\sigma} \mathbb{E}_{\phi_{Lip}} \right| \geq \epsilon \right)$$

$$\leq K(t+1)^3 K_{t-1} \exp\left\{ -\frac{\kappa \kappa_{t-1} n \epsilon^2}{(t+1)^7 L_\phi^2 L_g^{4t} \gamma_n^{t+1}} \right\}. \tag{98}$$

*The random variables $\tilde{Z}_0, \ldots, \tilde{Z}_t$ are as above.*

3. *Let $L_g > 0$ be the pseudo-Lipschitz constant for the denoiser functions $\{g_t\}_{t \geq 0}$ and let $X_n \doteq c$ be shorthand for*

$$\mathbb{P}(|X_n - c| \geq \epsilon) \leq K(t+1)^3 K_{t-1} \exp\left\{ -\frac{\kappa \kappa_{t-1} n \epsilon^2}{(t+1)^7 L_g^{4t+2} \gamma_n^{t+1}} \right\}.$$

*For all $0 \leq r \leq t$,*

$$\mathbb{P}\left(\left|\frac{1}{n}(\boldsymbol{h}^{r+1})^* \boldsymbol{h}^{t+1} - \sigma_{t+1}^n\right| \geq \epsilon\right) \leq K(t+1)^3 K_{t-1} \exp\left\{ -\frac{\kappa \kappa_{t-1} n \epsilon^2}{(t+1)^7 L_g^{4t} \gamma_n^{t+1}} \right\}.$$

$$\frac{1}{n}(\boldsymbol{m}^0)^* \boldsymbol{m}^{t+1} \doteq \tilde{\sigma}^n \mathbb{E}[g_{t+1}(\sqrt{\sigma_{t+1}^n}\tilde{Z}_{t+1}, X^n)], \quad \frac{1}{n}(\boldsymbol{m}^{r+1})^* \boldsymbol{m}^{t+1} \doteq \sigma_{t+2}^n.$$

$$\mathbb{P}\left(\left|c_{t+1} - \hat{c}_{t+1}\right| \geq \epsilon\right) \leq K(t+1)^3 K_{t-1} \exp\left\{ -\frac{\kappa \kappa_{t-1} n \epsilon^2}{(t+1)^7 L_g^{4(t+1)} \gamma_n^{t+1}} \right\}.$$

$$\frac{1}{n}(\boldsymbol{h}^{t+1})^* \boldsymbol{m}^{r+1} \doteq \hat{c}_{r+1} \sigma_{t+2}^n, \qquad \frac{1}{n}(\boldsymbol{h}^{r+1})^* \boldsymbol{m}^{t+1} \doteq \hat{c}_{t+1} \sigma_{t+2}^n.$$

4.

$$\mathbb{P}\left(\frac{1}{n}\mathbf{M}_{t+1}^* \mathbf{M}_{t+1} \text{ is singular}\right) \leq (t+1) K_{t-1} \exp\left\{ -\frac{\kappa_{t-1}\kappa n}{(t+1)^7 L_g^{4t+2} \gamma_n^{t+1}} \right\}. \tag{99}$$

*For $\hat{\alpha}^{t+1}$ defined in (90), when the inverse of $\frac{1}{n}\mathbf{M}_{t+1}^* \mathbf{M}_{t+1}$ exists, for $1 \leq i, j \leq t+1$,*

$$\mathbb{P}\left(\left|\left[\left(\frac{1}{n}\mathbf{M}_{t+1}^* \mathbf{M}_{t+1}\right)^{-1} - (\mathbf{C}^{t+1})^{-1}\right]_{i,j}\right| \geq \epsilon\right) \leq K K_{t-1} \exp\left\{ -\frac{\kappa \kappa_{t-1} n \epsilon^2}{L_g^{4t+2} \gamma_n^{t+1}} \right\},$$

$$\mathbb{P}\left(|\alpha_{i-1}^{t+1} - \hat{\alpha}_{i-1}^{t+1}| \geq \epsilon\right) \leq K(t+1)^4 K_{t-1} \exp\left\{ -\frac{\kappa \kappa_{t-1} n \epsilon^2}{(t+1)^9 L_g^{4(t+1)} \gamma_n^{t+1}} \right\}. \tag{100}$$

*In the above, the matrix $\mathbf{C}^{t+1} \in \mathbb{R}^{(t+1) \times (t+1)}$ has elements $[\mathbf{C}^{t+1}]_{i,j} = \sigma_{\max\{i,j\}}^n$ for $1 \leq i, j \leq t+1$.*

5. *With $\sigma_t^\perp$ defined in (91),*

$$\mathbb{P}\left(\left|\frac{1}{n}\|\boldsymbol{m}_\perp^{t+1}\|^2 - \sigma_{t+2}^\perp\right| \geq \epsilon\right) \leq K(t+1)^5 K_{t-1} \exp\left\{ -\frac{\kappa \kappa_{t-1} n \epsilon^2}{(t+1)^{11} L_g^{4(t+1)} \gamma_n^{t+1}} \right\}. \tag{101}$$

$\square$

## K.4   Proof of lemma 11

*Proof.* To begin with, we prove result (78) then we prove the other results, (79)–(82), inductively.

Before we do so we establish upper and lower bounds on $\tau_n^t$ defined in (10). Notice that for the Bernoulli case,

$$\tau_{t+1}^n = \mathbb{E}\left[ \frac{\rho_n^2}{\rho_n + (1-\rho_n)\exp\{-\frac{1}{2}\lambda_n \tau_t^n - \sqrt{\lambda_n \tau_t^n}Z\}} \right],$$

where $Z \sim \mathcal{N}(0, 1)$, as shown in appendix G result (55). Therefore, trivially $\tau_{t+1}^n \leq \rho_n$. We also wish to establish a lower bound. First, by Jensen's Inequality applied to the convex function $f(x) = 1/x$ on $x \in (0, \infty)$, we have that

$$\tau_{t+1}^n \geq \frac{\rho_n^2}{\rho_n + (1-\rho_n)\mathbb{E}\left[\exp\{-\frac{1}{2}\lambda_n \tau_t^n - \sqrt{\lambda_n \tau_t^n}Z\}\right]} \overset{(a)}{=} \rho_n^2,$$

where step $(a)$ uses that $\mathbb{E}[\exp\{-\frac{1}{2}\lambda_n \tau_t^n - \sqrt{\lambda_n \tau_t^n}Z\}] = 1$ since $\mathbb{E}[\exp\{-tZ\}] = \exp\{\frac{1}{2}t^2\}$. Thus, $\rho_n^2 \leq \tau_{t+1}^n \leq \rho_n$ and, in the regime of interest where $\lambda_n = \kappa \rho_n^{-2}$, using that $\mu_t^n = \sqrt{\lambda_n}\tau_t^n$ by (10), we find $(\mu_t^n)^2 = \lambda_n(\tau_t^n)^2 = \kappa \rho_n^{-2}(\tau_t^n)^2$ and therefore

$$\kappa' \rho_n^2 \leq (\mu_t^n)^2 = \kappa \rho_n^{-2}(\tau_t^n)^2 \leq \kappa. \tag{102}$$

**Result (78).** We first show that (78) follows immediately from theorem 4. Applying theorem 4 with pseudo-Lipschitz function $\phi(h_i^t, X_i^n) = (h_i^t + \mu_t^n X_i^n)^2$, having constant $L_\phi = 2\max\{1, (\mu_t^n)^2\}$, as is shown in lemma 17,

$$\mathbb{P}\Big(\Big|\frac{1}{n}\|\boldsymbol{h}^t + \mu_t^n \boldsymbol{X}^n\|^2 - (\mu_t^n)^2 \rho_n - \tau_t^n\Big| \geq \epsilon\Big) \leq CC_t \exp\Big\{\frac{-cc_t n \epsilon^2}{\max\{1, (\mu_t^n)^4\}\widetilde{\gamma}_n^t}\Big\} \leq \text{bound}_t, \quad (103)$$

where the final inequality follows since $(\mu_t^n)^4 \leq \kappa$ as discussed above in (102). We have also used that $\mathbb{E}[\phi(\sqrt{\tau_t^n}Z, X_0^n)] = \mathbb{E}[(\sqrt{\tau_t^n}Z + \mu_t^n X_0^n)^2] = \tau_t^n + (\mu_t^n)^2 \rho_n$. Then, since $(\mu_t^n)^2 \rho_n \leq 1$, choosing $\kappa_h^2 = 3 > 1 + (\mu_t^n)^2 \rho_n + \tau_t^n$, we recover the first result in (78) with

$$\mathbb{P}\Big(\frac{1}{\sqrt{n}}\big\|\boldsymbol{h}^t + \mu_t^n \boldsymbol{X}\big\| \geq \kappa_h\Big) \leq \mathbb{P}\Big(\Big|\frac{1}{n}\big\|\boldsymbol{h}^t + \mu_t^n \boldsymbol{X}\big\|^2 - (\mu_t^n)^2 \rho_n - \tau_t^n\Big| \geq 1\Big).$$

**Other results (79)–(82).** The proof is inductive on the iteration $t$. We first show the initialization case $t = 1$. Consider (79), then using the definitions of $\boldsymbol{x}^1 = \frac{\sqrt{\lambda_n}}{n}\boldsymbol{X}\langle \boldsymbol{X}, f_0(\boldsymbol{x}^0)\rangle + \boldsymbol{Z}f_0(\boldsymbol{x}^0)$ from (74) and $\boldsymbol{h}^1 = \boldsymbol{Z}g_0(\boldsymbol{h}^0, \boldsymbol{X})$ from (75) along with the fact that $f_0(\boldsymbol{x}^0) = g_0(\boldsymbol{h}^0, \boldsymbol{X})$,

$$\boldsymbol{x}^1 - \boldsymbol{h}^1 - \mu_1^n \boldsymbol{X} = \boldsymbol{X}\Big(\frac{\sqrt{\lambda_n}}{n}\langle \boldsymbol{X}, f_0(\boldsymbol{x}^0)\rangle - \mu_1^n\Big) = 0,$$

where the final inequality follows since $\mu_1^n = \sqrt{\lambda_n}\langle f_0(\boldsymbol{x}^0), \boldsymbol{X}\rangle/n$ by (7). Next for result (80), first notice that by the Triangle Inequality, $\|\boldsymbol{x}^1\| \leq \|\boldsymbol{x}^1 - \boldsymbol{h}^1 - \mu_1^n \boldsymbol{X}\| + \|\boldsymbol{h}^1 + \mu_1^n \boldsymbol{X}\|$. Then let $\kappa_x = 2\kappa_h + 2\kappa$ and therefore, by lemma 19,

$$\mathbb{P}\Big(\frac{1}{\sqrt{n}}\|\boldsymbol{x}^1\| \geq \kappa_x\Big) \leq \mathbb{P}\Big(\frac{1}{\sqrt{n}}\|\boldsymbol{x}^1 - \boldsymbol{h}^1 - \mu_1^n \boldsymbol{X}\| + \frac{1}{\sqrt{n}}\|\boldsymbol{h}^1 + \mu_1^n \boldsymbol{X}\| \geq 2\kappa_h + 2\kappa\Big)$$

$$\leq \mathbb{P}\Big(\frac{1}{\sqrt{n}}\|\boldsymbol{x}^1 - \boldsymbol{h}^1 - \mu_1^n \boldsymbol{X}\| \geq \kappa\Big) + \mathbb{P}\Big(\frac{1}{\sqrt{n}}\|\boldsymbol{h}^1 + \mu_1^n \boldsymbol{X}\| \geq \kappa_h\Big).$$

Then the upper bound follows by (78) and (79).

Next, notice that with the bound on $\mathbb{P}(\frac{1}{\sqrt{n}}\|\boldsymbol{x}^1\| \geq \kappa_x)$ established above, one can prove the $t = 1$ case for (81) and (82), as justified in the work in (83) – (84) along with results (78) and (79).

Now assume that all results (80)–(82) hold up until iteration $t - 1$ and we prove the results for iteration $t$. As justified in the work in (83) – (84), the results (81) and (82) follow immediately from (78) – (80) so we only aim to prove (79) and (80) here (as (78) was demonstrated above in (103)). We begin by proving (79) which we will then use to prove (80).

**Result (79).** Next we consider result (79). Using the definitions of $\boldsymbol{x}^{t+1}$ and $\boldsymbol{h}^{t+1}$ from (74) and (75) along with Cauchy-Schwarz inequality, we have that

$$\frac{1}{n}\sum_{i=1}^n \Big|x_i^t - h_i^t - \mu_t^n X_i\Big|^2$$

$$\leq \frac{3}{n}\Big|\frac{\sqrt{\lambda_n}}{n}\langle \boldsymbol{X}, f_{t-1}(\boldsymbol{x}^{t-1})\rangle - \mu_t^n\Big|^2 \sum_{i=1}^n X_i^2 + \frac{3}{n}\sum_{i=1}^n \Big|[\boldsymbol{Z}f_{t-1}(\boldsymbol{x}^{t-1})]_i - [\boldsymbol{Z}g_{t-1}(\boldsymbol{h}^{t-1}, \boldsymbol{X})]_i\Big|^2$$

$$+ \frac{3}{n}\sum_{i=1}^n \Big|\mathsf{b}_{t-1}f_{t-2}(x_i^{t-2}) - \mathsf{c}_{t-1}g_{t-2}(h_i^{t-2}, X_i)\Big|^2$$

$$= 3\Big|\frac{\sqrt{\lambda_n}}{n}\langle \boldsymbol{X}, f_{t-1}(\boldsymbol{x}^{t-1})\rangle - \mu_t^n\Big|^2 \frac{1}{n}\sum_{i=1}^n X_i^2 + \frac{3}{n}\Big\|\boldsymbol{Z}\big(f_{t-1}(\boldsymbol{x}^{t-1}) - g_{t-1}(\boldsymbol{h}^{t-1}, \boldsymbol{X})\big)\Big\|^2$$

$$+ \frac{3}{n}\Big\|\mathsf{b}_{t-1}f_{t-2}(\boldsymbol{x}^{t-2}) - \mathsf{c}_{t-1}g_{t-2}(\boldsymbol{h}^{t-2}, \boldsymbol{X})\Big\|^2.$$

$$(104)$$

Now we use the upper bounds in (104) along with lemma 19 to give the following upper bound on the probability on the LHS of (79):

$$\mathbb{P}\Big(\frac{1}{n}\big\|\boldsymbol{x}^t - \boldsymbol{h}^t - \mu_t^n \boldsymbol{X}\big\|^2 \geq \frac{\kappa\epsilon^2}{L_\psi^2}\Big) \leq \mathbb{P}\Big(\Big|\frac{\sqrt{\lambda_n}}{n}\langle \boldsymbol{X}, f_{t-1}(\boldsymbol{x}^{t-1})\rangle - \mu_t^n\Big|^2 \frac{1}{n}\sum_{i=1}^n X_i^2 \geq \frac{\kappa\epsilon}{L_\psi^2}\Big)$$

$$+ \mathbb{P}\Big(\frac{1}{n}\big\|\boldsymbol{Z}\big(f_{t-1}(\boldsymbol{x}^{t-1}) - g_{t-1}(\boldsymbol{h}^{t-1}, \boldsymbol{X}))\big\|^2 \geq \frac{\kappa\epsilon}{L_\psi^2}\Big)$$

$$+ \mathbb{P}\Big(\frac{1}{n}\big\|\mathsf{b}_{t-1}f_{t-2}(\boldsymbol{x}^{t-2}) - \mathsf{c}_{t-1}g_{t-2}(\boldsymbol{h}^{t-2}, \boldsymbol{X})\big\|^2 \geq \frac{\kappa\epsilon}{L_\psi^2}\Big).$$
(105)

We label the three terms in the above $T_1, T_2, T_3$ and provide an upper bound for each.

First consider term $T_1$ of (105), and recall that $\mu_{t-1}^n = \sqrt{\lambda_n}\tau_{t-1}^n$. Thus, we have the upper bound

$$T_1 \leq \mathbb{P}\Big(\Big|\frac{1}{n}\langle \boldsymbol{X}, f_{t-1}(\boldsymbol{x}^{t-1})\rangle - \tau_{t-1}^n\Big| \geq \frac{\kappa\sqrt{\epsilon}}{\sqrt{(1+\rho_n)\lambda_n}L_\psi}\Big) + \mathbb{P}\Big(\frac{1}{n}\sum_{i=1}^n X_i^2 \geq (1+\rho_n)\Big). \quad (106)$$

Notice that we can upper bound the second term in (106) with using $2e^{-2n}$ Hoeffding's inequality (lemma 15) as in (85). We can upper bound the first term in (106) using the induction hypothesis for result (79) for the pseudo-Lipschitz function $\widetilde{\psi}(a, b) = af_{t-1}(b)$ with constant $L_{\widetilde{\psi}} = L_f$. Thus,

$$\mathbb{P}\Big(\Big|\frac{1}{n}\langle \boldsymbol{X}, f_{t-1}(\boldsymbol{x}^{t-1})\rangle - \tau_{t-1}^n\Big| \geq \frac{\kappa\sqrt{\epsilon}}{\sqrt{(1+\rho_n)\lambda_n}L_\psi}\Big) \leq CC_{t-1}\exp\Big\{\frac{-cc_{t-1}n\epsilon^2}{L_\psi^2 L_f^2 \lambda_n(1+\rho_n)\widetilde{\gamma}_n^{t-1}}\Big\}.$$

Finally we notice that the desired result follows since $L_f = \sqrt{\lambda_n}$, proved in lemma 18, and $\lambda_n^2(1-\rho_n)\widetilde{\gamma}_n^{t-1} \leq \widetilde{\gamma}_n^t$ using the definition of $\widetilde{\gamma}_n^t$ in (72).

Now consider term $T_2$ of (105). First notice that, conditioned on event $\mathcal{F}_{t-1}$,

$$\frac{1}{\sqrt{n}}\big\|\boldsymbol{Z}\big(f_{t-1}(\boldsymbol{x}^{t-1}) - g_{t-1}(\boldsymbol{h}^{t-1}, \boldsymbol{X}))\big\| \leq \frac{1}{\sqrt{n}}\|\boldsymbol{Z}\|_{op}\big\|f_{t-1}(\boldsymbol{x}^{t-1}) - g_{t-1}(\boldsymbol{h}^{t-1}, \boldsymbol{X})\big\|$$

$$\overset{(a)}{\leq} \frac{1}{\sqrt{n}}\|\boldsymbol{Z}\|_{op}\big\|f_{t-1}(\boldsymbol{x}^{t-1}) - f_{t-1}(\boldsymbol{h}^{t-1} + \mu_{t-1}^n\boldsymbol{X})\big\| \overset{(b)}{\leq} \|\boldsymbol{Z}\|_{op}\sqrt{\frac{\lambda_n}{n}}\big\|\boldsymbol{x}^{t-1} - \boldsymbol{h}^{t-1} - \mu_{t-1}^n\boldsymbol{X}\big\|,$$

where step $(a)$ uses that $g_{t-1}(\boldsymbol{h}^{t-1}, \boldsymbol{X}) = f_{t-1}(\boldsymbol{h}^{t-1} + \mu_{t-1}^n\boldsymbol{X})$ and step $(b)$ uses the Lipschitz property of $f_{t-1}$. Therefore,

$$\mathbb{P}\Big(\frac{1}{\sqrt{n}}\big\|\boldsymbol{Z}\big(f_{t-1}(\boldsymbol{x}^{t-1}) - g_{t-1}(\boldsymbol{h}^{t-1}, \boldsymbol{X}))\big\| \geq \frac{\kappa\sqrt{\epsilon}}{L_\psi}\Big)$$

$$\leq \mathbb{P}\Big(\|\boldsymbol{Z}\|_{op}\sqrt{\frac{\lambda_n}{n}}\big\|\boldsymbol{x}^{t-1} - \boldsymbol{h}^{t-1} - \mu_{t-1}^n\boldsymbol{X}\big\| \geq \frac{\kappa\epsilon}{L_\psi}\Big)$$

$$\leq \mathbb{P}\Big(\frac{1}{\sqrt{n}}\big\|\boldsymbol{x}^{t-1} - \boldsymbol{h}^{t-1} - \mu_{t-1}^n\boldsymbol{X}\big\| \geq \frac{\kappa\epsilon}{L_\psi\sqrt{\lambda_n}}\Big) + \mathbb{P}\Big(\|\boldsymbol{Z}\|_{op} \geq \kappa\Big)$$

$$\leq CC_{t-1}\exp\Big\{\frac{-cc_{t-1}n\epsilon^2}{L_\psi^2\widetilde{\gamma}_n^t}\Big\} + C\exp\{-cn\},$$

where the final inequality follows from the inductive hypothesis for (79) and standard results about tail bounds for operator norms of GOE matrices. In particular, we have used the inductive hypothesis to find

$$\mathbb{P}\Big(\frac{1}{n}\big\|\boldsymbol{x}^{t-1} - \boldsymbol{h}^{t-1} + \mu_{t-1}^n\boldsymbol{X}\big\|^2 \geq \frac{\kappa\epsilon^2}{L_\psi^2\lambda_n}\Big) \leq CC_{t-1}\exp\Big\{\frac{-cc_{t-1}n\epsilon^2}{L_\psi^2\lambda_n\widetilde{\gamma}_n^{t-1}}\Big\}$$

$$\leq CC_{t-1}\exp\Big\{\frac{-cc_{t-1}n\epsilon^2}{L_\psi^2\widetilde{\gamma}_n^t}\Big\},$$

where the final inequality follows since $\lambda_n\widetilde{\gamma}_n^{t-1} \leq \widetilde{\gamma}_n^t$ using the definition of $\widetilde{\gamma}_n^t$ in (72).

Finally, consider term $T_3$ of (105). We first give an upper bound using the definition of $g_t$ and the Lipschitz property of $f_t$ with $L_f = \sqrt{\lambda_n}$ as follows:

$$
\begin{aligned}
\left\| \mathsf{b}_{t-1} f_{t-2}(\boldsymbol{x}^{t-2}) - \mathsf{c}_{t-1} g_{t-2}(\boldsymbol{h}^{t-2}, \boldsymbol{X}) \right\| &= \left\| \mathsf{b}_{t-1} f_{t-2}(\boldsymbol{x}^{t-2}) - \mathsf{c}_{t-1} f_{t-2}(\boldsymbol{h}^{t-2} + \mu_{t-2}^n \boldsymbol{X}) \right\| \\
&\leq |\mathsf{b}_{t-1}| \left\| f_{t-2}(\boldsymbol{x}^{t-2}) - f_{t-2}(\boldsymbol{h}^{t-2} + \mu_{t-2}^n \boldsymbol{X}) \right\| + |\mathsf{b}_{t-1} - \mathsf{c}_{t-1}| \left\| f_{t-2}(\boldsymbol{h}^{t-2} + \mu_{t-2}^n \boldsymbol{X}) \right\| \\
&\leq \lambda_n \left\| \boldsymbol{x}^{t-2} - \boldsymbol{h}^{t-2} - \mu_{t-2}^n \boldsymbol{X} \right\| + |\mathsf{b}_{t-1} - \mathsf{c}_{t-1}| \sqrt{n}.
\end{aligned}
$$
(107)

In the final step we use the lemma 18 results

$$
|\mathsf{b}_{t-1}| \leq \frac{1}{n} \sum_{i=1}^n \left| f'_{t-1}(x_i^{t-1}) \right| \leq \sqrt{\lambda_n}, \qquad \text{and} \qquad \left\| f_{t-2}(\boldsymbol{h}^{t-2} + \mu_{t-2}^n \boldsymbol{X}) \right\|^2 \leq n.
$$

We investigate the term $|\mathsf{b}_{t-1} - \mathsf{c}_{t-1}|$ and recall from their definitions in (6) and (75),

$$
\begin{aligned}
|\mathsf{b}_{t-1} - \mathsf{c}_{t-1}| &\leq \frac{1}{n} \sum_{i=1}^n \left| f'_{t-1}(x_i^{t-1}) - g'_{t-1}(h_i^{t-1}, X_i) \right| = \frac{1}{n} \sum_{i=1}^n \left| f'_{t-1}(x_i^{t-1}) - f'_{t-1}(h_i^{t-1} - \mu_{t-1}^n X_i) \right| \\
&\overset{(a)}{=} \frac{\sqrt{\lambda_n}}{n} \sum_{i=1}^n \left| f_{t-1}(x_i^{t-1})(1 - f_{t-1}(x_i^{t-1})) - f_{t-1}(h_i^{t-1} - \mu_{t-1}^n X_i)(1 - f_{t-1}(h_i^{t-1} - \mu_{t-1}^n X_i)) \right| \\
&\overset{(b)}{\leq} \frac{\sqrt{\lambda_n}}{n} \sum_{i=1}^n \left| f_{t-1}(x_i^{t-1}) - f_{t-1}(h_i^{t-1} - \mu_{t-1}^n X_i) \right| \overset{(c)}{\leq} \frac{\lambda_n}{\sqrt{n}} \left\| \boldsymbol{x}^{t-1} - \boldsymbol{h}^{t-1} - \mu_{t-1}^n \boldsymbol{X} \right\|.
\end{aligned}
$$
(108)

In the above, step $(a)$ uses lemma 18 for computing the derivative $f_t$, step $(b)$ uses the bound

$$
\left| f(a)(1 - f(a)) - f(b)(1 - f(b)) \right| \leq \left| f(a) - f(b) \right| + \left| [f(a)]^2 - [f(b)]^2 \right| \leq \kappa \left| f(a) - f(b) \right|,
$$

for $0 \leq f(a) \leq 1$ for all $a \in \mathbb{R}$, and step $(c)$ uses the Lipschitz property of $f_t$, namely $L_f = \sqrt{\lambda_n}$, and Cauchy Schwarz to give $\sum_{i=1}^n |a_i| \leq \sqrt{n} \|\mathbf{a}\|$. Plugging the bound in (108) into (107),

$$
\begin{aligned}
\left\| \mathsf{b}_{t-1} f_{t-2}(\boldsymbol{x}^{t-2}) - \mathsf{c}_{t-1} g_{t-2}(\boldsymbol{h}^{t-2}, \boldsymbol{X}) \right\| \\
\leq \lambda_n \left\| \boldsymbol{x}^{t-2} - \boldsymbol{h}^{t-2} - \mu_{t-2}^n \boldsymbol{X} \right\| + \lambda_n \left\| \boldsymbol{x}^{t-1} - \boldsymbol{h}^{t-1} - \mu_{t-1}^n \boldsymbol{X} \right\|.
\end{aligned}
$$

Now we have from lemma 19 that

$$
T_3 \leq 2 \mathbb{P} \left( \frac{1}{n} \left\| \boldsymbol{x}^{t-1} - \boldsymbol{h}^{t-1} - \mu_{t-1}^n \boldsymbol{X} \right\|^2 \geq \frac{\kappa \epsilon}{L_\psi^2 \lambda_n^2} \right),
$$

and the final bound follows from the inductive hypothesis for (81) using that $\lambda_n^2 \widetilde{\gamma}_n^{t-1} \leq \widetilde{\gamma}_n^t$ using the definition of $\widetilde{\gamma}_n^t$ in (72).

**Result** (80). To complete the proof, we consider result (80). First notice that by the Triangle Inequality, $\|\boldsymbol{x}^t\| \leq \|\boldsymbol{x}^t - \boldsymbol{h}^t - \mu_t^n \boldsymbol{X}\| + \|\boldsymbol{h}^t + \mu_t^n \boldsymbol{X}\|$. Then let $\kappa_x = 2\kappa_h + \frac{2\kappa}{L_\psi}$ and therefore, by lemma 19,

$$
\begin{aligned}
\mathbb{P} \left( \frac{1}{\sqrt{n}} \|\boldsymbol{x}^t\| \geq \kappa_x \right) &= \mathbb{P} \left( \frac{1}{\sqrt{n}} \|\boldsymbol{x}^t - \boldsymbol{h}^t - \mu_t^n \boldsymbol{X}\| + \frac{1}{\sqrt{n}} \|\boldsymbol{h}^t + \mu_t^n \boldsymbol{X}\| \geq 2\kappa_h + \frac{2\kappa}{L_\psi} \right) \\
&\leq \mathbb{P} \left( \frac{1}{\sqrt{n}} \|\boldsymbol{x}^t - \boldsymbol{h}^t - \mu_t^n \boldsymbol{X}\| \geq \frac{\kappa}{L_\psi} \right) + \mathbb{P} \left( \frac{1}{\sqrt{n}} \|\boldsymbol{h}^t + \mu_t^n \boldsymbol{X}\| \geq \kappa_h \right).
\end{aligned}
$$

Then the bound follows by (80) and (79). $\qquad \square$

### K.5  Useful lemmas

In this section we introduce a number of technical lemmas that are used to prove our main results. We include proofs only where the proof is non-standard.

**Lemma 14.** *Recall from [67], that a random variable, $X$, is sub-gaussian with variance factor $\nu$ if $\log \mathbb{E}[e^{t(X-\mathbb{E}[X])}] \leq t^2 \nu/2$ for all $t \in \mathbb{R}$. When $X \sim p_X$, we have $\nu \leq 1/4$ for $p_X \sim \mathrm{Ber}(\rho)$.*

*Proof.* By [67, Lemma 2.2 (Hoeffding's lemma)], any bounded random variable $a \leq X \leq b$ is sub-Gaussian with variance factor $(b-a)^2/4$. ☐

**Lemma 15** (Hoeffding's inequality). *If $X_1, \ldots, X_n$ are independent bounded random variables such that $a_i \leq X_i \leq b_i$, then for $\nu = 2[\sum_i (b_i - a_i)^2]^{-1}$, we have $\mathbb{P}(|\frac{1}{n}\sum_{i=1}^{n}(X_i - \mathbb{E}\{X_i\})| \geq \epsilon) \leq 2e^{-\nu n^2 \epsilon^2}$.*

**Lemma 16.** *Recall the definition of pseudo-Lipschitz functions of order $2$ given in Definition 1. The following functions $\psi : \mathbb{R}^2 \to \mathbb{R}$ are all pseudo-Lipschitz of order $2$ with pseudo-Lipschitz constant $2$.*

$$\psi_1(a,b) = (a-b)^2, \qquad \psi_2(a,b) = b^2, \qquad \psi_3(a,b) = ab. \qquad (109)$$

*Proof.* Verifying the pseudo-Lipschitz property for the functions in (109) is straightforward, so we omit the details. ☐

**Lemma 17.** *Recall the definition of pseudo-Lipschitz functions of order $2$ given in Definition 1. Let $f_t$ be the conditional expectation denoiser in (9) with Lipschitz constant $L_f^n$ and let $\psi : \mathbb{R}^2 \to \mathbb{R}$ be a pseudo-Lipschitz of order $2$ function with constant $L_\psi$. The following functions $\phi : \mathbb{R}^2 \to \mathbb{R}$ are all pseudo-Lipschitz of order $2$ with the stated pseudo-Lipschitz constants.*

$$\phi_1(a,b) = \psi(a + \mu_t^n b, b), \qquad L_{\phi_1} = 2L_\psi(1 + \mu_t^n)^2, \qquad (110)$$

$$\phi_2(a,b) = \psi(a, f_t(b)), \qquad L_{\phi_2} = 3L_\psi \max\{1, L_f\}, \qquad (111)$$

$$\phi_3(a,b) = (\mu_t^n)^{-1}a + b, \qquad L_{\phi_3} = \sqrt{2}\max\{1, (\mu_t^n)^{-1}\}, \qquad (112)$$

$$\phi_4(a,b) = (a + \mu_t^n b)^2, \qquad L_{\phi_4} = 2\max\{1, (\mu_t^n)^2\}. \qquad (113)$$

*Proof.* For function $\phi_1$ in (110), first notice

$$\begin{aligned}
|\phi_1(a,b) - \phi_1(\widetilde{a}, \widetilde{b})| &= |\psi(a + \mu_t^n b, b) - \psi(\widetilde{a} + \mu_t^n \widetilde{b}, \widetilde{b})| \\
&\leq L_\psi(1 + \|(a + \mu_t^n b, b)\| + \|(\widetilde{a} + \mu_t^n \widetilde{b}, \widetilde{b})\|) \times \|(a + \mu_t^n b, b) - (\widetilde{a} + \mu_t^n \widetilde{b}, \widetilde{b})\|.
\end{aligned} \qquad (114)$$

Next notice

$$\|(a + \mu_t^n b, b) - (\widetilde{a} + \mu_t^n \widetilde{b}, \widetilde{b})\| \leq |a - \widetilde{a}| + (1 + \mu_t^n)|b - \widetilde{b}| \leq \sqrt{2}(1 + \mu_t^n)\|(a,b) - (\widetilde{a}, \widetilde{b})\|,$$

and $\|(a + \mu_t^n b, b)\| \leq |a + \mu_t^n b| + |b| \leq |a| + (1 + \mu_t^n)|b| \leq \sqrt{2}(1 + \mu_t^n)\|(a,b)\|$. Thus, from (114), we have result (110):

$$|\phi_1(a,b) - \phi(\widetilde{a}, \widetilde{b})| \leq 2L_\psi(1 + \mu_t^n)^2(1 + \|(a,b)\| + \|(\widetilde{a}, \widetilde{b})\||) \times \|(a,b) - (\widetilde{a}, \widetilde{b})\|.$$

For function $\phi_2$ in (111), first notice

$$\begin{aligned}
|\phi_2(a,b) - \phi_2(\widetilde{a}, \widetilde{b})| &= |\psi(a, f_t(b)) - \psi(\widetilde{a}, f_t(\widetilde{b}))| \\
&\leq L_\psi(1 + \|(a, f_t(b))\| + \|(\widetilde{a}, f_t(\widetilde{b}))\|)\|(a, f_t(b)) - (\widetilde{a}, f_t(\widetilde{b}))\|.
\end{aligned} \qquad (115)$$

Next, notice that since $f_t(\cdot)$ is a Lipschitz function with constant $L_f$,

$$\begin{aligned}
\|(a, f_t(b)) - (\widetilde{a}, f_t(\widetilde{b}))\|^2 &= |f_t(b) - f_t(\widetilde{b})|^2 + |a - \widetilde{a}|^2 \\
&\leq L_f^2|b - \widetilde{b}|^2 + |a - \widetilde{a}|^2 \leq \max\{1, L_f^2\}\|(a,b) - (\widetilde{a}, \widetilde{b})\|^2,
\end{aligned}$$

and since our denoiser of interest $f_t$ in (9) is such that $|f_t(x)| \leq 1$,

$$\|(a, f_t(b))\| \leq |a| + |f_t(b)| \leq |a| + 1 \leq (1 + |a| + |b|) \leq \sqrt{2}(1 + \|(a,b)\|).$$

Thus, from (116),

$$|\widetilde{\psi}(a,b) - \widetilde{\psi}(\widetilde{a},\widetilde{b})| \leq 3L_\psi \max\{1, L_f\}(1 + \|(a,b)\| + \|(\widetilde{a},\widetilde{b})\|)\|(a,b) - (\widetilde{a},\widetilde{b})\|. \qquad (116)$$

Next, the bound for function $\phi_3$ in (112) is straightforward:

$$|\phi_3(a,b) - \phi_3(\widetilde{a},\widetilde{b})| = \left|(\mu_t^n)^{-1}a + b - (\mu_t^n)^{-1}\widetilde{a} - \widetilde{b}\right|$$

$$\leq (\mu_t^n)^{-1}|a - \widetilde{a}| + |b - \widetilde{b}| \leq \sqrt{2}\max\{1, (\mu_t^n)^{-1}\}\|(a,b) - (\widetilde{a},\widetilde{b})\|.$$

Finally, for function $\phi_4$ in (113), first notice

$$|\phi_4(a,b) - \phi_4(\widetilde{a},\widetilde{b})| = \left|(a + \mu_t^n b)^2 - (\widetilde{a} + \mu_t^n\widetilde{b})^2\right|$$

$$\leq \left|(a + \mu_t^n b) - (\widetilde{a} + \mu_t^n\widetilde{b})\right|\left|(a + \mu_t^n b) + (\widetilde{a} + \mu_t^n\widetilde{b})\right|$$

$$\leq 2\max\{1, (\mu_t^n)^2\}(\|(a,b)\| + \|(\widetilde{a},\widetilde{b})\|)(\|(a,b) - (\widetilde{a},\widetilde{b})\|),$$

where the final inequality uses that $|a + \mu_t^n b| \leq \sqrt{2}\max\{1, \mu_t^n\}\|(a,b)\|$ giving

$$\left|(a + \mu_t^n b) + (\widetilde{a} + \mu_t^n\widetilde{b})\right| \leq \sqrt{2}\max\{1, \mu_t^n\}(\|(a,b)\| + \|(\widetilde{a},\widetilde{b})\|)$$

and the fact that

$$\left|(a + \mu_t^n b) - (\widetilde{a} + \mu_t^n\widetilde{b})\right| \leq |a - \widetilde{a}| + \mu_t^n|b - \widetilde{b}| \leq \sqrt{2}\max\{1, \mu_t^n\}\|(a,b) - (\widetilde{a},\widetilde{b})\|.$$

$\square$

**Lemma 18.** *Recall the definition of pseudo-Lipschitz functions of order 2 given in Definition 1. The conditional expectation denoiser in (9) is Lipschitz with constant $L_f = \sqrt{\lambda_n}$ when $X_0^n \sim P_{X,n}$ and $P_{X,n}$ is either $\mathrm{Ber}(\rho_n)$ or Bernoulli-Rademacher and $\frac{\partial}{\partial x}f_t(x) = \sqrt{\lambda_n}f_t(x)(1 - f_t(x))$. Moreover, the Lipschitz constant can be strengthened to $\sqrt{\lambda_n}\rho_n$ on $x \in (-\infty, \frac{\mu_t^n}{2})$ and $f_t(0) \leq \rho_n$.*

*Proof.* First, recall that $f_t(\cdot)$ is the conditional expectation denoiser given in (9),

$$f_t(x) = \mathbb{E}\{X_0^n \mid \sqrt{\lambda_n}\tau_t^n X_0^n + \sqrt{\tau_t^n}Z = x\}.$$

Notice that for either the Bernoulli or Bernoulli-Rademacher case, we have that $|f_t(x)| \leq 1$ for all $x \in \mathbb{R}$ since $X_0^n \in \{-1, 0, 1\}$.

First consider $P_{X,n} \sim \mathrm{Ber}(\rho_n)$ and we show that $f_t(\cdot)$ is Lipschitz continuous with Lipschitz constant $\sqrt{\lambda_n}$. Let $\phi(x)$ denote the standard gaussian density evaluated at $x$. First, by Bayes' Rule,

$$f_t(x) = \mathbb{E}\{X_0^n \mid \sqrt{\lambda_n}\tau_t^n X_0^n + \sqrt{\tau_t^n}Z = x\}$$

$$= \mathbb{P}(X_0^n = 1 \mid \sqrt{\lambda_n}\tau_t^n X_0^n + \sqrt{\tau_t^n}Z = x) = \frac{\rho_n\phi\left(\frac{x - \sqrt{\lambda_n}\tau_t^n}{\sqrt{\tau_t^n}}\right)}{(1 - \rho_n)\phi\left(\frac{x}{\sqrt{\tau_t^n}}\right) + \rho_n\phi\left(\frac{x - \sqrt{\lambda_n}\tau_t^n}{\sqrt{\tau_t^n}}\right)}. \qquad (117)$$

Now notice that $\frac{\partial}{\partial x}\phi\left(\frac{x-a}{b}\right) = -\frac{(x-a)}{b^2}\phi\left(\frac{x-a}{b}\right)$. Using this and the representation above,

$$\frac{\partial}{\partial x}f_t(x) = \frac{\partial}{\partial x}\left[\frac{\rho_n\phi\left(\frac{x - \sqrt{\lambda_n}\tau_t^n}{\sqrt{\tau_t^n}}\right)}{(1 - \rho_n)\phi\left(\frac{x}{\sqrt{\tau_t^n}}\right) + \rho_n\phi\left(\frac{x - \sqrt{\lambda_n}\tau_t^n}{\sqrt{\tau_t^n}}\right)}\right]$$

$$= \frac{-f_t(x)}{\tau_t^n}\left[(x - \sqrt{\lambda_n}\tau_t^n) - \frac{x(1 - \rho_n)\phi\left(\frac{x}{\sqrt{\tau_t^n}}\right) + \rho_n(x - \sqrt{\lambda_n}\tau_t^n)\phi\left(\frac{x - \sqrt{\lambda_n}\tau_t^n}{\sqrt{\tau_t^n}}\right)}{(1 - \rho_n)\phi\left(\frac{x}{\sqrt{\tau_t^n}}\right) + \rho_n\phi\left(\frac{x - \sqrt{\lambda_n}\tau_t^n}{\sqrt{\tau_t^n}}\right)}\right]$$

$$= \frac{-f_t(x)(x - \sqrt{\lambda_n}\tau_t^n)}{\tau_t^n}\left[\frac{(1 - \rho_n)\phi\left(\frac{x}{\sqrt{\tau_t^n}}\right)\left[1 - \frac{x}{(x - \sqrt{\lambda_n}\tau_t^n)}\right]}{(1 - \rho_n)\phi\left(\frac{x}{\sqrt{\tau_t^n}}\right) + \rho_n\phi\left(\frac{x - \sqrt{\lambda_n}\tau_t^n}{\sqrt{\tau_t^n}}\right)}\right]$$

$$= \sqrt{\lambda_n}f_t(x)(1 - f_t(x)).$$

$$(118)$$

Therefore, using (118), we see that $\left|\frac{\partial}{\partial x}f_t(x)\right| \leq \sqrt{\lambda_n}$ and it follows that $f_t(\cdot)$ is Lipschitz continuous with Lipschitz constant $\sqrt{\lambda_n}$.

The fact that $f_t(\cdot)$ is Lipschitz continuous with Lipschitz constant $\sqrt{\lambda_n}$ can be shown similarly for the case where $P_{X,n}$ is Bernoulli-Rademacher.

Finally, notice that from (117) we have

$$f_t(x) = \frac{\rho_n}{(1-\rho_n)\exp\left\{\frac{1}{2}(\lambda_n\tau_t^n - 2x\sqrt{\lambda_n})\right\} + \rho_n}. \tag{119}$$

Then since $e^x \geq 1 + x$ (which can be seen by showing that $f(x) = e^x - (1+x)$ has a minimum at $f(0) = 0$),

$$f_t(x) \leq \frac{\rho_n}{(1-\rho_n)(1 + \frac{1}{2}(\lambda_n\tau_t^n - 2x\sqrt{\lambda_n})) + \rho_n} = \frac{\rho_n}{1 + \frac{1}{2}(1-\rho_n)(\lambda_n\tau_t^n - 2x\sqrt{\lambda_n})}.$$

The above implies that $f_t(0) \leq \rho_n$, and further, since

$$(1-\rho_n)(\lambda_n\tau_t^n - 2x\sqrt{\lambda_n}) \geq 0 \quad \text{when} \quad x \leq \frac{\sqrt{\lambda_n}\tau_t^n}{2},$$

we find the bound $0 \leq f_t(x) \leq \rho_n$ when $x \leq \frac{\sqrt{\lambda_n}\tau_t^n}{2}$.

Therefore, by (118), we have $|\frac{\partial}{\partial x}f_t(x)| \leq \sqrt{\lambda_n}f_t(x) \leq \sqrt{\lambda_n}\rho_n$ and it follows that $f_t(\cdot)$ is Lipschitz continuous with Lipschitz constant $\sqrt{\lambda_n}\rho_n$ on $x \in (-\infty, \frac{\mu_t^n}{2})$. $\qquad\square$

The proof of the following two lemmas can be found in [63, appendix A].

**Lemma 19** (Concentration of Sums). *If random variables $X_1, \ldots, X_M$ satisfy $P(|X_i| \geq \epsilon) \leq e^{-n\kappa_i\epsilon^2}$ for $1 \leq i \leq M$, then*

$$\mathbb{P}\Big(|\sum_{i=1}^{M}X_i| \geq \epsilon\Big) \leq \sum_{i=1}^{M}\mathbb{P}\left(|X_i| \geq \frac{\epsilon}{M}\right) \leq Me^{-n(\min_i \kappa_i)\epsilon^2/M^2}.$$

**Lemma 20** (Concentration of Powers). *Assume $c > 0$ and $0 < \epsilon \leq 1$. Then, if $\mathbb{P}(|X_n - c| \geq \epsilon) \leq e^{-\kappa n\epsilon^2}$, it follows that $\mathbb{P}(|X_n^2 - c^2| \geq \epsilon) \leq e^{-\kappa n\epsilon^2/[1+2c]^2}$.*

# L Algorithmic AMP phase transition regime

In this appendix we show that the right-hand side of the bound in theorem 2 for signal strength and sparsity scaling like $\lambda_n\rho_n^2 = w$ and $\rho_n = \Omega((\ln n)^{-\alpha})$ with $w, \alpha \in \mathbb{R}_+$, tends to zero as $n \to +\infty$. We focus on the Bernoulli prior case but the arguments generalizes to Bernoulli-Rademacher prior.

Let us first upper bound $\gamma_n^t$ in terms of $\lambda_n$ and $\rho_n$ in the Bernoulli case. First we use the bound $|f_t'(x)| \leq \sqrt{\lambda_n}$ (see lemma 18) to bound

$$\max\{1, \hat{b}_1\}\max\{1, \hat{b}_2\}\cdots\max\{1, \hat{b}_{t-1}\} \leq \lambda_n^{\frac{t-1}{2}}.$$

From the explicit AMP iteration (see appendix G second formula for example) we have $\tau_t^n \leq \rho_n$. Since $\nu_n \leq 1/4$ (see lemma 14) we get $(\nu^n + \tau_1^n)(\nu^n + \tau_1^n + \tau_2^n)\cdots(\nu^n + \sum_{i=1}^{t}\tau_i^n) \leq (\frac{1}{4} + \rho_n)(\frac{1}{4} + 2\rho_n)\cdots(\frac{1}{4} + t\rho_n) \leq (\frac{1}{4} + t)!$. Putting everything together we get:

$$\gamma_n^t \leq (\frac{1}{4} + t)!\,\lambda_n^{2t-1+\frac{t-1}{2}}.$$

Now we use the scaling (which is the correct scale for the phase transition to happen) $\lambda_n = w\rho_n^{-2}$ and get:

$$\gamma_n^t \leq (\frac{1}{4} + t)!\,\frac{w^{\frac{5t-3}{2}}}{\rho_n^{5t-3}}.$$

Therefore

$$\text{bound}_t \leq CC_t \exp\left\{-\frac{c}{w^{\frac{5t-3}{2}}}\frac{c_t}{(\frac{1}{4}+t)!}\rho_n^{5t-3}n\epsilon^2\right\}.$$

Now the $t$ dependence in the constant $c_t = [C^t(t!)^C]^{-1}$ (from now on $C$ is a generic positive constant) and using Stirling's approximation $t! \approx \sqrt{2\pi t}\, t^t\, e^{-t}$ this scales at dominant order as $[C^t(t^t)^C]^{-1}$. So we have at dominant order

$$\text{bound}_t \approx CC_t \exp\{-Ce^{\pm Ct - Ct\ln t}e^{(5t-3)\ln(\rho_n)}e^{\ln n}\epsilon^2\}.$$

Now set the number of iterations to $t = o(\frac{\ln n}{\ln\ln n})$. We get $t\ln t = o(\ln n)$ so $\pm Ct - Ct\ln t = o(\ln n)$ and

$$\text{bound}_t \approx CC_t \exp\{-Ce^{-o(\ln n)}e^{o(\frac{\ln n}{\ln\ln n})\ln(\rho_n)}e^{\ln n}\epsilon^2\}. \tag{120}$$

We set $\rho_n = \Theta\left(\frac{1}{(\ln n)^\alpha}\right) = \frac{C}{(\ln n)^\alpha}$. Then $\ln\rho_n = \ln C - \alpha\ln\ln n$ and we get

$$\text{bound}_t \approx CC_t \exp\{-Ce^{-o(\ln n)}e^{o(\frac{\ln n}{\ln\ln n})(C-\alpha\ln\ln n)}e^{\ln n}\epsilon^2\}.$$

This leads to

$$\text{bound}_t \approx CC_t \exp\{-Ce^{-o(\ln n)+Co(\frac{\ln n}{\ln\ln n})-\alpha o(\ln n)}e^{\ln n}\epsilon^2\}$$
$$\approx CC_t \exp\{-Ce^{(\ln n)-(1+\alpha)o(\ln n)+Co(\frac{\ln n}{\ln\ln n})}\epsilon^2\}$$
$$\approx CC_t \exp\{-Ce^{(\ln n)[1-(1+\alpha)o(1)]+Co(\frac{1}{\ln\ln n})}\epsilon^2\}$$
$$\approx CC_t \exp\{-Cn^{1-o_\alpha(1)}\epsilon^2\}.$$

One can check that the prefactor $C_t = [C^t(t!)^C]$ does not change the dominant order for $t = o(\frac{\ln n}{\ln\ln n})$. This shows that the bound vanishes as $n \to +\infty$ for $\lambda = w\rho_n^{-2}$ and $\rho_n = \Theta(\frac{1}{(\ln n)^\alpha})$ for any $\alpha \geq 0$. As seen from (120) the bound worsen with decreasing $\rho_n$. So the result extends to $\rho_n = \Omega(\frac{1}{(\ln n)^\alpha})$.

Note also that in the case of the rescaled bound of remark 1 below theorem 2, the previous derivation is unchanged, up to the constant appearing in the $o_\alpha(1)$ that is changed some other $o_\alpha(1)$ (for $n$ big enough). Indeed, because $\rho_n = \Omega(\frac{1}{(\ln n)^\alpha})$ the $\rho_n^2$ or $\rho_n^4$ appearing in the rescaled bound can be absorbed in the $o_\alpha(1)$ of the previous derivation, for $n$ large enough.