[Reviews · NeurIPS 2020]

Review 1

Summary and Contributions: The authors consider the recovery problem for spiked Wigner model. Assuming that the spike is sparse, the authors prove a sharp phase transition in the mutual information of the spike and the data. It is also proved that the mean-square error of AMP algorithm exhibits a phase transition that is different from the transition of the mutual information.

Strengths: The authors rigorously analyze the sparse spiked Wigner model and find the correct scaling of the critical SNR. The analysis is profound and can be adapted in other related problems.

Weaknesses: In its current form, the scaling regimes for the statistical phase transition and the AMP phase transition do not match, and thus the statistical-to-algorithm gap is not proved. It is desirable to mention whether the condition \beta<1/6 is simply a technical one or not.

Correctness: The claims seem correct, but I did not check all the detail in the supplementary materials.

Clarity: The paper is very clearly written.

Relation to Prior Work: It is clearly discussed in the introduction.

Reproducibility: Yes

Additional Feedback: In Theorem 1, the limit of the mutual information is given as the infimum of the potential function. I guess it would be better if it is written in terms of the singular function given in line 213, since the authors focus the Bernoulli and Rademacher-Bernoulli cases only. It is rather unusual that a theorem is given in the appendix. edit: I have no further change in my review after rebuttal.


Review 2

Summary and Contributions: The authors studied the sparse spiked Wigner matrix model in the case where the number of non-zero entries is sublinear compared to the size of the system. The result obtained by the authors was heuristically found in ref. [52] expanding the result for a number of non-zero entries that scales a the input dimension. The authors present a rigorous derivation of the result.

Strengths: The sublinear sparsity regime is poorly understood from the theoretical point of view. Therefore the result presented in the manuscript is an interesting contribution to the field. The techniques presented could eventually be extended to deal with more complex models.

Weaknesses: Nothing to report.

Correctness: Claims and methods are correct.

Clarity: The paper is well written. ----- I agree with the other reviewer that was frustrating not to read the scaling on the SNR for several pages in the paper. I would appreciate if the authors put forward the information since the beginning.

Relation to Prior Work: Previous works are reported and the contributions are clearly discussed.

Reproducibility: Yes

Additional Feedback: The supplementary should be reorganized to contain only the appendix.


Review 3

Summary and Contributions: The authors present a rigorous analysis of mutual information and AMP for estimation sparse, rank-one matrices from iid Gaussian noise corrupted measurements in the sublinear sparsity regime.

Strengths: The mathematical depth of the paper is significant.

Weaknesses: The topic is quite far from the interests of most Neurips attendees.

Correctness: Although I did not check the proofs, the results appear to be correct and agree with previous non-rigorous analyses.

Clarity: There are several places in the introduction where improvements can be made. The term "true sparsity" as used on page 2 is very confusing and should be avoided. The authors are simply talking about sparsity in the sublinear regime. Please stick to the standard terminology. The details of the sublinear regime do not appear until page 4. As a reader, I was frustrated with the lack of information about the behavior of $n rho_n$ as $n$ tends to infinity. All we are told is that it does not grow to infinity. As described around (2) and (3), the authors prove that phase transitions exist for particular sequences of $lambda_n$ and $rho_n$. The reader is left wonder about what happens with other sequences. I would think that it is possible to prove that no phase transition exists for some sequences, which suggest that a table could be made to catalog when phase transitions exist, when they don't exist, and when the behavior is unknown. I am confused by the statement in line 188: "as long as $rho_n -> infty$". In the previous paragraph, the quantity $rho_n$ had very specific rates of decay with $n$, while here the claim seems to be for any rate of decay. This could benefit from some clarification.

Relation to Prior Work: The most closely related work is well discussed. However, the authors seem to give the impression that sublinear sparsity is a very new idea and discussed only in 2 papers. In fact, it is well studied, but perhaps not in the same way that the authors study it. Thus, the paper would benefit from a more complete discussion of prior work on sublinear sparsity analysis.

Reproducibility: Yes

Additional Feedback: ** I have read the other reviews and the author's response, and my opinion remains the same. **

[Author Response · NeurIPS 2020]

Reviewer 1: *In its current form, the scaling regimes for the statistical phase transition and the AMP phase transition do*
*not match, and thus the statistical-to-algorithm gap is not proved.*

We prove that the stat. trans. happens for $\lambda_n = \Theta(|\ln \rho_n|/\rho_n)$, with $\rho_n = \Omega(n^{-\beta})$, $\beta$ small enough (in the paper Thm
1 states $\rho_n = \Theta(n^{-\beta})$: this can be relaxed and we only require the weaker condition $\rho_n = \Omega(n^{-\beta})$ as seen from Thm
3; which will be corrected). We also check the algo. trans. happens for $\lambda_n = \Theta(1/\rho_n^2)$ for $\rho_n = \Omega((\ln n)^{-\alpha})$ for any
$\alpha \geq 0$ (same relaxation $\Theta \to \Omega$). The fact that the scaling regimes for $\lambda_n$ of the stat. and algo. transitions are different
is intrinsic to the problem, and actually does prove the presense of a diverging (as $\rho_n \to 0$) statistical-to-algorithm gap.
After these amendments, since Thm 1 and Corollary 1 hold for $\rho_n = \Omega(n^{-\beta})$ and thus for $\rho_n = \Omega((\ln n)^{-\alpha})$, then
both trans. (and thus the stat.-to-algo. gap) are proven for $\rho_n = \Omega((\ln n)^{-\alpha})$. All that will be clarified, thank you!

*It is desirable to mention whether the condition $\beta < 1/6$ is simply a technical one or not.*

This restriction on the sparsity is probably a consequence of the sub-optimality of our analysis. We will mention it.

*In Thm 1 the limit of the MI is given as the infimum of the potential function. It would be better written in terms of the*
*singular function given in line 213, since the authors focus the Bernoulli and Rademacher-Bernoulli cases.*

Thm 1 applies more generically than for Ber and Rad-Ber signals. It applies to signals verifying the hyp. in the setting
(line 61). It is true that the all-or-nothing is then studied, starting from Thm 1, only for Ber and Rad-Ber signals, but
nevertheless Thm 1 applies more generically as it precisely bounds the deviation of the Mut. Info. from a simple
single-letter formula for any finite size $n$, and for a much wider class of signals. Moreover the singular function of line
213 is equal to the infimum of the potential function only in the limit $n \to \infty, \rho_n \to 0$ while, again, Thm 1 is a finite
size bound that contains more than just the asymptotic limit. We hope this clarifies our choice of presentation.

*It is rather unusual that a theorem is given in the appendix.*

You are probably talking about appendix A in which we simply give an even more general form of Thm 1 that includes
not only the specific scaling regime (2) where the all-or-nothing statistical transition happens, but also a wider class of
scaling regimes, see (15). As the formulation of this more general theorem is long and not directly used in the main part
we made the choice to defer it to an appendix for people interested in more general bounds.

Reviewer 4: *The topic is quite far from the interests of most Neurips attendees.*

The present submission is part of a recent line of work mostly concerned with information theoretic results (note that we
bring about, in addition of information-theoretic results, strong rigorous algorithmic bounds). But we believe that it is
only a matter of time before the ML community realize the relevance of studying such scaling regimes. Let us cite just
as an example among others the recent work of Goldt et al `https://arxiv.org/pdf/1909.11500.pdf` that clearly
indicates that in the high-sparsity regime, which translates in their setting into a low intrinsic dimension of the data,
learning performance strongly increases, see their Fig. 5. Moreover our work concerns sparse PCA, a key ML problem.

*The term "true sparsity" as used on page 2 is very confusing and should be avoided. The authors are simply talking*
*about sparsity in the sublinear regime. Please stick to the standard terminology.*

This terminology is used to contrast with recent literature that also studies a sparse limit, but in the linear regime, with a
sparsity tending to 0 only after the large size limit has been taken. We will add a few words to clarify why we wish to
use this terminology.

*The details of the sublinear regime do not appear until page 4. As a reader, I was frustrated with the lack of information*
*about the behavior of $n\rho_n$ as $n$ tends to infinity. All we are told is that it does not grow to infinity.*

We will add details earlier in the paper, thanks for the suggestion.

*As described around (2) and (3), the authors prove that phase transitions exist for particular sequences of $\lambda_n$ and $\rho_n$.*
*The reader is left wonder about what happens with other sequences. I would think that it is possible to prove that no*
*phase transition exists for some sequences, which suggest that a table could be made to catalog when phase transitions*
*exist, when they don't exist, and when the behavior is unknown.*

That is a good point. But this question is out of the scope of the paper, as it is not clear yet to us what is the criterion /
property that generically underlies the presence or not of the all-or-nothing. This question is clearly to be considered for
future work, thanks for that.

*I am confused by the statement in line 188: "as long as $\rho_n \to \infty$". In the previous paragraph, the quantity $\rho_n$ had very*
*specific rates of decay with $n$, while here the claim seems to be for any rate of decay. This could benefit from some*
*clarification.*

We agree the sentence is not precise: we indeed prove things for specific regimes and the sentence will be reformulated.

[Meta-Review · NeurIPS 2020]

The authors consider the recovery problem for spiked Wigner model, and they prove a sharp phase transition in the mutual information of the spike and the data and also that the mean-square error of AMP algorithm exhibits a phase transition that is different from the transition of the mutual information. Several suggestions for improvement are given in the reviews. Overall, the paper is a theoretical paper with nontrivial technical contribution, the results are of interest to Neurips community, and they are well presented.